# SCALECUA: SCALING OPEN-SOURCE COMPUTER USE AGENTS WITH CROSS-PLATFORM DATA

**Zhaoyang Liu**[*♠◇] **Jingjing Xie**[*◇] **Zichen Ding**[*◇] **Zehao Li**[*◇] **Bowen Yang**[*◇]
**Zhenyu Wu**[*◇] **Xuehui Wang**[◇] **Qiushi Sun**[◇] **Shi Liu**[◇] **Weiyun Wang**[◇] **Shenglong Ye**[◇]
**Qingyun Li**[◇] **Zeyue Tian**[♠] **Gen Luo**[◇] **Xiangyu Yue**[◇] **Biqing Qi**[◇] **Kai Chen**[◇]
**Bowen Zhou**[◇] **Yu Qiao**[◇] **Qifeng Chen**[†♠] **Wenhai Wang**[†◇]
♠HKUST     ◇Shanghai AI Laboratory

## ABSTRACT

Vision-Language Models (VLMs) have enabled computer use agents (CUAs) that operate GUIs autonomously with great potential. However, developing robust CUAs requires extensive in-domain knowledge about software interfaces and operations. Unlike image–text pairs that are widely available on the Internet, computer-use data, particularly operation trajectories, are rare, costly to collect. Consequently, advancement in this field remains constrained by both data scale and the limited transferability of existing VLMs. In this work, we introduce ScaleCUA, a step toward scaling open-source CUAs. It offers a large-scale dataset spanning six operating systems and 3 task domains, via a closed-loop pipeline uniting automated agents with human experts. Trained on this scaled-up data, ScaleCUA can operate seamlessly across platforms. Specifically, it delivers substantial gains over baselines (+26.6 on WebArena-Lite-v2, +10.7 on ScreenSpot-Pro) and sets new state-of-the-art results (94.4% on MMBench-GUI L1-Hard, 60.6% on OSWorld-G, 47.4% on WebArena-Lite-v2). These findings underscore the power of data-driven scaling for general-purpose cross-platform CUAs. We release data, models, and code to advance future research: `https://github.com/OpenGVLab/ScaleCUA`.

## 1 INTRODUCTION

Humans are able to interact with digital environments through graphical user interfaces (GUIs) to acquire information and accomplish tasks efficiently. The recent advances in Vision-Language Models (VLMs), which exhibit powerful capabilities in visual perception and task planning, have made it increasingly feasible to automate such interactions. Consequently, recent research has increasingly focused on computer use agents (**CUAs**), also referred to as GUI agents, aiming to autonomously operate desktop, mobile, and web platforms by relying exclusively on visual observations.

Some works (Qin et al., 2025; Anthropic, 2025; Hong et al., 2025; OpenAI, 2025; Guo et al., 2025; Hong et al., 2025) demonstrate strong performance on computer use, while they are typically built on closed-source models or inaccessible proprietary datasets. More fundamentally, effective computer use requires rich in-domain knowledge of software and operational procedures, which remains a substantial gap for current foundation models. Unlike image–text pairs that are abundantly available on the Internet, computer-use data, particularly fine-grained action trajectories, are scarce, costly to collect, and not naturally archived online. Furthermore, as software, web pages and operating systems evolve rapidly, existing trajectories face the risk of obsolescence, further limiting their utility. These challenges result in a significant bottleneck for scaling computer use agents in both data scale and model generalizability. To tackle these limitations, we make significant efforts on two aspects: (a) **constructing a large-scale, cross-platform and GUI-centric training corpus**, and (b) **developing a family of scalable, versatile foundation models for general-purpose computer use**.

We first present a *Cross-Platform Interactive Data Pipeline* composed of two synergistic loops. The *Agent-Environment Interaction Loop* enables automated agents to interact with diverse GUI environments, while the *Agent-Human Hybrid Data Acquisition Loop* integrates expert-collected

---

*Co-first authors. [†] Corresponding authors (cqf@ust.hk & wangwenhai362@gmail.com).

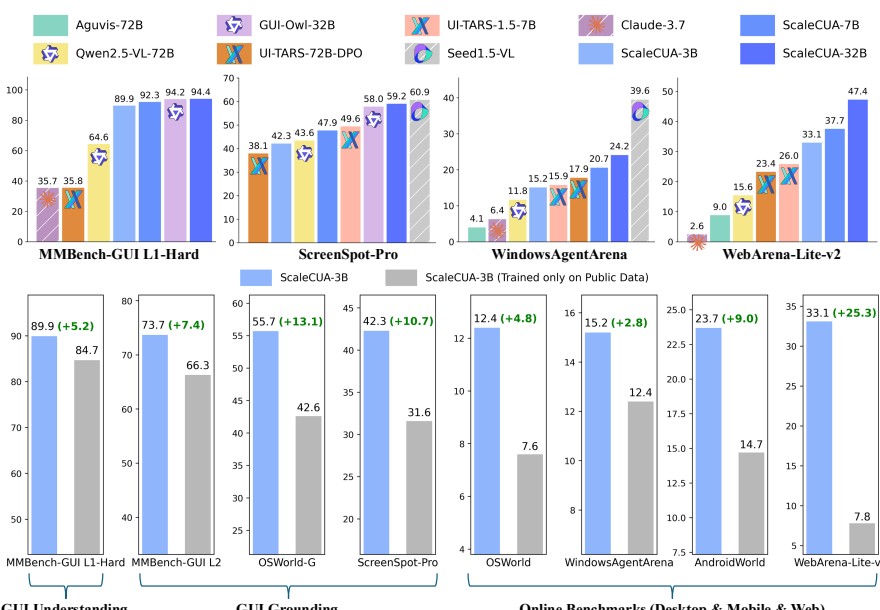

Figure 1: **Performance comparison**. The top row showcases performance overview on GUI-centric benchmarks. The bottom row demonstrates the consistent improvements from our collected data.

trajectories to ensure coverage and quality. The pipeline spans six major platforms, including Windows, macOS, Linux, Android, iOS, and Web, which facilitates the collection of rich screen-state observations, metadata (*e.g.*, A11y Trees, XLM, DOM structures, *etc.*), and raw trajectories. In this pipeline, we design a unified action space, allowing for more consistent and efficient interaction with diverse real-world environments. Leveraging this infrastructure, we curate and annotate a comprehensive training dataset with advanced VLMs such as Claude-3.7 for an open computer use dataset, covering three major task families: (a) *GUI Understanding* with 471K examples covering regional captioning, OCR, and layout comprehension, *etc.*; (b) *GUI Grounding* with 17.1M training samples supporting more accurate UI element localization; and (c) *Task Completion* with over 15K weak-semantic trajectories and 4K high-level goal-directed trajectories.

Building upon this corpus, we train a series of base agent models termed as **ScaleCUA** with Qwen2.5-VL (Bai et al., 2025). It supports three inference paradigms to offer enhanced flexibility and compatibility with agent frameworks: (a) a *Grounding Mode*, which focuses on locating UI elements based on textual descriptions, allowing for integration with more powerful planners, (b) a *Direct Action Mode*, which enables efficient task completion by directly generating executable actions without additional intermediate reasoning and (c) a *Reasoned Action Mode*, which enhances task planning with Chain-of-Thought process before generating the following action. We conduct extensive quantitative studies to investigate how different data sources, diverse training tasks, agent designs, *etc.*, influence agent performance. Our findings highlight the benefits of data augmentation, weak semantic trajectories, and general reasoning data for enhancing planning capabilities. As previous studies (Xu et al., 2024; Qin et al., 2025; Anthropic, 2025) also probe into the important research questions with limited open-sourced training data or under closed conditions with proprietary data, our investigations aim to provide foundational and unified insights for advancing vision-based computer automation.

Our contributions are summarized as follows:

1) We curate a cross-platform computer use dataset, collected via an interactive data pipeline that integrates automated agents with human experts. It covers six major platforms (Windows, macOS, Linux, Android, iOS, and Web) and three GUI-centric task domains (*i.e.*, understanding, grounding, and task completion), which provide a robust foundation for studying and training universal CUAs.

2) We develop ScaleCUA, a family of robust base agent models that unify perception, reasoning, and action into a single model. It supports flexible inference paradigms, including grounding, direct action, and reasoned action, along with a unified action space for seamless cross-platform interaction.

3) We conduct a comprehensive evaluation spanning understanding, grounding, and end-to-end task completion across several platforms. The results not only demonstrate that our agents can achieve competitive performance but also provide fundamental insights for developing more powerful CUAs.

## 2 RELATED WORK

**Vision-Language Models (VLM).** Recent years have witnessed rapid progress in VLMs spanning proprietary APIs (Team et al., 2023; 2024; Anthropic, 2024a; xAI, 2025; OpenAI, 2023; Hurst et al., 2024) and open-source models (Wang et al., 2024; Bai et al., 2025; Chen et al., 2024b; Zhu et al., 2025; Xiaomi, 2025; Team et al., 2025a; MetaAI, 2025), greatly expanding task coverage. Some VLMs (Team et al., 2025c; Guo et al., 2025; Bai et al., 2025; Xiaomi, 2025; Wang et al., 2025a) integrate GUI knowledge during pre-training or SFT, thereby gaining explicit computer-use abilities. Yet, despite strong generalization and planning capabilities, they still rely on proprietary GUI corpora.

**Computer Use Agents (CUAs) / GUI Agents.** Advances in general-purpose VLMs (e.g., GPT-4o) have enabled modular CUAs that decompose decision-making into *planner–grounder* roles (Cheng et al., 2024; Hong et al., 2024; Lu et al., 2024b; Yu et al., 2025; Wu et al., 2025; Gou et al., 2024; Zhang et al., 2025b; Wu et al., 2024b; Zhou et al., 2025). A VLM-based planner predicts high-level operations, while a specialized grounder localizes targets. Enhancements such as incorporating action histories (Yang et al., 2024) improve contextual grounding, and multi-agent *agentic workflows* (Wu et al., 2023b; Li et al., 2023; Hong et al., 2023; Wu et al., 2024a; Liu et al., 2025a; Zhao et al., 2025; Agashe et al., 2025; Chen et al., 2025b) coordinate planning, reflection, and memory. Despite strong performance, such workflows incur high computational latency and token cost, remaining bounded by underlying VLM capacity. In contrast, *native agents* (Xu et al., 2024; Wu et al., 2024b; Sun et al., 2024b; Qin et al., 2025; Luo et al., 2025; Liu et al., 2025b; Sun et al., 2025) integrate planning and grounding end-to-end, directly predicting low-level executable actions from raw visual inputs. Systems such as *AGUVIS* (Xu et al., 2024) and *UI-TARS* (Qin et al., 2025) trained on extensive trajectories show strong reasoning and adaptability. Native agents thus achieve tighter perception–action alignment while also benefiting modular setups. Our work extends this direction by training cross-platform base models and open-sourcing all data.

**GUI Datasets.** Open-source datasets have accelerated CUA's development by capturing diverse interactions and instruction-following behaviors. For *mobile*, *RICO* (Deka et al., 2017) contains 70k+ Android screens, *AITW* (Rawles et al., 2023) offers ∼715k demonstrations with 30k commands, and AitZ (Zhang et al., 2024) provides 18,643 screen–action pairs with action-thought annotations. In the *web* domain, *MiniWoB* (Shi et al., 2017) simulates diverse tasks, *WebShop* (Yao et al., 2022) collects language-driven e-commerce trajectories, and *Mind2Web* (Deng et al., 2023) scales to 137 websites and 2,350 open-ended tasks. For *desktop*, Xie et al. (2024) synthesizes 4M examples to boost grounding, and He et al. (2025) adds 312 human-annotated, trajectory-boosted samples. Scalable data generation includes *OS-Genesis* (Sun et al., 2024b) for mobile/web exploration and *AGUVIS* (Xu et al., 2024) for multimodal grounding–reasoning corpora. Tutorial-style datasets mitigate scarcity: *META-GUI*(Sun et al., 2022) introduces dialogue-based annotations; *TongUI*(Zhang et al., 2025a) offers ∼143k trajectories linking instructions to screenshots; and *GUI-World* (Chen et al., 2025a) records 12k GUI videos for temporal understanding. Nevertheless, coverage and diversity remain limited, especially for desktop, posing challenges for UI element grounding and multi-step planning.

## 3 CROSS-PLATFORM INTERACTIVE DATA PIPELINE

Collecting computer use trajectories is exceptionally costly and inefficient, primarily due to the dynamic nature of environments and their frequent dependency on task-specific resources. In this section, we elaborate on the pipeline of data collection and annotation.

### 3.1 DATA ACQUISITION

Existing computer-use datasets generally rely on either manual trajectory collection or automated search-based exploration. While manual collection (Zhang et al., 2024; Rawles et al., 2023; Deng et al., 2023; Lu et al., 2024a) yields high-quality trajectories, it is costly and difficult to scale. Automated exploration (Sun et al., 2024b) is more scalable but typically noisy. Neither approach

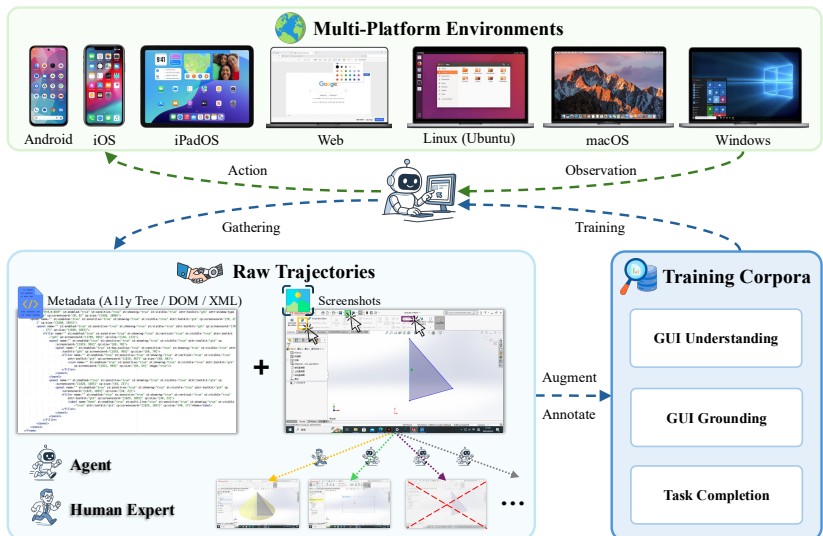

Figure 2: **Cross-platform interactive data pipeline.** Our pipeline consists of two synergistic loops: (1) the **Agent-Environment Interaction Loop**, where agents interact with multi-platform GUI environments via observation and actions; and (2) the **Agent-Human Hybrid Data Acquisition Loop**, where both autonomous agents and human experts contribute to collecting raw trajectories with screenshots and structural metadata. The resulting trajectories are then annotated and processed into several GUI-centric tasks such as understanding, grounding, and task completion.

alone achieves the required balance of quality and diversity for training versatile GUI agents. To this end, we propose a *Cross-Platform Interactive Data Pipeline* integrating automated agents with human experts. It operates in two synergistic loops as shown in Fig. 2. The *Agent-Environment Interaction Loop* enables agents or humans to interact with multi-platform environments, while the *Agent-Human Hybrid Data Acquisition Loop* merges trajectories from autonomous agents and experts.

Specifically, in the agent-environment interaction loop, we standardize observation acquisition and action execution across Windows, Ubuntu, macOS, Web browsers, Android, and iOS. This unified abstraction supports closed-loop data collection and diverse agent architectures. Platform-specific metadata is extracted from A11y Trees (Desktop), DOM (Web), and XML layout files (Android). When metadata is incomplete or restricted, as in iOS/iPadOS, OmniParser (Yu et al., 2025) estimates UI bounding boxes. In the agent-human hybrid data acquisition loop, human experts and automated agents both share the same interfaces to collect diverse trajectories. For automated agents, we evaluate two exploration strategies: VLM-driven agents (e.g., GPT-4o, Claude-3.7, *etc.*) and rule-driven random-walk agents. The former relied on proprietary VLMs, which often led to significant bias and hallucinations, especially for computer use, and thus was not used as the primary strategy for data collection. The latter performs depth-first exploration, randomly selecting actions from the available action space at each step. Heuristic pruning removes redundant or uninformative branches, broadening GUI coverage. Although these trajectories often lack clear high-level goals, their sub-sequences still yield valuable supervision for the agent. As both system-derived metadata and vision-based bounding boxes can be noisy, we complement it with expert-curated trajectories. Human experts first create a task list and then collect trajectory data in the environment. In addition, human experts are required to randomly sample and review 20% of the agent-collected trajectories after both collection and annotation to ensure quality. This is what we refer to as hybrid data acquisition. This unified pipeline decouples front-end interfaces from back-end environments, allowing collectors to efficiently switch between platforms and complete domain-specific tasks. These screenshots and metadata are then annotated into GUI-centric tasks such as understanding, grounding, and task completion, forming a robust foundation for training generalizable agents.

## 3.2 DATA ANNOTATION AND STATISTICS

This dual-loop framework collects extensive screenshots, structural metadata, and raw trajectories across Windows, macOS, Linux, Android, iOS, and Web platforms. Advanced VLMs (e.g., GPT-4o

Table 1: Datasets comparisons on computer-use datasets in terms of platform coverage, data types (Understanding, Grounding and Trajectories), and collection methods.

| Data source | Platform | | | Understanding | Grounding | Trajectories | | Collection Method |
|---|---|---|---|---|---|---|---|---|
| | Desktop | Mobile | Web | #Samples | #Elements | #Samples | Avg. Steps | |
| SeeClick (2024) | ✗ | ✓ | ✓ | – | 763K | – | – | Auto |
| GUIEnv (2024a) | ✗ | ✗ | ✓ | – | 10.7M | – | – | Auto |
| Widget Captioning (2020b) | ✗ | ✓ | ✗ | – | 163K | – | – | Human |
| RicoSCA (2020a) | ✗ | ✓ | ✗ | – | 178K | – | – | Auto |
| RICO (2017) | ✗ | ✓ | ✗ | – | 72K | – | – | Hybrid |
| OmniACT (2024) | ✓ | ✗ | ✓ | – | 9.8K | – | – | Auto |
| GUIAct (2024a) | ✗ | ✗ | ✓ | – | 67K | 5.7K | 6.7 | Auto |
| AitZ (2024) | ✗ | ✓ | ✗ | – | – | 2.5K | 6.0 | Human |
| AndroidControl (2024a) | ✗ | ✓ | ✗ | – | – | 13.6K | 5.5 | Human |
| GUI Odyssey (2024a) | ✗ | ✓ | ✗ | – | – | 7.7K | 15.3 | Human |
| AMEX (2024) | ✗ | ✓ | ✗ | – | – | 3.0K | 11.9 | Human |
| AitW (2023) | ✗ | ✓ | ✗ | – | – | 2.3K | 8.1 | Human |
| OS-Atlas (2024b) | ✗ | ✓ | ✓ | – | 13.6M | – | – | Auto |
| OS-Genesis (2024b) | ✗ | ✓ | ✓ | – | – | 2.5K | 6.4 | Auto |
| JEDI (2025) | ✓ | ✗ | ✗ | 877K | 3.1M | – | – | Auto |
| AgentNet (2025b) | ✓ | ✗ | ✗ | – | – | 22K | – | Human |
| Ours | ✓ | ✓ | ✓ | 471K | 17.1M | 19.0K | 9.0 | Hybrid |

and Claude-3.7) are then used to annotate the corpus into three major task families: **GUI Understanding**, **GUI Grounding**, and **Task Completion**. At the element and screenshot levels, understanding tasks cover visual description, OCR, layout reasoning, interface captioning, and state transition analysis, while grounding tasks provide point, bounding-box, and action-level supervision to align natural-language instructions with UI regions. Task Completion is composed of a) weak-semantic trajectories derived from rule-driven exploration that supply low-cost navigation patterns, and b) expert-curated demonstrations with realistic, goal-directed signals for reasoning and planning. Augmentation techniques such as element cropping, synthetic resolution scaling, and reasoning-prompt enrichment further diversify the training data. The final corpus spans 471K GUI-understanding examples, over 17.1M grounding annotations, and 19K trajectories averaging 9 steps each. As summarized in Table 1, we believe this dataset enables balanced evaluation of understanding, grounding, and task completion across all platforms. More statistics are shown in the Appendix.

**Discussions**. By leveraging a dual-loop pipeline, we ensure coverage of low-level element recognition, mid-level grounding, and high-level task planning. Compared with current works (Sun et al., 2024b; Wu et al., 2024b; Zhang et al., 2024; Rawles et al., 2023), we explore more diverse data collection strategies (human experts and automated agents) and cover a broader range of platforms (desktop, mobile, and web). Specifically, for the random-walk agent, we designed a more efficient algorithm through extensive experimentation and iterative improvements, significantly enhancing both data collection efficiency and GUI coverage. With this pipeline, we have collected over 2M raw screenshots across multiple platforms. We acknowledge that this pipeline is conceptually straightforward, but executing it across heterogeneous operating systems and software ecosystems entails substantial non-trivial engineering. Our contributions in the data pipeline are threefold: 1) We propose a robust and scalable data acquisition pipeline that balances automation and expert supervision, along with a set of effective heuristics improving data diversity and quality. 2) We summarize a comprehensive guideline covering platform-specific issues and their resolutions in the Appendix, which significantly improves the purity and efficiency of data collection. 3) We commit to releasing all data, ensuring transparency and reusability for future research. Generally, we emphasize that our work delivers a practically validated, cross-platform solution addressing real-world bottlenecks in scaling computer-use agents. Despite involving many engineering-oriented efforts, we still aim to share these experiences and provide valuable guidance for future developments in this field.

# 4 THE DESIGN OF COMPUTER USE AGENTS

## 4.1 TASK DEFINITION

VLMs allow agents to achieve pixel-level perception and interaction on graphical user interfaces. We formulate the interaction between the agent and environment at one time step as follows:

$$a_t = \pi_\theta(task, o_t, h_{<t}), \quad o_{t+1} = \mathcal{E}(a_t), \tag{1}$$

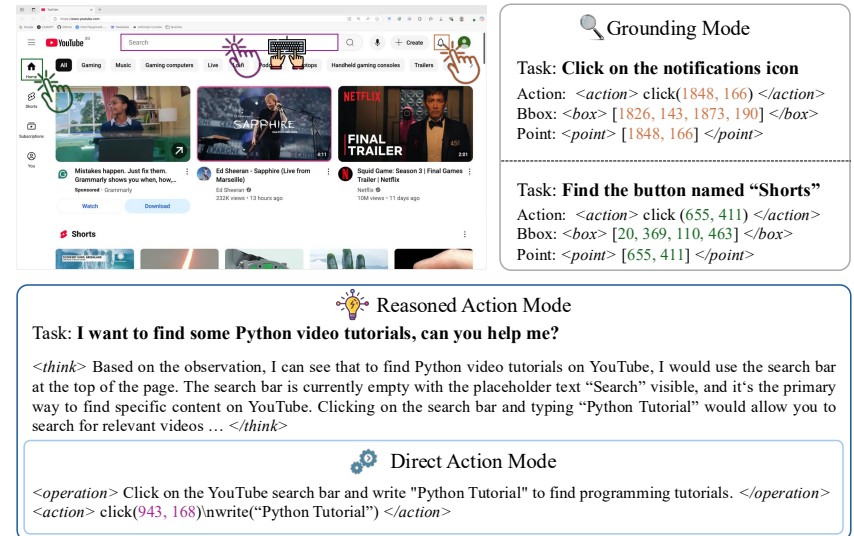

Figure 3: **Three Inference Paradigms in ScaleCUA**: (1) **Grounding Mode**, which focuses on localizing target UI elements; (2) **Direct Action Mode**, where the agent solely generates executable actions based on current observations and instructions; and (3) **Reasoned Action Mode**, where the agent first generates a chain-of-thought rationale before producing structured actions. These modes enable varying levels of functionality for computer use agents to complete tasks.

where $\pi$ denotes the agent model parameterized by $\theta$, and $\mathcal{E}$ represents the environment, such as virtual machines or Docker containers. The $task$ denotes the task instruction. The observation $o$ encompasses elements such as raw screen pixels, accessibility trees, or DOM data. The history $h_{<t} = \{(a_0, o_0), \ldots, (a_{t-1}, o_{t-1})\}$ provides context for agent's decision-making process. Similar to some works (Sun et al., 2024b; Xu et al., 2024), we choose to generate natural language descriptions for $(a_i, o_i)$ as history, as it can save a large amount of inference cost budget. Each action specifies an operation with corresponding arguments, as detailed in Table 14, which is then executed in the environment. In this work, we adopt screenshots as the observation space. This paradigm aligns with human behavior and effectively avoids interference from noisy accessibility Tree and DOM data.

## 4.2 AGENT MODELS

We build our ScaleCUA family upon Qwen2.5-VL for its strong multimodal understanding and scalability across diverse GUI platforms. As shown in Fig. 3, it supports three inference paradigms. In *Grounding Mode*, the model localizes UI elements via points, boxes, or coordinate-based actions from screenshots and instructions, making it suitable as a modular "grounder" for external planners. In *Direct Action Mode*, the model directly emits low-level instructions and executable actions, enclosed in `<operation>` and `<action>` tags. Given the current screen and interaction history, it enables fast perception–action loops without explicit reasoning. In *Reasoned Action Mode*, it first generates a rationale inside `<think>` tags before producing the action, improving reliability and interpretability on ambiguous or long-horizon tasks with extra latency. This design allows flexible integration with different agentic workflows while maintaining consistent control semantics across platforms.

**Action Space.** We design a **unified action space** for data collection and environment interaction. Table 14 summarizes our cross-platform action space spanning desktop, mobile, and web. It combines universal operations (*e.g.*, `click`, `write`, *etc.*) with platform-specific actions (*e.g.*, `long_press` and `open_app` for mobile), ensuring consistent behavior modeling and simplifying downstream policy learning. More details appear in Sec. A.4.

**Training Recipes.** We train three model scales under hardware-aware configurations: ScaleCUA-3B (mini-batch 4, grad-accum 1 on 128 A100 GPUs), ScaleCUA-7B (mini-batch 2, grad-accum 2 on 128 A100 GPUs), and ScaleCUA-32B (mini-batch 2, grad-accum 2 on 128 H200 GPUs). All use a learning rate of $1 \times 10^{-5}$ and a maximum token length of 40,960. To balance general multimodal

knowledge with GUI-specific skills, we vary the ratio of general-purpose data to GUI data: 25% for 3B, 50% for 7B, and 75% for 32B. Empirically, this scaling yields substantial gains on GUI understanding, grounding, and task completion benchmarks, confirming that larger models can absorb higher proportions of general data without diluting GUI competence.

## 5 EXPERIMENTS

**Evaluation Setup.** We comprehensively evaluate our ScaleCUA across three dimensions: GUI understanding, GUI grounding, and end-to-end task completion. All evaluations are performed under pure visual observation to align with real-world usage. For **GUI understanding**, we use MMBench-GUI L1 (Wang et al., 2025c), which tests fine-grained perception and reasoning about interface content. For **GUI grounding**, we conduct structured evaluations on ScreenSpot-v2 (Wu et al., 2024b), ScreenSpot-Pro (Li et al., 2025), and OSWorld-G (Xie et al., 2025), covering cross-platform localization and domain-specific scenarios. By default, ScreenSpot-Pro is evaluated at 2K resolution and other benchmarks at 1080p. For **end-to-end task completion**, we test our models on AndroidControl, OSWorld (Xie et al., 2024), WindowAgentArena (WAA) (Bonatti et al., 2024), macOSArena (MA) (Wang et al., 2025c), AndroidWorld (AW) (Rawles et al., 2024), and WebArena-Lite-v2 (WAL-v2). These benchmarks span desktop, mobile, and web settings, with a 50-step budget applied when not specified, enabling a realistic assessment of platform-specific performance. We further validate **general vision-language capabilities** on several well-known benchmarks (Yue et al., 2024; Lu et al., 2023; Liu et al., 2024b; xAI, 2024). In addition, we deploy Qwen2.5VL models with vLLM (Kwon et al., 2023) to ensure scalable and consistent online evaluation.

### 5.1 COMPREHENSIVE AGENT EVALUATION

**GUI Understanding.** MMBench-GUI L1 (GUI Content Understanding) assesses fine-grained perception and reasoning across six platforms following MMBench-GUI protocols. In Table 2, our ScaleCUA consistently delivers competitive or superior results. Even the lightweight ScaleCUA-3B attains 89.9%, surpassing Qwen2.5-VL-72B by +25.3 points. ScaleCUA-7B further improves to 92.3%, while ScaleCUA-32B reaches 94.4%. These results highlight the efficacy of scaling with cross-platform GUI-specific data, confirming that diverse training corpora substantially enhance visual comprehension across heterogeneous environments.

Table 2: Results on MMBench-GUI L1 (GUI Content Understanding) (Wang et al., 2025c).

| Model | Easy | Medium | Hard |
|---|---|---|---|
| GPT-4o (2024) | 60.2 | 57.2 | 53.5 |
| Claude-3.7 (2025) | 39.1 | 38.4 | 35.7 |
| Qwen2.5-VL-72B (2025) | 67.0 | 67.5 | 64.6 |
| UI-TARS-72B-DPO (2025) | 40.2 | 41.8 | 35.8 |
| InternVL3-72B (2025) | 79.2 | 77.9 | 75.7 |
| GUI-Owl-7B (2025) | 84.5 | 86.9 | 90.9 |
| GUI-Owl-32B (2025) | **92.8** | 91.7 | 94.2 |
| ScaleCUA-3B | 83.6 | 85.6 | 89.9 |
| ScaleCUA-7B | 88.4 | 90.1 | 92.3 |
| ScaleCUA-32B | 92.5 | **92.5** | **94.4** |

**GUI Grounding.** We then evaluate models on GUI grounding, which measures the ability to localize and associate visual elements with textual or functional references across desktop, mobile, and web. As shown in Fig. 4, our ScaleCUA consistently achieves state-of-the-art performance across different benchmarks. On the challenging ScreenSpot-Pro, ScaleCUA-32B again dominates, achieving 59.2% overall and delivering strong accuracy across diverse domains such as Creative software, CAD, and office applications. More detailed comparisons are presented in A.2. Overall, these results demonstrate that scaling with GUI-specific data yields substantial benefits for grounding. The consistent improvements across GUI grounding benchmarks confirm the effectiveness of our dual-loop data pipeline in learning robust UI element localization.

**Task Completion.** We evaluate end-to-end task completion on Mobile (AndroidWorld), Ubuntu (OSWorld), Windows (WindowsAgentArena), macOS (MacOSArena), and Web (WebArena-Lite-v2), considering both native agents and planner–grounder workflows. The results is shown in Table 3. First, our native ScaleCUA-32B achieves the strongest Web performance: 44.2% (15 steps budget) and 47.4% (50 steps), outperforming the best native baseline (UI-TARS-72B-DPO) by +20.8 and +26.0 points, respectively, and substantially surpassing Qwen2.5-VL-72B. Then, the workflow setting with GPT-4o as planner and ScaleCUA-7B as the grounder yields 48.3% on AndroidWorld and 28.1% on OSWorld (50 steps), outperforming other strong grounders such as JEDI-7B. Beyond these

Table 3: Online evaluation across different platforms. AndroidWorld has its own predefined step budget. ♣ denotes the unkown step budget and ★ indicates more than 50 steps is used.

| Method | Mobile (AndroidWorld) Predefined Steps | Ubuntu (OSWorld) 15 Steps | 50 Steps | Windows (WindowsAgentArena) 15 Steps | 50 Steps | MacOS (MacOSArena) 15 Steps | 50 Steps | Web (WebArena-Lite-v2) 15 Steps | 50 Steps |
|---|---|---|---|---|---|---|---|---|---|
| *Native Agent* | | | | | | | | | |
| Kimi-VL-A3B (2025b) | – | 8.2♣ | | 10.4♣ | | – | – | – | – |
| Seed1.5-VL (2025) | **62.1** | 36.7★ | | 39.6★ | | – | – | – | – |
| GLM-4.1V-Thinking (2025) | 41.7 | 14.9★ | | – | – | – | – | – | – |
| GLM-4.5V-Thinking (2025) | 57.0 | 35.8★ | | – | – | – | – | – | – |
| COMPUTERRL (2025) | – | 47.3♣ | | – | – | – | – | – | – |
| PC Agent-E (2025) | – | 14.9♣ | | – | – | – | – | – | – |
| GPT-4o (2024) | 21.6 | 6.8 | 10.1 | 5.6 | 3.5 | 0.0 | 1.4 | 2.0 | 3.3 |
| Claude-3.7 (2025) | 11.2 | 7.4 | 10.3 | 7.1 | 6.4 | 5.7 | 7.1 | 2.0 | 2.6 |
| Qwen2.5-VL-72B (2025) | 27.6 | 9.8 | 10.6 | 11.8 | 9.7 | 1.4 | 5.7 | 15.6 | 14.4 |
| InternVL3.5-241B-A28B (2025a) | 29.7 | 11.1 | 11.6 | 15.2 | 18.0 | 2.9 | 5.7 | 11.7 | 11.7 |
| Aguvis-72B (2024) | 26.1 | 3.8 | 4.2 | 4.1 | 3.5 | 0.0 | 0.0 | 5.8 | 9.0 |
| UI-TARS-7B-SFT (2025) | 33.0 | 17.7 | – | – | – | – | – | 11.0 | 13.6 |
| UI-TARS-1.5-7B (2025) | 31.6 | 22.1 | 23.9 | 11.1 | 15.9 | 7.1 | 7.1 | 20.8 | 26.0 |
| UI-TARS-72B-DPO (2025) | 46.6 | 24.2 | 25.2 | 11.1 | 17.9 | **8.6** | **8.6** | 23.4 | 21.4 |
| OpenCUA-7B (2025b) | – | 24.3 | 28.1 | – | – | – | – | – | – |
| OpenCUA-32B (2025b) | – | **29.7** | **34.1** | – | – | – | – | – | – |
| ScaleCUA-3B | 23.7 | 9.6 | 12.4 | 13.1 | 15.2 | 0.0 | 1.4 | 31.8 | 33.1 |
| ScaleCUA-7B | 27.2 | 14.3 | 15.0 | 18.0 | 20.7 | 4.3 | 4.3 | 37.7 | 37.7 |
| ScaleCUA-32B | 30.6 | 16.5 | 17.7 | **21.4** | **24.2** | 7.1 | 7.1 | **44.2** | **47.4** |
| *Agentic Workflow* | | | | | | | | | |
| **Planner** | **Grounder** | | | | | | | | |
| | Aria-UI (2024) | 44.8 | 15.2♣ | | – | | – | – | – | – |
| | OS-Atlas-7B (2024b) | – | 14.6♣ | | – | | – | – | – | – |
| | UGround-V1-7B (2024) | 32.8 | 13.1 | 16.1 | 13.1 | 20.7 | 1.4 | 0.0 | 23.2 | 26.5 |
| GPT-4o | UI-TARS-1.5-7B (2025) | 37.9 | 16.5 | 19.1 | 14.5 | 26.2 | 1.4 | 0.0 | **28.6** | 28.6 |
| | JEDI-3B (2025) | – | 22.4 | – | 29.1 | – | – | – | – | – |
| | JEDI-7B (2025) | – | 22.7 | 25.0 | 30.2 | 32.9 | – | – | – | – |
| | ScaleCUA-7B | **48.3** | **22.9** | **28.1** | **31.7** | **36.6** | **5.7** | **8.6** | **28.6** | **35.1** |

highlights, several trends emerge. (i) Scaling from 3B→7B→32B produces monotonic gains on different platforms, indicating that our cross-platform data and unified action space translate into stronger computer use agents as capacity grows. (ii) The effect of the step budget is consistent: a majority of the agents, including ScaleCUA, achieve substantial performance improvements under a 50-step limit. (iii) Even employing our proposed data, the planning ability of our model still lags substantially behind GPT-4o in agentic workflows, and models trained with existing and proprietary datasets continue to exhibit a considerable performance gap. We must acknowledge that there remains significant room for improvement and further development.

## 5.2 DIAGNOSTIC ANALYSIS ON COMPUTER USE AGENTS

To elucidate the main factors that affect agent performance, we conduct detailed ablations, which reveal key trade-offs between accuracy, efficiency, and generalization:

**Input Resolution:** OSWorld-G uses strictly standardized 1080p frames. When the input resolution is set at or above 1080p, the performance saturates because the inputs still match the maximum training resolution, *i.e.*, 1080p. In ScreenSpot-v2, the majority of screenshots are at or below 1080p, yet this results in negative impacts when the resolution is increased further. By contrast, ScreenSpot-Pro contains a large proportion of native 4K screenshots. The performance on it benefits from higher

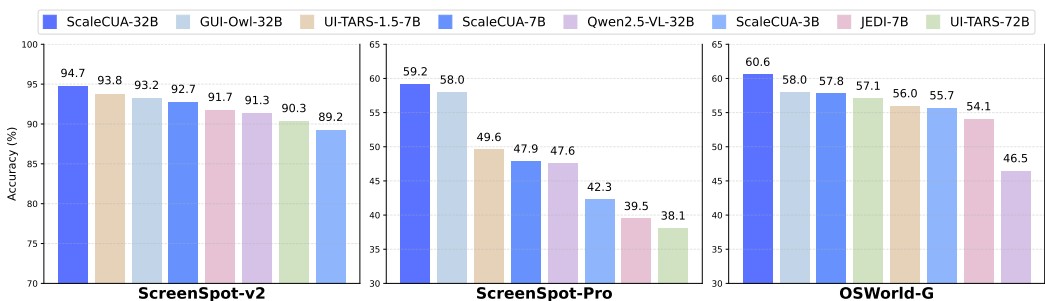

Figure 4: Results on GUI grounding datasets.

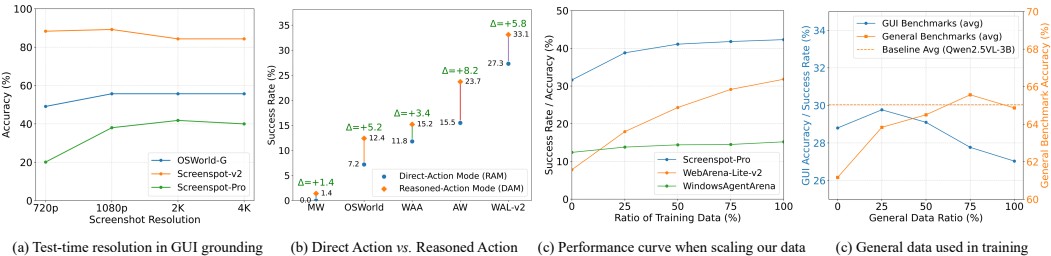

(a) Test-time resolution in GUI grounding (b) Direct Action *vs.* Reasoned Action (c) Performance curve when scaling our data (c) General data used in training

Figure 5: Evaluations across diverse conditions. (a) Accuracy of GUI grounding under different screenshot resolutions. (b) Success rates of Direct Action *vs.* Reasoned Action Modes, where reasoning consistently improves performance. (c) Training data scaling. (d) Effect of using general data, showing distinct trends between GUI and multimodal benchmarks.

resolutions up to 2K but drops at 4K. Overall, we observe that the impact of resolution on grounding performance depends largely on the benchmark's data distribution.

**Inference Modes:** Fig. 5 (b) compares the two inference modes for computer use agents. Across all benchmarks, reasoned action mode (RAM) yields higher success rates than direct action mode (DAM), with absolute gains ranging from +1.4% to + 8.2%. However, this mode also incurs longer inference time and greater token cost. DAM, in contrast, produces actions directly from the visual–textual context, yielding faster responses but being more prone to cumulative drift in long-horizon tasks. In the Table 3, when ScaleCUA-7B as a grounding model is integrated with GPT-4o under an agentic workflow, it shows higher success on task completion benchmarks than the reasoned action mode (e.g., 28.1% *vs.* 15.0% on OSWorld, 36.6% *vs.* 20.7% on WindowsAgentArena, *etc.*). The agentic workflow allows GPT-4o to handle long-context planning while leveraging grounding mode in ScaleCUA, demonstrating complementarity between ScaleCUA and general VLMs. Nevertheless, this paradigm cannot generate actions in an end-to-end manner and brings higher costs even than RAM.

**Data Scaling:** In Fig. 5 (c), success rates generally improve with more training data. Specifically, WebArena-Lite-v2 shows nearly linear gains, whereas ScreenSpot-Pro reaches strong accuracy with about half the data. For WindowsAgentArena, the observed gains appear smaller primarily because tasks in the online benchmark are more difficult with relatively low baseline scores, where even small improvements are challenging to achieve. These results intuitively reflect the task's difficulty, and also imply a larger data volume required to achieve the desired performance.

**General Multimodal Data:** Fig. 5 (d) analyzes the effect of employing general-purpose multimodal data in training. We find a clear divergence: GUI benchmarks suffer a gradual decline in performance as the ratio of general data increases, while general benchmarks improve steadily, peaking around 75%. As the multimodal corpus expands, the model's general capabilities improve, but GUI-specific knowledge may be diluted. The results indicate that a data-balanced training strategy is crucial for preserving GUI specialization without compromising general reasoning abilities. Since the larger models are able to memorize more knowledge, Since the larger VLMs can memorize more knowledge, it is reasonable to increase the ratio to 50% for the 7B model and further to 75% for the 32B model.

**Multi-platform Ablation:** Furthermore, the Fig. 6 shows that models trained exclusively on a single domain slightly outperform the cross-domain model on desktop and web benchmarks, whereas the cross-domain model performs better on the mobile benchmark. One plausible reason lies in the inherent differences in aspect ratio and UI layout across platforms. Mobile interfaces typically feature more vertically constrained layouts and standardized components with larger, touch-friendly elements, whereas desktop and web pages provide horizontally richer screens with denser and more variable UI structures. Since web/desktop data can enrich the feature space without fundamentally altering the underlying interaction patterns, the model trained on cross-platform data can thus generalize more effectively to the mobile domain with simpler visual hierarchies. Conversely, models trained on

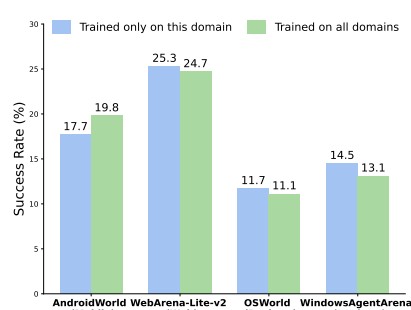

Figure 6: The effects of training on domain-specific data.

desktop or web data are exposed to information-dense layouts where UI elements may be small, overlapping, or nested within complex DOM structures. Introducing mobile data during multi-domain training can dilute the model's specialized representations for these fine-grained desktop layouts, leading to small performance drops in desktop and web benchmarks.

Generally, high-resolution inputs and reasoning-based inference enhance grounding and task completion but incur extra cost. Data scaling remains crucial but benchmark-sensitive, and heterogeneous data mixtures improve general reasoning at the expense of GUI capabilities. These insights motivate scalable, cross-platform training pipelines with deliberate data composition to build robust agents.

## 5.3 ABLATION ON DATA

Table 4: Ablation studies on data. The maximum steps used in online benchmarks are set to 50.

(a) The ablation on data augmentation. We only use GUI-related data in training.

| Model | Training Data | Aug. | SS-Pro |
|---|---|---|---|
| Qwen2.5VL-3B | ours-only | ✗ | 37.8 |
| | | ✓ | **41.3** |

(b) The ablation on weak semantic trajectories. The public datasets used are shown in Table 13.

| Model | Training Data | + WS | OSWorld | WAL-v2 |
|---|---|---|---|---|
| Qwen2.5VL-3B | public-only | ✗ | 7.6 | 8.4 |
| | | ✓ | **8.5** | **14.3** |

(c) The ablation on coordinate types.

| Model | Type | ScreenSpot-Pro |
|---|---|---|
| Qwen2.5VL-3B | Norm. | 37.9 |
| | Raw | **42.3** |

(d) The ablation on the maximum resolution during training.

| Model | Res. | SS-Pro | OSWOrld-G | OSWorld | AW |
|---|---|---|---|---|---|
| Qwen2.5VL-3B | 1080P | 42.3 | **54.3** | **12.4** | **23.3** |
| | 2K | **45.5** | 52.5 | 11.2 | 13.4 |

In this section, we aim to ablate our data. As shown in Fig. 1, training with our curated training corpus yields consistent improvements over the baseline trained on public data. In Table 4, we further highlight the effects of augmentation, weak semantic trajectories, coordinate formats, and resolution.

First, the results verify that data augmentation can improve performance by 3.5% on ScreenSpot-Pro. This confirms that augmentation enhances generalization and robustness by exposing the model to a wider range of visual conditions. Second, we investigate weak semantic trajectories derived from rule-based random exploration. Despite lacking explicit high-level goals, these trajectories provide low-cost supervision of interface navigation. Third, we study the impact of coordinate representations in grounding. Models trained with raw coordinates outperform those with normalized coordinates. This indicates that GUI grounding should follow the absolute position used in Qwen2.5VL. Finally, we ablate the training resolution. Higher resolutions yield trade-offs across benchmarks: while 2K improves grounding on ScreenSpot-Pro (45.5% *vs.* 42.3%) and preserves OSWorld-G accuracy (52.5% *vs.* 54.3%), it slightly reduces agent success rates on OSWorld and AndroidWorld. This suggests that fine-grained grounding benefits from high-resolution supervision, whereas agentic benchmarks may suffer from overfitting to pixel-level details.

## 6 CONCLUSION

In this work, we curate a large-scale multi-platform dataset with our dual-loop data pipeline that integrates automated agents and human experts into data construction. The training corpus spans understanding, element grounding, and task completion. With this dataset, we develop a new family of CUAs, *i.e.*, **ScaleCUA**, which support flexible inference paradigms for scalable integration with agent frameworks. Extensive experiments demonstrated the efficacy of our proposed method. Together, these contributions advance the frontier of computer use agents by bridging vision-language modeling with practical GUI interaction. We hope that ScaleCUA and its released resources will serve as a solid foundation for future research in building capable, trustworthy, and deployable CUAs.

**Limitations.** Although our framework study multi-platform agents with a scalable data pipeline, several challenges remain. First, integrating automatic data collection with iterative refinement into a self-improving loop is still insufficiently explored. Second, we have not employed advanced agentic techniques such as reflection or reinforcement learning, which are likely to improve long-horizon control. Third, the current history design is flat and cannot fully capture long-term dependencies. Despite not exploring these aspects in this work, we believe that releasing the full data, models, and training configurations lays a solid foundation for future progress in computer-use agents.

## ACKNOWLEDGMENTS

The work in this paper was partially supported by a grant from the Research Grants Council of the Hong Kong Special Administrative Region, China (Project Reference Number: AoE/E-601/24-N).

## ETHICS STATEMENT

Our work complies with the ICLR Code of Ethics. The proposed dataset and models are constructed without collecting any personally identifiable information or sensitive data. All screenshots, metadata, and trajectories are obtained from synthetic or publicly accessible software environments and do not involve real users' private data. When automated agents interact with platforms, they operate within controlled virtualized settings to avoid unintended data capture. Human experts are limited to interface-level information (e.g., UI element labels, bounding boxes, or action descriptions) without exposure to personal content. The released resources (dataset, models, and code) are intended solely for research purposes to advance open and reproducible study of cross-platform computer use agents. We explicitly discourage any misuse of these models in ways that could compromise privacy, security, or fairness. No conflicts of interest or sponsorship bias exist in this work, and all authors adhere to research integrity practices, including transparent documentation of data sources, collection procedures, and evaluation protocols.

## REPRODUCIBILITY STATEMENT

We have taken extensive measures to ensure reproducibility of our results. This work elaborate on the data acquisition pipeline (Sec. 3), dataset composition and statistics (Table 1 and Fig. 11), unified action space (Table 14), training recipes for different model scales (Sec. 4.2), and comprehensive evaluation protocols (Sec. 5). Additional implementation details and ablation studies are provided in the Appendix to guide replication of our experiments. We will release the dataset, model checkpoints, and source code to facilitate verification and reproducibility. Together, these resources allow researchers to reproduce our key findings and build upon our work with minimal additional assumptions.

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

click

## A  APPENDIX

This section provides supplementary materials that complement the main paper.

**A.1** – Large Language Model Usage: We make a clarification on large language model usage.

**A.2** – More Results: We report extended evaluations across multiple benchmarks (MMBench-GUI L2, OSWorld-G, AndroidControl, ScienceBoard, and general multimodal benchmarks), highlighting the scalability and cross-platform generalization of our models.

**A.3** – Public Data Used in Training: We list the public datasets incorporated into ScaleCUA training, specifying the portion of each source utilized.

**A.4** – Action Space: We describe the unified action space that abstracts platform-specific operations into a concise yet expressive set of commands, enabling consistent control across desktop, mobile, and web environments.

**A.5** – Error Case Analysis: We provide representative failure cases on desktop, Android, and web platforms to analyze limitations such as incomplete procedural understanding, insufficient state tracking, and positional reasoning errors.

**A.6** – Details of Data Curation: We detail the multi-platform GUI data collection process underlying ScaleCUA, including sources, application coverage, and platform diversity, which jointly ensure comprehensive domain knowledge and improved generalization.

**A.7** – Data Visualization: Consolidates illustrative figures for GUI Understanding, GUI Grounding, Weak-Semantic and Human-Curated Trajectories, and trajectory annotation to aid qualitative inspection.

**A.8** – Lessons from Data Acquisition: We summarizes common pitfalls and platform-specific notes (Windows, Ubuntu, macOS, Mobile, Web), distilling practical guidance for future collection runs.

**A.9** – The Details of WebArena-Lite-v2: We clarifies details of the benchmark construction and evaluation protocols (e.g., step budgets, metrics) to ensure fair comparisons .

**A.10** – Prompt Engineering: We release prompt templates for both agent inference and annotation workflows to facilitate reproducibility and adaptation.

### A.1  LARGE LANGUAGE MODEL USAGE

In this submission, we utilize LLMs (GPT-5, Gemini, *etc.*) to help us polish paper writing and summarize related works.

### A.2  MORE RESULTS

To fully demonstrate the potential of ScaleCUA, we provide additional results on serveral benchmarks.

On **MMBench-GUI L2** (Wang et al., 2025c), which incorporates stratified grounding difficulty across major operating systems, ScaleCUA-32B demonstrates performance comparable to state-of-the-art methods as shown in Table 8. It achieves leading scores in the basic difficulty setting across several platforms including Android (96.4), Web (93.9), Linux (81.2), and macOS (88.1), while maintaining competitive results in the advanced difficulty setting (e.g., Web 76.3, Android 81.7). Furthermore, ScaleCUA-7B and ScaleCUA-3B achieve average scores of 78.2 and 73.7, respectively. They demonstrate particularly robust performance in the basic difficulty setting, especially on Windows, where both models score 78.6, and on iOS, with respective scores of 96.1 and 93.0.

On **OSWorld-G** (Xie et al., 2025) for Ubuntu grounding, ScaleCUA-32B demonstrates impressive results with an overall performance of 60.6 shown in Table 9, which includes strong marks in layout understanding (70.0), element recognition (66.7), and fine-grained manipulation (51.0). All of our models underperform on the Refusal subtask because we deliberately excluded the Refusal-specific training data provided by JEDI (Xie et al., 2025). Incorporating these examples may pose a risk of biasing the model toward emitting an await/refusal state in complex grounding scenarios. Such bias diminishes the agent's propensity for active exploration within the environment, thereby degenerating its success rate in task completion.

Table 5: Results on MMBench-GUI L1 (GUI Content Understanding) (Wang et al., 2025c).

| Model | Windows | MacOS | Linux | iOS | Android | Web | Overall |
|---|---|---|---|---|---|---|---|
| Easy Level | | | | | | | |
| GPT-4o (2024) | 62.5 | 67.9 | 62.4 | 58.5 | 56.4 | 58.5 | 60.2 |
| Claude-3.5 (2024a) | 41.3 | 50.0 | 41.6 | 42.0 | 39.0 | 41.8 | 41.5 |
| Claude-3.7 (2025) | 34.7 | 49.1 | 39.4 | 42.8 | 37.5 | 40.8 | 39.1 |
| Qwen-Max-VL (2023) | 69.1 | 72.5 | 69.9 | 70.8 | 63.1 | 69.5 | 68.2 |
| Qwen2.5-VL-72B (2025) | 65.9 | 75.2 | 73.0 | 67.2 | 58.1 | 72.1 | 67.0 |
| UI-TARS-72B-DPO (2025) | 41.6 | 28.5 | 35.2 | 31.1 | 52.3 | 35.3 | 40.2 |
| InternVL3-72B (2025) | 74.7 | 78.7 | 79.2 | 83.6 | 80.1 | 81.2 | 79.2 |
| GUI-Owl-7B (2025) | 83.0 | 84.5 | 85.6 | 82.6 | 83.3 | 88.1 | 84.5 |
| GUI-Owl-32B (2025) | **93.7** | 89.3 | 93.3 | **95.7** | 90.5 | **94.1** | **92.8** |
| ScaleCUA-3B | 86.4 | 83.5 | 79.9 | 85.4 | 80.3 | 87.4 | 83.6 |
| ScaleCUA-7B | 89.5 | 86.9 | 89.1 | 86.2 | 87.0 | 90.1 | 88.4 |
| ScaleCUA-32B | 93.4 | **91.7** | **94.3** | 93.1 | 90.5 | 92.3 | 92.5 |
| Medium Level | | | | | | | |
| GPT-4o (2024) | 56.3 | 63.1 | 59.7 | 54.1 | 57.7 | 55.0 | 57.2 |
| Claude-3.5 (2024a) | 39.3 | 47.6 | 46.0 | 44.6 | 42.0 | 34.3 | 41.3 |
| Claude-3.7 (2025) | 39.3 | 39.2 | 42.3 | 39.5 | 36.1 | 36.2 | 38.4 |
| Qwen-Max-VL (2023) | 63.4 | 73.9 | 66.9 | 68.0 | 63.7 | 64.6 | 65.4 |
| Qwen2.5-VL-72B (2025) | 66.3 | 72.7 | 72.6 | 59.3 | 66.2 | 68.2 | 67.5 |
| UI-TARS-72B-DPO (2025) | 38.8 | 41.6 | 37.1 | 41.7 | 54.7 | 31.6 | 41.8 |
| InternVL3-72B (2025) | 71.5 | 78.6 | 79.9 | 78.4 | 81.4 | 78.7 | 77.9 |
| GUI-Owl-7B (2025) | 88.9 | 88.1 | 91.2 | 84.4 | 85.3 | 83.6 | 86.9 |
| GUI-Owl-32B (2025) | 94.1 | 84.5 | 95.9 | 87.8 | **92.8** | **88.6** | 91.7 |
| ScaleCUA-3B | 91.8 | 78.5 | 88.7 | 74.8 | 88.6 | 79.5 | 85.6 |
| ScaleCUA-7B | 93.6 | **91.7** | 93.4 | 84.3 | 89.6 | 85.8 | 90.1 |
| ScaleCUA-32B | **95.1** | 89.4 | **96.3** | 92.2 | 92.6 | 87.2 | **92.5** |
| Hard Level | | | | | | | |
| GPT-4o (2024) | 60.7 | 60.4 | 52.4 | 45.3 | 50.9 | 50.8 | 53.5 |
| Claude-3.5 (2024a) | 37.4 | 42.7 | 34.1 | 40.9 | 37.0 | 38.1 | 37.6 |
| Claude-3.7 (2025) | 33.0 | 34.5 | 32.0 | 39.2 | 37.0 | 38.9 | 35.7 |
| Qwen-Max-VL (2023) | 66.6 | 67.6 | 65.8 | 60.2 | 58.8 | 65.3 | 63.7 |
| Qwen2.5-VL-72B (2025) | 70.7 | 68.9 | 71.0 | 57.6 | 53.9 | 68.1 | 64.6 |
| UI-TARS-72B-DPO (2025) | 31.5 | 35.9 | 24.2 | 36.3 | 58.1 | 19.9 | 35.8 |
| InternVL3-72B (2025) | 75.1 | 77.4 | 76.2 | 70.4 | 75.7 | 78.1 | 75.7 |
| GUI-Owl-7B (2025) | 87.8 | 96.4 | 94.3 | 87.8 | 88.9 | **94.1** | 90.9 |
| GUI-Owl-32B (2025) | **93.3** | 95.2 | 95.9 | 92.2 | **95.4** | 92.7 | 94.2 |
| ScaleCUA-3B | 92.3 | 89.4 | 93.8 | 85.3 | 88.3 | 88.6 | 89.9 |
| ScaleCUA-7B | 91.9 | 91.9 | 94.9 | 89.6 | 92.9 | 91.4 | 92.3 |
| ScaleCUA-32B | 93.0 | **96.5** | **96.4** | **93.1** | 94.5 | 94.0 | **94.4** |

On **AndroidControl** (Li et al., 2024b) which is an offline planning benchmark developed for the Android, all ScaleCUA variants exhibit consistently strong performance demonstrated in Table 10. On the AndroidControl-Low, ScaleCUA-7B attains the highest task completion rate, whereas ScaleCUA-32B achieves the most reliable grounding, indicating that the compact model favors execution efficiency while the larger capacity maximizes perceptual fidelity. As for AndroidControl-High, ScaleCUA-32B demonstrates the highest success rate while showing the smallest degradation from Low to High. ScaleCUA-3B and ScaleCUA-7B achieve a favorable trade-off, sustaining solid performance across both low and high settings. The relatively small variance in type prediction across sizes suggests that residual failures arise more from long-horizon interaction and error accumulation than from intent misclassification or localization.

On **ScienceBoard** (Sun et al., 2025), a computer use benchmark designed for scientific professionals, our models show modest yet meaningful capability as shown in 11. The ScaleCUA-32B outperforms strong VLMs such as GPT-4o (1.6) while remaining below Qwen2.5-VL-72B (12.9) and Claude-3.7-Sonnet (10.5). Our model excels in domains demanding factual and visual-text reasoning over those requiring specialized symbolic workflows.

To evaluate the transfer learning capabilities of ScaleCUA-32B, we augment our training with a diverse set of multimodal data focusing on coding, math and reasoning. These data, sourced from the post-training corpus of InternVL3 (Zhu et al., 2025), encompass a range of tasks, including OCR, mathematics, coding, reasoning-QA, and general multimodal understanding. We then assess performance on four standard **General Multimodal Benchmarks** shown in Table 12. These benchmarks jointly evaluate skills such as mathematical and commonsense reasoning, text comprehension, and

Table 6: Results on ScreenSpot-v2 (Wu et al., 2024b).

| Method | Mobile | | Desktop | | Web | | Avg |
|---|---|---|---|---|---|---|---|
| | Text | Icon/Widget | Text | Icon/Widget | Text | Icon/Widget | |
| *Proprietary Models* | | | | | | | |
| Operator (2025) | 47.3 | 41.5 | 90.2 | 80.3 | 92.8 | 84.3 | 70.5 |
| Claude-3.7-Sonnet (2025) | – | – | – | – | – | – | 87.6 |
| Seed-1.5-VL (2025) | – | – | – | – | – | – | **95.2** |
| *General Open-source Models* | | | | | | | |
| Kimi-VL-A3B-Thinking-2506 (2025b) | – | – | – | – | – | – | 91.4 |
| MiMo-VL-7B-RL (2025) | – | – | – | – | – | – | 90.5 |
| InternVL3.5-241B-A28B (2025a) | 97.9 | 91.5 | 97.4 | 82.9 | 94.0 | 89.2 | 92.9 |
| Qwen2.5-VL-3B (2025) | 93.4 | 73.5 | 88.1 | 58.6 | 88.0 | 71.4 | 80.9 |
| Qwen2.5-VL-7B (2025) | 97.6 | 87.2 | 90.2 | 74.2 | 93.2 | 81.3 | 88.8 |
| Qwen2.5-VL-32B (2025) | 97.9 | 88.2 | 98.5 | 79.3 | 91.2 | 86.2 | 91.3 |
| *GUI Specialist* | | | | | | | |
| OS-Atlas-Base-7B (2024b) | 95.2 | 75.8 | 90.7 | 63.6 | 90.6 | 77.3 | 84.1 |
| UI-TARS-2B (2025) | 95.2 | 79.1 | 90.7 | 68.6 | 87.2 | 78.3 | 84.7 |
| UI-TARS-7B (2025) | 96.9 | 89.1 | 95.4 | 85.0 | 93.6 | 85.2 | 91.6 |
| UI-TARS-72B (2025) | 94.8 | 86.3 | 91.2 | 87.9 | 91.5 | 87.7 | 90.3 |
| UI-TARS-1.5 (2025) | – | – | – | – | – | – | 94.2 |
| GUI-Owl-7B (2025) | 99.0 | **92.4** | 96.9 | 85.0 | 93.6 | 85.2 | 92.8 |
| GUI-Owl-32B (2025) | 98.6 | 90.0 | 97.9 | 87.8 | 94.4 | 86.7 | 93.2 |
| *GUI Grounding Models* | | | | | | | |
| SeeClick (2024) | 78.4 | 50.7 | 70.1 | 29.3 | 55.2 | 32.5 | 55.1 |
| OmniParser-v2 (2024b) | 95.5 | 74.6 | 92.3 | 60.9 | 88.0 | 59.6 | 80.7 |
| JEDI-3B (2025) | 96.6 | 81.5 | 96.9 | 78.6 | 88.5 | 83.7 | 88.6 |
| JEDI-7B (2025) | 96.9 | 87.2 | 95.9 | 87.9 | 94.4 | 84.2 | 91.7 |
| GUI-Actor-7B (2025) | 97.6 | 88.2 | 96.9 | 85.7 | 93.2 | 86.7 | 92.1 |
| GUI-G$^2$-7B (2025) | 98.3 | 91.9 | 95.4 | 89.3 | 94.0 | 87.7 | 93.3 |
| InfiGUI-G1-3B (2025) | 99.3 | 88.2 | 94.8 | 82.9 | 94.9 | 80.3 | 91.1 |
| InfiGUI-G1-7B (2025b) | **99.0** | 91.9 | 94.3 | 82.1 | **97.9** | 89.2 | 93.5 |
| GTA1-7B (2025) | **99.0** | 88.6 | 94.9 | 89.3 | 92.3 | 86.7 | 92.4 |
| GTA1-32B (2025) | 98.6 | 89.1 | 96.4 | 86.4 | 95.7 | 88.7 | 93.2 |
| *Ours* | | | | | | | |
| ScaleCUA-3B | 94.1 | 86.3 | 94.9 | 79.3 | 89.7 | 85.7 | 89.2 |
| ScaleCUA-7B | 97.3 | 90.5 | 95.4 | 87.9 | 94.0 | 88.7 | 92.7 |
| ScaleCUA-32B | 98.6 | 91.9 | **99.0** | **90.0** | 94.4 | **91.6** | 94.7 |

open-domain visual question answering, which are also fundamental for computer-use agents. The "ScaleCUA-3B (25%)" specifies the proportion of this general-purpose data relative to the core GUI data used in training.

Based on Table 12, several consistent trends emerge regarding the interaction between the proportion of general-purpose data and agent performance on general VLM benchmarks. First, incorporating moderate amounts of general-purpose data (e.g., 25–50% relative to GUI-specific data) yields notable gains over the 0% setting, particularly on MathVista and MMMU$_{valid}$, suggesting that exposing the agent to broader multimodal reasoning tasks improves its mathematical and cross-domain inference ability. For instance, ScaleCUA-3B rises from 52.8 to 58.7 on MathVista and from 48.8 to 52.4 on MMMU when increasing general data to 50%, while maintaining stable performance on RealWorldQA. Second, the results indicate a saturation effect: pushing the general data ratio to 75% or 100% offers only marginal or inconsistent benefits. Third, scaling model capacity amplifies the positive effect of general data. However, our 7B and 32B models still exhibit a substantial performance gap compared to the baseline on general benchmarks, indicating that the proportion of general-purpose data could be further increased. Such adjustments must also consider their potential impact on the computer-use capability of agent models.

## A.3 PUBLIC DATA USED IN TRAINING

Table 13 summarizes the public datasets used for training ScaleCUA. Please note that the reported statistics refer to the portion of each dataset actually utilized in our experiments, rather than the original sizes of the source datasets.

## A.4 ACTION SPACE

To enable robust cross-platform control, we define a unified action space that abstracts low-level GUI actions into a concise yet expressive set of semantic commands. As shown in Table 14, this action space is designed to be platform-aware yet semantically consistent, allowing our agents to

Table 7: Results on ScreenSpot-Pro (Li et al., 2025).

| Agent Model | Development | | | Creative | | | CAD | | | Scientific | | | Office | | | OS | | | Avg |
|---|---|---|---|---|---|---|---|---|---|---|---|---|---|---|---|---|---|---|---|
| | Text | Icon | Avg | Text | Icon | Avg | Text | Icon | Avg | Text | Icon | Avg | Text | Icon | Avg | Text | Icon | Avg | |
| *Proprietary Models* | | | | | | | | | | | | | | | | | | | |
| Claude (2024b) | 22.0 | 3.9 | 12.6 | 25.9 | 3.4 | 16.8 | 14.5 | 3.7 | 11.9 | 33.9 | 15.8 | 25.8 | 30.1 | 16.3 | 26.9 | 11.0 | 4.5 | 8.1 | 17.1 |
| Operator (2025) | 50.0 | 19.3 | 35.1 | 51.5 | 23.1 | 39.6 | 16.8 | 14.1 | 16.1 | 58.3 | 24.5 | 43.7 | 60.5 | 28.3 | 53.0 | 34.6 | 30.3 | 32.7 | 36.6 |
| *General Open-source Models* | | | | | | | | | | | | | | | | | | | |
| Qwen2-VL-7B (2024) | 2.6 | 0.0 | 1.3 | 1.5 | 0.0 | 0.9 | 0.5 | 0.0 | 0.4 | 6.3 | 0.0 | 3.5 | 3.4 | 1.9 | 3.0 | 0.9 | 0.0 | 0.5 | 1.6 |
| CogAgent-18B (2024) | 14.9 | 0.7 | 8.0 | 9.6 | 0.0 | 5.6 | 7.1 | 3.1 | 6.1 | 22.2 | 1.8 | 13.4 | 13.0 | 0.0 | 10.0 | 5.6 | 0.0 | 3.1 | 7.7 |
| Qwen2.5-VL-3B (2025) | 38.3 | 3.4 | 21.4 | 40.9 | 4.9 | 25.8 | 22.3 | 6.3 | 18.4 | 44.4 | 10.0 | 29.5 | 48.0 | 17.0 | 40.9 | 33.6 | 4.5 | 20.4 | 25.9 |
| Qwen2.5-VL-7B (2025) | 51.9 | 4.8 | 29.1 | 36.9 | 8.4 | 24.9 | 17.8 | 1.6 | 13.8 | 48.6 | 8.2 | 31.1 | 53.7 | 18.9 | 45.7 | 34.6 | 7.9 | 22.4 | 27.6 |
| Qwen2.5-VL-32B (2025) | 74.0 | 21.4 | 48.5 | 61.1 | 13.3 | 41.1 | 38.1 | 15.6 | 32.6 | 78.5 | 29.1 | 57.1 | 76.3 | 37.7 | 67.4 | 55.1 | 27.0 | 42.3 | 47.6 |
| *GUI Specialist* | | | | | | | | | | | | | | | | | | | |
| ShowUI-2B (2024) | 16.9 | 1.4 | 9.4 | 9.1 | 0.0 | 5.3 | 2.5 | 0.0 | 1.9 | 13.2 | 7.3 | 10.6 | 15.3 | 7.5 | 13.5 | 10.3 | 2.2 | 6.6 | 7.7 |
| OS-Atlas-7B (2024b) | 33.1 | 1.4 | 17.7 | 28.8 | 2.8 | 17.9 | 12.2 | 4.7 | 10.3 | 37.5 | 7.3 | 24.4 | 33.9 | 5.7 | 27.4 | 27.1 | 4.5 | 16.8 | 18.9 |
| UI-TARS-2B (2025) | 47.4 | 4.1 | 26.4 | 42.9 | 6.3 | 27.6 | 17.8 | 4.7 | 14.6 | 56.9 | 17.3 | 39.8 | 50.3 | 17.0 | 42.6 | 21.5 | 5.6 | 14.3 | 27.7 |
| UI-TARS-7B (2025) | 58.4 | 12.4 | 36.1 | 50.0 | 9.1 | 32.8 | 20.8 | 9.4 | 18.0 | 63.9 | 31.8 | 50.0 | 63.3 | 20.8 | 53.5 | 30.8 | 16.9 | 24.5 | 35.7 |
| UI-TARS-72B (2025) | 63.0 | 17.3 | 40.8 | 57.1 | 15.4 | 39.6 | 18.8 | 12.5 | 17.2 | 64.6 | 20.9 | 45.7 | 63.3 | 26.4 | 54.8 | 42.1 | 15.7 | 30.1 | 38.1 |
| UI-TARS-1.5-7B (2025) | – | – | – | – | – | – | – | – | – | – | – | – | – | – | – | – | – | – | 49.6 |
| UI-TARS-1.5 (2025) | – | – | – | – | – | – | – | – | – | – | – | – | – | – | – | – | – | – | 61.6 |
| GUI-Owl-7B (2025) | 76.6 | 31.0 | 54.5 | 59.6 | 27.3 | 46.1 | 64.5 | 21.9 | 54.1 | 79.1 | 37.3 | 61.0 | 77.4 | 39.6 | 68.7 | 59.8 | 33.7 | 47.9 | 54.9 |
| GUI-Owl-32B (2025) | **84.4** | **39.3** | **62.5** | 65.2 | 18.2 | 45.5 | 62.4 | 28.1 | 54.0 | **82.6** | 39.1 | **63.8** | **81.4** | 39.6 | 71.8 | **70.1** | 36.0 | **54.6** | **58.0** |
| *GUI Grounding Models* | | | | | | | | | | | | | | | | | | | |
| SeeClick (2024) | 0.6 | 0.0 | 0.3 | 1.0 | 0.0 | 0.6 | 2.5 | 0.0 | 1.9 | 3.5 | 0.0 | 2.0 | 1.1 | 0.0 | 0.9 | 2.8 | 0.0 | 1.5 | 1.1 |
| Aria-UI (2024) | 16.2 | 0.0 | 8.4 | 23.7 | 2.1 | 14.7 | 7.6 | 1.6 | 6.1 | 27.1 | 6.4 | 18.1 | 20.3 | 1.9 | 16.1 | 4.7 | 0.0 | 2.6 | 11.3 |
| UGround-V1-7B (2024) | – | – | 35.5 | – | – | 27.8 | – | – | 13.5 | – | – | 38.8 | – | – | 48.8 | – | – | 26.1 | 31.1 |
| UGround-V1-72B (2024) | – | – | 31.1 | – | – | 35.8 | – | – | 13.8 | – | – | 50.0 | – | – | 51.3 | – | – | 25.5 | 34.5 |
| JEDI-3B (2025) | 61.0 | 13.8 | 38.1 | 53.5 | 8.4 | 34.6 | 27.4 | 9.4 | 23.0 | 54.2 | 18.2 | 38.6 | 64.4 | 32.1 | 57.0 | 38.3 | 9.0 | 25.0 | 36.1 |
| JEDI-7B (2025) | 42.9 | 11.0 | 27.4 | 50.0 | 11.9 | 34.0 | 38.0 | 14.1 | 32.2 | 72.9 | 25.5 | 52.4 | 75.1 | 47.2 | 68.7 | 33.6 | 16.9 | 26.0 | 39.5 |
| UI-R1-3B (2025) | 22.7 | 4.1 | – | 27.3 | 3.5 | – | 11.2 | 6.3 | – | 42.4 | 11.8 | – | 32.2 | 11.3 | – | 13.1 | 4.5 | – | 17.8 |
| InfiGUI-R1-3B (2025b) | 51.3 | 12.4 | – | 44.9 | 7.0 | – | 33.0 | 14.1 | – | 58.3 | 20.0 | – | 65.5 | 28.3 | – | 43.9 | 12.4 | – | 35.7 |
| InfiGUI-R1-7B (2025b) | 57.4 | 23.4 | – | 74.7 | 24.1 | – | **64.6** | 15.4 | – | 80.6 | 31.8 | – | 75.7 | 39.6 | – | 57.0 | 29.2 | – | 51.9 |
| GUI-G1-3B (2025) | 50.7 | 10.3 | 31.1 | 36.6 | 11.9 | 26.6 | 39.6 | 9.4 | 32.2 | 61.8 | 30.0 | 48.0 | 67.2 | 32.1 | 59.1 | 23.5 | 10.6 | 16.1 | 37.1 |
| GUI-G²-7B (2025) | 55.8 | 12.5 | – | 68.8 | 17.2 | – | 57.1 | 15.4 | – | 77.1 | 24.5 | – | 74.0 | 32.7 | – | 57.9 | 21.3 | – | 47.5 |
| *Ours* | | | | | | | | | | | | | | | | | | | |
| ScaleCUA-3B | 57.8 | 18.6 | 38.8 | 42.9 | 16.8 | 32.0 | 54.3 | 28.1 | 47.9 | 64.6 | 35.5 | 52.0 | 66.7 | 37.7 | 53.9 | 31.8 | 16.9 | 25.0 | 42.3 |
| ScaleCUA-7B | 66.2 | 20.7 | 44.1 | 56.6 | 20.3 | 41.3 | 54.8 | 21.9 | 46.7 | 77.1 | 24.5 | 54.3 | 74.0 | 45.3 | 67.4 | 49.5 | 18.0 | 35.2 | 47.9 |
| ScaleCUA-32B | 75.3 | 35.2 | 55.8 | 73.2 | 30.8 | 55.4 | 60.4 | **39.1** | **55.2** | 76.4 | **46.4** | 63.4 | 81.4 | **49.1** | **73.9** | 63.6 | **41.6** | 53.6 | **59.2** |

operate seamlessly across Desktop (Windows, macOS, Ubuntu), Mobile (Android, iOS), and Web platforms. The action set includes universally supported operations such as `click`, `write`, `wait`, and `terminate`, which are shared across all platforms. It also accommodates platform-specific interactions, including `swipe` and `long_press` for mobile devices, and fine-grained mouse or keyboard controls such as `doubleClick`, `rightClick`, `dragTo`, and `hotkey` for desktop and web interfaces. To handle modern interactive elements, `swipe` operation has also been implemented for Web. By standardizing the operation interface through a shared action space, we simplify training and inference while supporting both generalization and specialization. Each action is defined with explicit arguments (e.g., coordinates, keypresses), enabling precise control and compatibility with structured outputs in grounding, direct-action, and reasoned-action inference modes. This design facilitates modular training, policy transfer, and scalable data annotation, forming a critical foundation for developing universal GUI agents.

Table 8: Performance on the MMBench-GUI L2 (GUI Element Grounding) (Wang et al., 2025c).

| Model | Windows | | MacOS | | Linux | | iOS | | Android | | Web | | Avg |
|---|---|---|---|---|---|---|---|---|---|---|---|---|---|
| | Basic | Adv. | Basic | Adv. | Basic | Adv. | Basic | Adv. | Basic | Adv. | Basic | Adv. | |
| GPT-4o (2024) | 1.5 | 1.1 | 8.7 | 4.3 | 1.1 | 1.0 | 5.1 | 3.3 | 2.5 | 1.4 | 3.2 | 2.9 | 2.9 |
| Claude-3.7 (2025) | 1.5 | 0.7 | 12.5 | 7.5 | 1.1 | 0.0 | 13.7 | 10.6 | 1.4 | 1.4 | 3.2 | 2.3 | 4.7 |
| Qwen-Max-VL (2023) | 43.9 | 36.8 | 58.8 | 56.1 | 53.9 | 30.1 | 77.4 | 59.1 | 79.5 | 70.1 | 74.8 | 58.8 | 58.0 |
| Aguvis-7B-720P (2024) | 37.3 | 21.7 | 48.1 | 33.3 | 33.5 | 25.0 | 67.5 | 65.2 | 61.0 | 51.0 | 61.6 | 45.5 | 45.7 |
| ShowUI-2B (2024) | 9.2 | 4.4 | 24.1 | 10.4 | 15.1 | 11.7 | 29.0 | 19.7 | 17.4 | 8.7 | 22.9 | 12.7 | 16.0 |
| OS-Atlas-Base-7B (2024b) | 36.9 | 18.8 | 44.4 | 21.7 | 31.4 | 13.3 | 74.8 | 48.8 | 69.6 | 46.8 | 61.3 | 35.4 | 41.4 |
| UGround-V1-7B (2024) | 66.8 | 39.0 | 71.3 | 48.6 | 56.5 | 31.1 | 92.7 | 70.9 | 93.5 | 71.0 | 88.7 | 64.6 | 65.7 |
| InternVL3-72B (2025) | 70.1 | 42.6 | 75.7 | 52.3 | 59.2 | 41.3 | 93.6 | 80.6 | 92.7 | 78.6 | 90.7 | 65.9 | 72.2 |
| Qwen2.5-VL-72B (2025) | 55.7 | 33.8 | 49.9 | 30.1 | 40.3 | 20.9 | 56.1 | 28.2 | 55.6 | 25.4 | 68.4 | 45.8 | 41.8 |
| Qwen2.5-VL-7B (2025) | 31.4 | 16.5 | 31.3 | 22.0 | 21.5 | 12.2 | 66.6 | 55.2 | 35.1 | 35.2 | 40.3 | 32.5 | 33.9 |
| UI-TARS-1.5-7B (2025) | 68.3 | 39.0 | 69.0 | 44.5 | 64.4 | 37.8 | 88.5 | 69.4 | 90.5 | 69.3 | 81.0 | 56.5 | 64.3 |
| UI-TARS-72B-DPO (2025) | 78.6 | 51.8 | 80.3 | 62.7 | 68.6 | 51.5 | 90.8 | 81.2 | 93.0 | 80.0 | 88.1 | 68.5 | 74.3 |
| GUI-Owl-7B (2025) | **86.3** | 61.8 | 81.7 | 64.5 | 74.4 | 61.7 | 94.9 | 83.0 | 95.8 | 83.7 | 93.2 | 72.7 | 80.5 |
| GUI-Owl-32B (2025) | 85.6 | **65.1** | 84.9 | **67.1** | 77.0 | 63.3 | 95.2 | 85.5 | 96.1 | **87.0** | 95.5 | 80.8 | **83.0** |
| InfiGUI-G1-3B (2025b) | 74.2 | 47.1 | 78.8 | 55.2 | 65.4 | 41.8 | 95.2 | 78.8 | 92.1 | 78.0 | 89.7 | 64.3 | 73.4 |
| InfiGUI-G1-7B (2025b) | 82.7 | 61.8 | 83.8 | 63.9 | 72.3 | 52.0 | 94.9 | **89.4** | 95.2 | 85.6 | 93.5 | 76.3 | 80.8 |
| ScaleCUA-3B | 78.6 | 46.0 | 79.4 | 52.9 | 73.3 | 49.0 | 93.0 | 73.3 | 94.1 | 74.4 | 92.6 | 63.6 | 73.7 |
| ScaleCUA-7B | 78.6 | 54.0 | 82.3 | 58.7 | 74.4 | 56.6 | 94.3 | 81.8 | **96.1** | 81.1 | 92.6 | 73.1 | 78.2 |
| ScaleCUA-32B | 83.0 | 62.9 | **88.1** | 64.2 | **81.2** | **65.8** | **95.9** | 84.9 | **96.4** | 81.7 | 93.9 | 76.3 | 82.0 |

Table 9: Performance comparison on OSWorld-G (Xie et al., 2025).

| Agent Model | Text Matching | Element Recognition | Layout Understanding | Fine-grained Manipulation | Refusal | Overall |
|---|---|---|---|---|---|---|
| Gemini-2.5-Pro (2025) | 59.8 | 45.5 | 49.0 | 33.6 | **38.9** | 45.2 |
| Operator (2025) | 51.3 | 42.4 | 46.6 | 31.5 | 0.0 | 40.6 |
| Seed1.5-VL (2025) | **73.9** | **66.7** | 69.6 | 47.0 | 18.5 | **62.9** |
| OS-Atlas-7B (2024b) | 44.1 | 29.4 | 35.2 | 16.8 | 7.4 | 27.7 |
| UGround-V1-7B (2024) | 51.3 | 40.3 | 43.5 | 24.8 | 0.0 | 36.4 |
| Aguvis-7B (2024) | 55.9 | 41.2 | 43.9 | 28.2 | 0.0 | 38.7 |
| UI-TARS-7B (2025) | 60.2 | 51.8 | 54.9 | 35.6 | 0.0 | 47.5 |
| UI-TARS-1.5-7B (2025) | 70.1 | 57.9 | 59.7 | 51.7 | 0.0 | 56.0 |
| UI-TARS-72B (2025) | 69.4 | 60.6 | 62.9 | 45.6 | 0.0 | 57.1 |
| Qwen2.5-VL-3B (2025) | 41.4 | 28.8 | 34.8 | 13.4 | 0.0 | 27.3 |
| Qwen2.5-VL-7B (2025) | 45.6 | 32.7 | 41.9 | 18.1 | 0.0 | 31.4 |
| Qwen2.5-VL-32B (2025) | 63.2 | 47.3 | 49.0 | 36.9 | 0.0 | 46.5 |
| InternVL3.5-241B-A28B (2025a) | 64.4 | 58.8 | 55.3 | 43.0 | 0.0 | 53.2 |
| JEDI-3B (2025) | 67.4 | 53.0 | 53.8 | 44.3 | 7.4 | 50.9 |
| JEDI-7B (2025) | 65.9 | 55.5 | 57.7 | 46.9 | 7.4 | 54.1 |
| ScaleCUA-3B | 64.8 | 61.8 | 64.0 | 43.6 | 0.0 | 55.7 |
| ScaleCUA-7B | 67.8 | 61.8 | 64.8 | 49.7 | 0.0 | 57.8 |
| ScaleCUA-32B | 69.0 | **66.7** | **70.0** | **51.0** | 0.0 | 60.6 |

Table 10: Performance comparison on AndroidControl (Li et al., 2024b).

| Agent Model | AndroidControl-Low | | | AndroidControl-High | | |
|---|---|---|---|---|---|---|
| | Type | Grounding | SR | Type | Grounding | SR |
| Claude (2024b) | 74.3 | 0.0 | 19.4 | 63.7 | 0.0 | 12.5 |
| GPT-4o (2024) | 74.3 | 0.0 | 19.4 | 66.3 | 0.0 | 20.8 |
| SeeClick (2024) | 93.0 | 73.4 | 75.0 | 82.9 | 62.9 | 59.1 |
| InternVL-2-4B (2024b) | 90.9 | 84.1 | 80.1 | 84.1 | 72.7 | 66.7 |
| Qwen2-VL-7B (2024) | 91.9 | 86.5 | 82.6 | 83.8 | 77.7 | 69.7 |
| Aria-UI (2024) | – | 87.7 | 67.3 | – | 43.2 | 10.2 |
| OS-Atlas-4B (2024b) | 91.9 | 83.8 | 80.6 | 84.7 | 73.8 | 67.5 |
| OS-Atlas-7B (2024b) | 93.6 | 88.0 | 85.2 | 85.2 | 78.5 | 71.2 |
| Aguvis-7B (2024) | – | – | 80.5 | – | – | 61.5 |
| Aguvis-72B (2024) | – | – | 84.4 | – | – | 66.4 |
| OS-Genesis-7B (2024b) | 91.3 | – | 74.2 | 66.2 | – | 44.5 |
| UI-TARS-2B (2025) | **98.1** | 87.3 | 89.3 | 81.2 | 78.4 | 68.9 |
| UI-TARS-7B (2025) | 98.0 | 89.3 | 90.8 | 83.7 | 80.5 | 72.5 |
| UI-TARS-72B (2025) | **98.1** | 89.9 | 91.3 | 85.2 | 81.5 | 74.7 |
| Qwen2.5-VL-3B (2025) | – | – | 90.8 | – | – | 63.7 |
| Qwen2.5-VL-7B (2025) | – | – | 91.4 | – | – | 60.1 |
| Qwen2.5-VL-32B (2025) | – | – | 93.3 | – | – | 69.6 |
| Qwen2.5-VL-72B (2025) | – | – | **93.7** | – | – | 67.4 |
| InternVL3.5-241B-A28B (2025a) | 88.1 | 93.4 | 82.1 | 81.0 | 81.5 | 68.2 |
| ScaleCUA-3B | 91.4 | 93.7 | 84.1 | 81.4 | 83.9 | 70.3 |
| ScaleCUA-7B | 93.3 | 93.1 | 86.0 | 86.3 | **84.3** | 74.8 |
| ScaleCUA-32B | 91.9 | **94.7** | 85.7 | 85.7 | **87.3** | **75.9** |

## A.5 ERROR CASE ANALYSIS

We here provide several error cases across different platforms to analyze the limitations of our ScaleCUA.

On desktop platforms, ScaleCUA frequently violates *procedural prerequisites* shown in Fig 7 and Fig 8, such as attempting to compress files without selecting them or changing font styles without highlighting the target text. These issues stem from an incomplete understanding of interface states and sub-task dependencies. Moreover, a significant limitation of ScaleCUA emerges when actions result in silent failures, characterized by a lack of discernible state transition. In such instances, the model tends to perseverate on the unsuccessful operation, revealing the absence of a robust error-recovery mechanism. This issue underscores the critical requirement for fine-grained perception

Table 11: Performance comparison on ScienceBoard (Sun et al., 2025).

| Model | Algebra | Biochem | GIS | ATP | Astron | Doc | Overall |
|---|---|---|---|---|---|---|---|
| GPT-4o (2024) | 3.2 | 0.0 | 0.0 | 0.0 | 0.0 | 6.3 | 1.6 |
| Claude-3.7-Sonnet (2025) | 9.7 | **37.9** | 2.9 | 0.0 | 6.1 | 6.3 | 10.5 |
| Gemini-2.0-Flash (2024) | 6.5 | 3.5 | 2.9 | 0.0 | 0.0 | 6.1 | 3.2 |
| Qwen2.5-VL-72B (2025) | **22.6** | 27.6 | **5.9** | 0.0 | 9.1 | **12.5** | **12.9** |
| InternVL3-78B (2025) | 6.5 | 3.5 | 0.0 | 0.0 | 0.0 | 6.3 | 2.7 |
| UI-TARS-1.5-7B (2025) | 12.9 | 13.8 | 0.0 | 0.0 | 6.1 | 0.0 | 5.9 |
| ScaleCUA-3B | 6.5 | 13.8 | 0.0 | 0.0 | 0.0 | 0.0 | 3.6 |
| ScaleCUA-7B | 3.2 | 3.4 | 0.0 | 0.0 | 1.8 | 0.0 | 1.8 |
| ScaleCUA-32B | 9.7 | 10.3 | 0.0 | 0.0 | **12.1** | 0.0 | 5.9 |

Table 12: Performance on General VLM Benchmarks. ScaleCUA-3B (25%) denotes that, during training, the number of general-purpose data samples was set to 25% of the GUI data samples (*e.g.*, Understanding, Grounding, and Planning).

| Model | MathVista$_{\text{MINI}}$ (2023) | OCRBench (2024b) | MMMU$_{\text{valid}}$ (2024) | RealWorldQA (2024) |
|---|---|---|---|---|
| Qwen2.5-VL-3B (2025) | **62.3** | 797 (79.7) | 53.1 | **65.4** |
| ScaleCUA-3B (0%) | 52.8 | 819 (81.9) | 48.8 | 65.2 |
| ScaleCUA-3B (25%) | 58.6 | 823 (82.3) | 50.6 | **65.4** |
| ScaleCUA-3B (50%) | 58.7 | **824 (82.4)** | 52.4 | 65.1 |
| ScaleCUA-3B (75%) | 59.3 | 818 (81.8) | **55.6** | 65.2 |
| ScaleCUA-3B (100%) | 60.6 | 806 (80.6) | 53.4 | 63.5 |
| Qwen2.5VL-7B (2025) | **68.2** | **864 (86.4)** | **58.6** | 68.5 |
| ScaleCUA-7B (50%) | 65.4 | 852 (85.2) | 54.7 | **69.8** |
| Qwen2.5-VL-32B (2025) | **74.7** | **854 (85.4)** | **70.0** | 72.2 |
| ScaleCUA-32B (75%) | 69.8 | 827 (82.7) | 61.9 | **72.3** |

and a robust understanding of element state to interact with context-dependent UI elements, such as focus and selection.

For the Android platform, there exist precision and positional challenges demonstrated in Fig 9. In the first case, the instruction explicitly requires appending text to the top of a file within a note-taking application (Markor). However, the agent fails to recognize this positional constraint, instead inserting

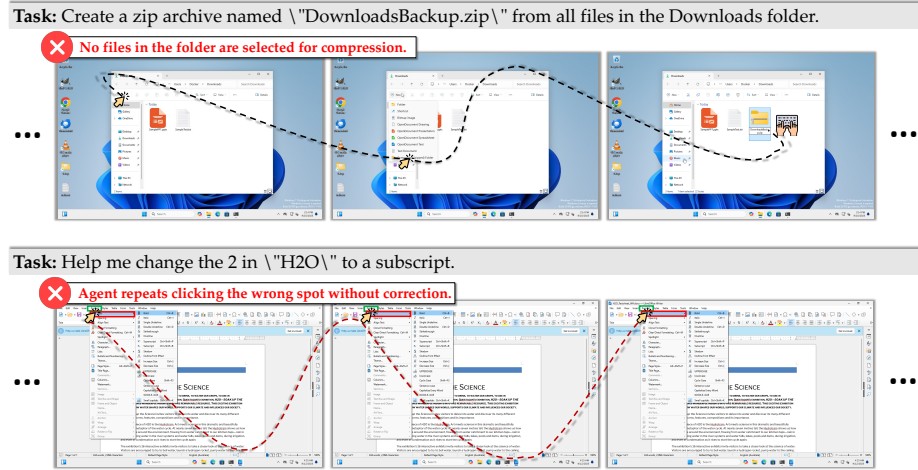

Figure 7: Error cases on the Windows platform. The first case shows ScaleCUA creating an archive without having selected any files, revealing that it sometimes fails to follow the full instruction and only completes a sub-step. The second case shows ScaleCUA persistently repeating the same action until the step limit, when it misses the correct element and the screen remains unchanged.

Table 13: Public data sources used for training our models. The "*" indicates that we count the number of samples we use rather than the full size of the original dataset because we have processed these datasets and filtered some noisy examples. Some statistics are calculated from Aguvis (2024).

| GUI Task | Data source | Platform | #Elements / #Steps |
|---|---|---|---|
| Grounding | SeeClick (2024) | Website | 271K |
| | GUIEnv (2024a) | Website | 328K |
| | GUIAct (2024a) | Website | 67K |
| | WebUI (2023a) | Website | 57K |
| | Widget Captioning (2020b) | Mobile | 101K |
| | RicoSCA (2020a) | Mobile | 173K |
| | UI RefExp (2021) | Mobile | 16K |
| | RICO Icon (2017) | Mobile | 16K |
| | OmniACT (2024) | Desktop & Website | 7K |
| | UGround (2024)* | Website | 1404K |
| | OS-Atlas (2024b)* | Desktop | 799K |
| | JEDI (2025)* | Desktop | 550K |
| | **Total** | – | **3789K** |
| Planning | MM-Mind2Web (2023) | Website | 7.8K |
| | GUIAct (2024a) | Website | 16.6K |
| | MiniWoB++ (2017) | Website | 9.9K |
| | AitZ (2024) | Mobile | 11.9K |
| | AndroidControl (2024a) | Mobile | 74.8K |
| | GUI Odyssey (2024a) | Mobile | 118.3K |
| | AMEX (2024) | Mobile | 35.6K |
| | AitW (2023) | Mobile | 19.0K |
| | PC Agent-E (2025) | Desktop | 27.8K |
| | **Total** | – | **321.7K** |

Table 14: Actions space.

| Action | Platforms | Description |
|---|---|---|
| `click(x, y, clicks, button)` | All | Perform a mouse click at coordinates (x, y) using the specified button and number of clicks. |
| `write(message)` | All | Input the given message. |
| `wait(seconds)` | All | Pause execution for the specified number of seconds. |
| `response(answer)` | All | Submit a response to the environment or task prompt. |
| `terminate(status)` | All | Terminate the current task with a given completion status. |
| `scroll(clicks, x, y)` | Desktop | Performs a scroll of the mouse scroll wheel at position (x, y). |
| `doubleClick(x, y, button)` | Desktop & Web | Perform a double click at coordinates (x, y) with the specified button. |
| `rightClick(x, y, button)` | Desktop & Web | Perform a right click at coordinates (x, y) with the specified button. |
| `hotkey(*args)` | Desktop & Web | Trigger a keyboard shortcut composed of one or more keys. |
| `moveTo(x, y)` | Desktop & Web | Move the mouse pointer to the specified (x, y) position. |
| `dragTo(x, y, button)` | Desktop & Web | Drag the mouse to (x, y) while holding the specified button. |
| `press(keys, presses)` | Desktop & Web | Press the specified key(s) a given number of times. |
| `keyDown(key)` | Desktop & Web | Press and hold a key without releasing it. |
| `keyUp(key)` | Desktop & Web | Release a previously held key. |
| `swipe(from, to, direction, amount)` | Mobile & Web | Swipe from a start to end point in the specified direction with a given intensity or distance. |
| `navigate_home()` | Mobile | Return to the mobile home screen. |
| `navigate_back()` | Mobile | Navigate back to the previous screen on mobile. |
| `long_press()` | Mobile | Perform a long-press gesture on the current focus or location. |
| `open_app(app_name)` | Mobile | Launch a mobile application by its name. |

content at the current cursor location without adjusting it. This suggests that ScaleCUA lacks a fine-grained understanding of positional semantics in natural language instructions, as well as the ability to reason about UI state changes like cursor positioning. In the second case, the agent is instructed to "Take one photo." Despite correctly launching the camera and triggering the shutter once, the agent erroneously repeats the same actions multiple times. This behavior stems from a failure to detect visual feedback or confirm state transitions (e.g., a captured photo thumbnail),

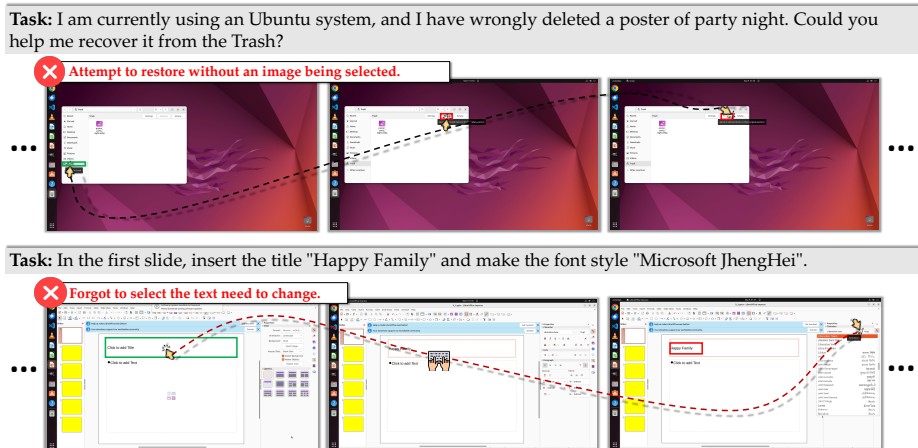

Figure 8: Error cases on the Ubuntu platform. ScaleCUA repeatedly fails tasks because it does not comprehend procedural prerequisites. The agent attempts to execute a final command without first performing the necessary intermediate step of selecting the target object. For instance, it tries to restore a file without selecting it from the trash or alter a font without highlighting the text. Critically, this operational flaw generates no explicit error message, trapping the agent in a repetitive loop of ineffective actions.

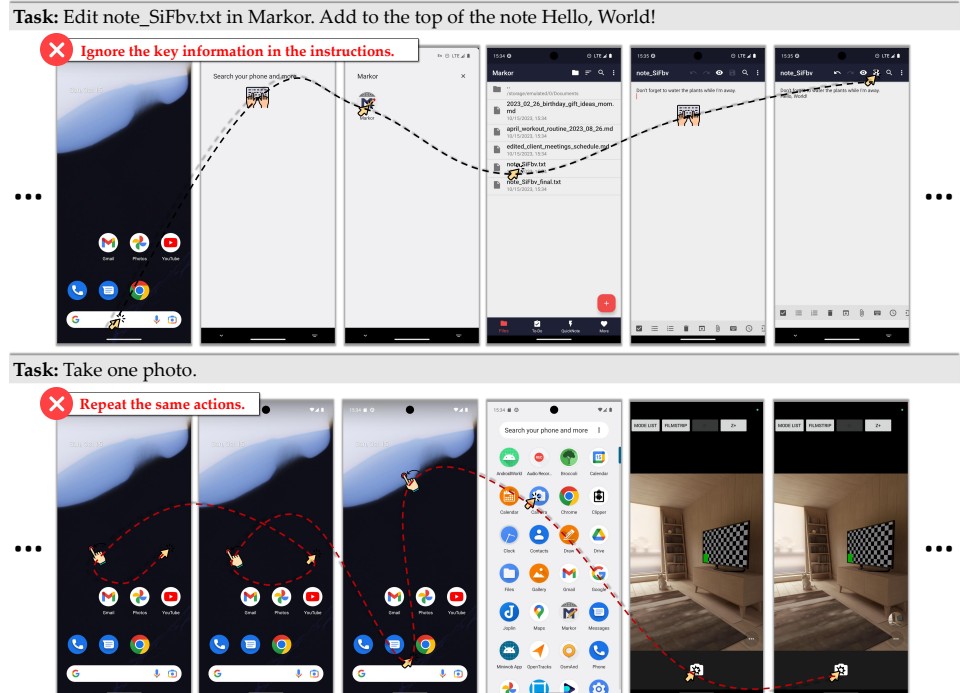

Figure 9: Error cases on the Android platform. The first case shows an instruction requiring content to be inserted at the top of a document; however, ScaleCUA opens the file and inserts directly at the current cursor location, ignoring the positional prerequisite. The second case shows that when the UI exhibits no obvious state change after an operation, ScaleCUA repeats the same action multiple times, causing tasks such as taking a photo to fail.

leading to unnecessary repetition. These failure modes indicate two key limitations: (1) insufficient grounding of spatial and contextual cues embedded in task descriptions, and (2) inadequate visual state tracking, particularly under conditions where UI feedback is subtle. Addressing these issues may require enhanced visual reasoning modules, memory-based state modeling, or task-guided grounding refinements.

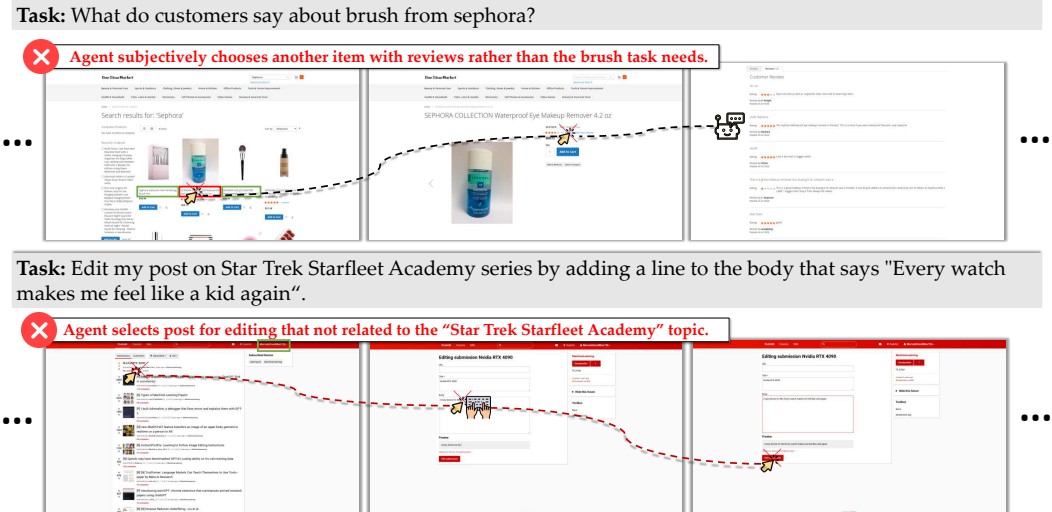

Figure 10: Error cases on the Web platform. The first case shows ScaleCUA made subjective analytical assumptions, presuming the product necessarily contained reviews, while disregarding the explicitly specified product category in the task instructions. The second case shows ScaleCUA struggles with complex tasks in complex initial environments (where numerous posts already exist on the starting interface). When faced with multifaceted requirements (needing to identify both "my" posts and posts on a specified topic), it neglected the explicitly stated topic in the instructions, instead selecting only posts visible in the current observation space that belonged to me.

Empirical analysis of trajectories from web platform reveals that ScaleCUA may struggle with semantic disambiguation. ScaleCUA often selects visually salient but instruction-inconsistent elements (e.g., wrong product category or unrelated post) as presented in Fig 10, revealing a bias toward superficial cues over explicit constraints like ownership ("my post") or topical relevance ("Starfleet Academy").

To mitigate these issues, three avenues may show promise: (1) **Reflection and State Verification.** Integrating lightweight screen-change detectors and visual precondition checkers can allow agents to validate action effects and avoid ineffective loops. (2) **Reinforcement Learning with Recovery Signals.** Reward structures should penalize redundant, non-progressive behaviors and incentivize predicate satisfaction (e.g., "text selected", "correct tab active") before proceeding. (3) **Memory-Augmented Planning.** By introducing episodic memory to recall past interactions (e.g., whether a menu opened successfully), the agent can reason across time and avoid retrying failed subgoals.

## A.6 THE DETAILS OF DATA CURATION

### A.6.1 DATA SOURCES

We systematically collect GUI data across diverse platforms to construct ScaleCUA-Data, including desktop, mobile, and web environments. As shown in Table. 15, ScaleCUA-Data spans 7 major operation systems: Windows, Ubuntu, macOS, iOS, iPadOS, Android, and Web. Each platform features a broad spectrum of frequently used applications designed for productivity, communication, entertainment, browsing, and utilities.

On desktop platforms, Windows includes both native and third-party applications such as Microsoft Office Suite, Adobe Creative Cloud, Visual Studio, and system utilities, offering a comprehensive view of traditional GUI layouts. Ubuntu and macOS incorporate open-source and system software, including LibreOffice, GIMP, Terminal, Finder, and Safari.

Mobile data is collected from the iOS and Android platforms. The data from the iOS platform includes system applications such as Settings, Safari, Calendar, and Health, as well as third-party applications including Weibo, Notability, and Spotify. The Android platform, by virtue of its open ecosystem, serves as the greatest diversity of data sources, encompassing both system applications

Table 15: The main sources of GUI corpora across different platforms.

| Platform | Application |
|---|---|
| Windows | File Explorer, OS, Chrome, Microsoft Edge, Word, Excel, PowerPoint, LibreOffice Calc, LibreOffice Impress, LibreOffice Writer, Maps, Camera, Calculator, Microsoft Store, Clock, Photos, Outlook, Media Player, VLC Media Player, Calendar, Paint, Paint 3D, QQ Music, KuGou Music, Spotify, Tencent QQ, Visual Studio Code, Dev-C++, Microsoft Solitaire & Casual Game, Pycharm, Android Studio, Vmware Workstation Pro, Vmware Fusion, Adobe Photoshop, Adobe Premiere Pro, Adobe Illustrator, Blender, FL Studio, Unreal Engine, DaVinci Resolve, AutoCAD, SolidWorks, Inventor, Vivado, MATLAB, Origin, Stata, Eviews |
| Ubuntu | Files, OS, Firefox, Chrome, LibreOffice Calc, LibreOffice Impress, LibreOffice Writer, OneNote, GIMP, Slack, Thunderbird, Visual Studio Code, Zotero |
| MacOS | Finder, OS, Safari, Chrome, Pages, Numbers, Keynote, Calculator, Maps, Notes, Calendar, Contacts, Reminders, Apple Music, Podcasts, Weather, Stocks, Freeform, Terminal, Clock, Pycharm, Android Studio, App Store, Mail, Visual Studio Code |
| iOS | Weather, Maps, Find My, Settings, Stocks, Safari, Mail, Calendar, App Store, Home, Camera, Files, Wallet, Contacts, Shortcuts, Clock, Twitter, Weibo, Outlook, Reddit, Instagram, Notes, Keynote, Reminders, Notability, GoodNotes, Rednote, Translate, Calculator, Voice Memos, Shadowrocket, Music, Podcasts, Spotify, iTunes Store, Apple TV, Books, Zhihu, Health |
| iPadOS | Weather, Settings, Safari, Camera, Goodnotes, Translate, Notes, Freeform, Chrome |
| Android | Settings, Clock, Desktop Clock, Calendar, Contacts, Files, Camera, LinkedIn, Weibo, Twitter, Tieba, Reddit, Zoom, Gmail, Duolingo, Xueersi, Wikipedia, XuetangX, edX, Coursera, Skillshare, ZLibrary, To Do, Word, Excel, PowerPoint, OneNote, Taskade, Notion, TickTick, Google Maps, AMap, Tencent Map, Qunar, Trip.com, Ctrip, Qunar, LY.com, Fliggy, Zhixing Train Tickets, Map.me, Booking, Amazon, eBay, Taobao, Alipay, Poizon, VIPShop, 58.com, Beike, Anjuke, Zhuanzhuan, Douyin Mall, Shihuo, Nike, Bilibili, Bilibili CN, QQ Music, himalaya, Classical Music, News, Toutiao, Sohu News, NetEase News, Hupu, Huya, Sohu Video, Pi Music Player, NetEase Cloud Music, Kuaishou, Kugou, WeSing, Douban, Xiaohongshu, Zhihu, Qidian, Xiaoheihe, Prime Video, CNN, Quora, Cantook, Spotify, Apple Music, YouTube, Fitness, Health, JD Health, Translate, Moji Weather, App Store, Google Chrome, BlueCoins, VPN, Shadowrocket, Surfboard, Speedtest, Meitu, Jianying, Canva, Procreate, Pinterest, GitHub, DeepSeek, Grok |
| Web | 5i5j(sh.5i5j.com), AccuWeather(accuweather.com), adidas China(adidas.com.cn), Adobe(adobe.com), Amazon(amazon.com), American Kennel Club(akc.org), Apple(apple.com), arXiv(arxiv.org), BabyCenter(babycenter.com), Baidu(baidu.com), Baidu Baike(baike.baidu.com), Baidu Tieba(tieba.baidu.com), Beihang University(buaa.edu.cn), Bilibili(bilibili.com), BoardGameGeek(boardgamegeek.com), BoardMix(boardmix.cn), Booking.com(booking.com), Budget(budget.com), Cambridge Dictionary(dictionary.cambridge.org), Cars.com(cars.com), CNBlogs(cnblogs.com), CNN(cnn.com), CoinMarketCap(coinmarketcap.com), Coursera(coursera.org), CSDN(csdn.net), Ctrip(ctrip.com), Damai(damai.cn), Dianping(dianping.com), Dior(dior.com), Douban(douban.com), Douyin(douyin.com), Drugs.com(drugs.com), eBay(ebay.com), Britannica(britannica.com), ePay(epay.com), Epicurious(epicurious.com), Facebook(facebook.com), Fastly(fastly.com), FedEx(fedex.com), Fliggy(fliggy.com), Food Network(foodnetwork.com), Gaode Maps(gaode.com), Gmail(gmail.com), GitHub(github.com), Google Finance(finance.google.com), Google Maps(map.google.com), Google Scholar(scholar.google.com), GOV.UK(gov.uk), Healthline(healthline.com), Hugging Face(huggingface.co), Hupu(hupu.com), IGN(ign.com), IMDb(imdb.com), Indeed UK(uk.indeed.com), iQiyi(iqiyi.com), JD.com(jd.com), JetBrains(jetbrains.com), KAYAK(kayak.com), Kohl's(kohls.com), Last.fm(last.fm), LeetCode(leetcode.cn), LinkedIn(linkedin.com), Marriott(marriott.com), Microsoft Azure(azure.microsoft.com), Microsoft Office(office.com), ModelScope(modelscope.cn), MSN(msn.com), NBA(nba.com), National Relocation(nationalrelocation.com), NetEase Cloud Music(music.163.com), Newegg(newegg.com), OpenStreetMap(openstreetmap.org), PayPal(paypal.com), PJLab GitLab(gitlab.pjlab.org.cn), QQ(qq.com), QQ Music(y.qq.com), QS China(qschina.cn), Reddit(reddit.com), Redfin(redfin.com), REI(rei.com), Rotten Tomatoes(rottentomatoes.com), Ryanair(ryanair.com), Samsung(samsung.com), Shimo(shimo.im), Sina News(news.sina.com.cn), Skype(skype.com), SpotHero(spothero.com), Stack Overflow(stackoverflow.com), Steam Store(store.steampowered.com), Student.com(student.com), TensorFlow(tensorflow.org), Tencent Docs(docs.qq.com), Tencent Video(v.qq.com), The Weather Channel(weather.com), The Weather Network(theweathernetwork.com), Thumbtack(thumbtack.com), Ticket Center(ticketcenter.com), Trip.com US(us.trip.com), TripAdvisor(tripadvisor.com), UNIQLO China(uniqlo.cn), United Airlines(united.com), University of Cambridge(cam.ac.uk), University of Michigan(umich.edu), Vmall(vmall.com), Virginia DMV(dmv.virginia.gov), WebArena Forum(wa_forum), WebArena GitLab(wa_gitlab), WebArena Shopping(wa_shopping), WebArena CMS(wa_shopping_admin), WebMD(webmd.com), Weibo(weibo.com), Wikipedia(wikipedia.org), WolframAlpha(wolframalpha.com), X(x.com), Xiaohongshu(xiaohongshu.com), Yahoo Finance(finance.yahoo.com), Yahoo Sports(sports.yahoo.com), Yelp(yelp.com), YouTube(youtube.com), Zhihu(zhihu.com), Zhaopin(i.zhaopin.com), Zhaopin Landing Page(landing.zhaopin.com), Zhipin(zhipin.com) and ∼ 0.2M URLs selected from TOP-1M URLs(*https://tranco-list.eu/*) |

and a broad array of commercial software from domains such as productivity, e-commerce, social media, and multimedia (e.g., WeChat, Taobao, TikTok, and Google Suite).

For tablet interfaces, our data collection primarily focused on iPadOS, encompassing a selection of its most frequently utilized system applications.

As for Web, we collected pages from over 200 frequently accessed websites spanning e-commerce, social media, education, government services, travel, and developer tools. These sources encompass major websites such as Amazon, YouTube, Reddit, Wikipedia, Coursera, and GitHub, with data captured through both static DOM snapshots and dynamic interaction traces.

The collected dataset constitutes a high-coverage, cross-platform corpus of real-world graphical interfaces which endows the model with comprehensive domain knowledge and leads to significantly improved generalization.

### A.6.2 GUI UNDERSTANDING

To support the development of general-purpose computer use agents, we construct a large-scale corpus for GUI understanding that encompasses both element-level and screenshot-level semantics. This corpus is designed to facilitate fine-grained perception and reasoning over static and dynamic user interfaces.

For element-level understanding, we define five task formulations targeting visual appearance, spatial layout, textual grounding, and semantic functionality. First, we introduce the *Element Appearance Captioning* task, which requires the model to describe visual features (e.g., shape, color, borders) of a given GUI component. These attributes often signal affordances and can help distinguish between interactive and static elements. Second, we incorporate *Referring OCR*, a referring task where the model extracts the textual content within a specified bounding box, enabling alignment between visual context and embedded text. Third, to capture spatial organization, the *Element Layout Understanding* task asks the model to predict both absolute screen coordinates and relative positions with respect to nearby components. Fourth, to understand the operational roles of components, we define the *Element Functionality Captioning* task, where the model infers the intended function of a labeled element within its surrounding interface. Finally, we propose a *User Intention Prediction* task, where the model is asked to infer the user's likely goal based on contextual clues and ongoing interactions.

For screenshot-level understanding, we formulate two tasks that promote global comprehension. The *Interface Captioning* task prompts the model to generate a high-level textual description summarizing the overall structure, visual hierarchy, and content of the interface. This encourages holistic reasoning and layout recognition. Complementarily, the *Screen Transition Captioning* task focuses on temporal changes by asking the model to describe the differences between two consecutive screenshots. This enables the model to understand GUI dynamics, such as state updates, navigation events, or content refreshes.

Together, these tasks define a comprehensive benchmark for GUI understanding. We leverage vision-language models to automatically generate annotations for both element-level and screenshot-level tasks, using visual context, structural metadata, and interaction histories. This corpus provides the foundation for training agents capable of fine-grained perception, robust grounding, and high-level reasoning in complex GUI environments.

### A.6.3 METADATA EXTRACTION

**Windows Platform.** To facilitate the automated analysis and interaction with graphical user interfaces (GUIs), we design and implement a framework for extracting UI metadata on the Windows operating system. The core of this framework leverages the UI Automation (UIA) technology to perform a depth-first traversal of an application's A11y Trees, initiated from the foreground window identified via native `Win32`[1] API calls. Subsequently, the collected raw data undergoes a multi-stage filtering and refinement pipeline to ensure its relevance and actionability. This pipeline first performs a geometric validity check to filter out improperly sized or off-screen controls, followed by a visibility and occlusion analysis to retain only the topmost, unobscured elements. Furthermore, a semantic pruning module uses a predefined keyword list (e.g., "close", "save") to remove controls that

---

[1]https://learn.microsoft.com/en-us/windows/win32/

might cause task interruption, while a system component exclusion module discards elements within standard OS regions like the taskbar based on their absolute coordinates. Each element that successfully passes through this pipeline is then abstracted into a structured JSON object. This object encapsulates its multi-dimensional attributes, including identity properties (`control_type`, `name`), state information (`is_enabled`), spatial coordinates (`bbox`), and descriptive text (`description`, `tooltip`). The aggregation of these objects yields a comprehensive metadata representation of the UI, establishing the foundation for subsequent automated tasks.

**Ubuntu Platform.** To extract Ubuntu metadata, we process an XML string representation of the A11y Trees, leveraging Python's built-in `xml` library for parsing[2]. The process commences by parsing the raw XML data into a tree structure. Following this, we linearize these nodes into structural elements. Specifically, for each node in this set, we programmatically extract key attributes, including its tag (representing the element's role), name, class, and description. To capture the semantic content robustly, the element's text is derived either directly from its text content or inferred from its value attribute, particularly for input fields. Positional and dimensional data are extracted from screencoord and size attributes, which together define the element's bounding box. The final output is a structured, tab-separated string where each line represents a single UI element. This entry is composed of seven fields: (1) `tag` indicating the UI type, (2) `name` for the element's given name, (3) `text` capturing its content or value, (4) `class` specifying its component class, (5) `description` for accessibility-related details, (6) `position` as a top-left (x, y) coordinate, and (7) `size` as a width and height pair. In essence, this process distills raw, platform-specific A11y Trees into a flattened, semantically-annotated dataset, providing a crucial foundation for downstream understanding, grounding tasks.

**MacOS Platform.** We extract UI metadata from macOS applications by leveraging the macOS Accessibility API, primarily via the `ApplicationServices`[3] frameworks. It allows structured traversal of the A11y Trees by programmatically accessing on-screen UI windows and querying attributes such as `AXPosition`, `AXSize`, `AXRole`, `AXTitle`, `AXValue`, and `AXDescription`. To initiate the process, we identify top-level windows from the system window list using `CGWindowListCopyWindowInfo`, filter for visible application windows, and create AX references using `AXUIElementCreateApplication`. A recursive collection strategy is then applied, traversing each window's A11y Trees up to a bounded depth while filtering out off-screen or irrelevant elements. To ensure semantic clarity, we enrich metadata by inferring contextual labels for interactive elements (e.g., `AXButton`, `AXTextField`) based on their surrounding static text, spatial layout, and role. Further, we apply spatial deduplication heuristics to eliminate overlapping or redundant elements, and merge content-bearing `AXStaticText` regions with their parent interactive widgets when appropriate. The final output is a flattened list of UI elements, each annotated with role, text content, description, and bounding box information. Structurally, each metadata entry consists of: (1) `role` indicating UI type (e.g., `AXButton`), (2) `text` and `description` capturing semantic content, (3) a `bbox` dictionary with `x`, `y`, `width`, and `height`, and (4) optionally a list of `children` for nested components. This pipeline enables robust and interpretable extraction of macOS GUI structures, supporting downstream tasks such as screen annotation, interaction modeling, and agent behavior learning. Additionally, due to the limited accessibility information exposed by some system-level macOS applications or the difficulty in filtering non-visible elements, we incorporate `omniparser-v2` as a complementary mechanism to refine and validate extracted elements based on screenshot alignment and bounding box overlap.

**Mobile Platform.** For Android, we begin by using UIAutomator2[4] to dump the current app's accessibility hierarchy as XML and parse it into an in-memory `lxml` tree. In a depth-first walk, we record each node's `class`, `resourceID`, `text` and `content description`, and parse its `bounds` string (e.g. `"[x1,y1][x2,y2]"`) into integer coordinates to build robust locators and raw geometry. During this pass we filter out any control that is off-screen, too small (for example, width $< 5$ px or height $< 15$ px), devoid of both text/description and interaction flags (`clickable`, `focusable`, `scrollable`, or `long-clickable`), or fully occluded by its parent—leaving only truly visible, actionable elements. For each remaining node, we generate a concise label by combining up to the first ten words of its text or description with its UI role (e.g. "Button" or

[2]https://docs.python.org/3/library/xml.etree.elementtree.html
[3]https://developer.apple.com/documentation/applicationservices
[4]https://uiautomator2.readthedocs.io/en/latest/

"EditText") and infer possible actions (click, swipe, long press, write). In a second sweep, we detect exactly which elements support taps, focus moves, scrolling, or long presses, then wrap each into a structured record containing its unique identifier, bounding-box coordinates, a summary of core attributes (ID, text, type, state flags like `enabled` and `visible_to_user`), and the full raw attribute map (`package`, `index`, `checkable`/`checked`, `password`, etc.). Finally, we serialize this collection as a flat JSON array or tab-separated lines, producing a complete, coordinate-aware metadata set that underpins precise mobile UI analysis and automated testing. For iOS, we feed the screenshot directly into OmniParser V2 (Yu et al., 2025), which parses the page elements—extracting their type, bounding box, interactivity, content, and so on—and uses this information as metadata.

**Web Platform.** Our web metadata extraction pipeline employs Selenium WebDriver[5] with ChromeDriver[6] to automate web interaction trajectory acquisition using a random walk algorithm. At each step, it leverages browser-native rendering to ensure visual fidelity while capturing the current page's element metadata, including coordinates, descriptions, types, and special attribute information. The pipeline executes a JavaScript parsing pipeline via Chrome DevTools Protocol (CDP) that implements a comprehensive element classification and filter methodology. Clickable elements are identified through a multi-criteria approach combining semantic **HTML tags** (`<a>`, `<button>`, `<input>`, `<select>`, `<textarea>`, `<option>`, `<video>`), **CSS properties** (`cursor:pointer`, since CSS properties cascade to child elements, we only treat an element as clickable if its parent lacks cursor:pointer, ensuring accurate detection of standalone clickable elements), **JavaScript click event listeners**, and **element attributes** (`onclick`, `ondblclick`, `roles` contain button, option, tab); Non-interactive elements are systematically classified as text objects, media objects, or structural panels through DOM hierarchy analysis. All elements undergo rigorous validation including **geometric verification** using `getBoundingClientRect()` to filter occluded components, **visibility validation** through CSS property checks (`display:none`, `visibility:hidden`, `opacity:0`), and **active validation** via `document.elementFromPoint()` center-point sampling to confirm visual prominence and top-layer activity. Finally, we perform a set difference operation with the elements from the last step to filter out the set of new elements for the random walk. Text description metadata aggregation incorporates content from over 12 attributes including `textContent`, `innerText`, `value`, `alt`, `title`, and `aria-label`, normalized through whitespace compression algorithms. The framework implements multiple integrity safeguards including dynamic language detection via langdetect[7], sensitive lexicon pattern matching, and visual anomaly detection with adaptive boundary refinement. Cross-resolution robustness is achieved through randomized viewport initialization spanning device pixel ratios (1.4–2.1) and common resolutions (720p, 1080p, 2K, 4K, 2560×1600), stabilized via CSS viewport normalization techniques. This comprehensive web trajectory metadata extraction pipeline ensures exceptional data integrity, security, diversity, granularity, and accuracy, thereby establishing a robust foundation for instruction construction and model training.

### A.6.4   GUI GROUNDING

GUI grounding is a fundamental capability for computer use agents, enabling them to associate the natural language instruction with a corresponding region of interest. Effective grounding determines whether the agent can interact with the correct interface components, directly impacting its ability to complete downstream tasks. In fact, a grounding-only agent can be paired with a general-purpose planner (e.g., GPT-4o (Hurst et al., 2024)) to complete tasks via a modular style.

To support various grounding demands, we construct a multi-format GUI grounding corpus with three distinct supervision targets: point grounding, bounding box grounding, and action grounding. Point grounding requires the model to identify a single pixel-level location, typically the center of a button, icon, or control, that corresponds to a user instruction. Bounding box grounding extends this capability by predicting rectangular regions that encapsulate target elements, which is particularly useful for operations involving region selection, such as dragging or editing. Action grounding combines spatial localization with operational semantics by producing an executable command, such as `click(x=105, y=23)`, that aligns with the intended interaction. As for the annotation, we reuse structured annotations generated during the GUI understanding stage. Specifically, appearance,

---

[5]https://www.selenium.dev/

[6]https://www.google.cn/chrome

[7]https://github.com/Mimino666/langdetect

spatial, and functional descriptions of each UI element provide rich supervision signals. The center point and bounding box coordinates are extracted directly from UI layout metadata or visual parsing modules. Action-level grounding pairs these spatial targets with predefined atomic operations based on the element's inferred function. In addition, we explore data augmentation strategies to expand the grounding corpus. Specifically, we filter out previously annotated elements from the metadata and use prompt templates combined with GPT-4o to generate a larger set of grounding annotations. This augmented data is designed to improve the model's generalization ability across diverse GUI layouts and interaction patterns. This annotated corpus serves as a foundation for learning robust visual-linguistic alignment and facilitates both direct interaction and integration with high-level task planners.

### A.6.5 WEAK-SEMANTIC TRAJECTORY

While the trajectories collected by rule-based agents do not correspond to explicit task objectives, we incorporate heuristics into the exploration process to encourage transitions into deeper and less frequently visited interface states. This results in more diverse and representative interaction sequences, which are critical for training agents to generalize across complex GUI structures.

To further exploit the potential of these unsupervised trajectories, we segment long interaction sequences into shorter, weakly semantic sub-trajectories. The segmentation is based on screenshot similarity: when a current screen is visually similar to a previous one, it often indicates that the agent has reached a terminal or redundant interface state with minimal novelty in further interactions. These similarity-based boundaries serve as natural points for restarting exploration, thereby improving coverage and trajectory diversity.

We refer to the resulting sequences as *weak-semantic trajectories*, as they preserve partial continuity and structural coherence without being aligned to manually defined tasks. Despite their lack of strong supervision, such trajectories often reflect meaningful UI flows, especially when the agent is biased toward newly rendered elements.

We hypothesize that exposure to weak-semantic trajectories can help the agent internalize common patterns of GUI interaction and enhance its planning ability. If validated, this approach may offer a cost-effective alternative to large-scale manual annotation, accelerating the evolution of more capable computer use agents through low-cost, high-coverage exploration.

### A.6.6 HUMAN-CURATED TRAJECTORY

In addition to rule-driven exploration, we incorporate human-curated trajectories to address the limitations of automatically collected data. While rule-driven agents enable scalable collection, they inherently exhibit stochasticity and often fail to uncover certain goal-directed operations, especially for tasks requiring deep or context-specific interactions. Moreover, although weak-semantic trajectories segmented from raw explorations provide partial structure, their action sequences are not always aligned with human reasoning. As a result, they may contain fragmented or noisy behaviors that limit their utility for downstream training.

To overcome these limitations, we design a human-in-the-loop protocol for collecting high-quality task trajectories. We begin by constructing a seed task set, categorizing applications into common use domains such as daily utilities, entertainment, and productivity. For each domain, we identify representative applications and select frequently used functions based on user documentation and empirical analysis. Annotators are then instructed to convert these functions into clear, goal-oriented task descriptions, ensuring linguistic clarity and operational feasibility. Using our unified cross-platform recording system, human experts remotely interact with each application environment encapsulated within a Docker container. This design provides process isolation, avoids side effects such as misoperation. Annotators are able to finish tasks in a natural and fluent manner, producing coherent action trajectories that reflect realistic usage patterns across platforms.

These curated trajectories serve as high-quality supervision for training agents with accurate planning and execution capabilities. They complement the broader, noisier dataset collected via automation, and provide reference paths that guide model alignment with human intent and behavior.

### A.6.7 ANNOTATION SCHEMES

In our data acquisition, we collect screenshots along with their metadata, which includes all potentially interactive elements on the page. Since different exploration paths can lead to the same state and common states like the homepage are visited frequently, we employed image feature similarity to deduplicate these screenshots. This yields a unique set of interface screenshots paired with their corresponding metadata. To reduce redundancy and mitigate noise within the metadata, we randomly sample 25 to 40 elements per screenshot. These elements are then semantically filtered using GPT-4o to ensure both efficacy and diversity.

For each retained element, we mark its position on the image using a red box with an arrow. By combining with associated metadata, we prompt GPT-4o to generate appearance and position descriptions, and Claude-3.7-Sonnet (Anthropic, 2025) to generate functional descriptions. These serve as ground truth annotations for our Element Appearance Captioning, Element Layout Understanding, and Element Functionality Captioning, respectively. These descriptions are further used to construct grounding tasks, where the appearance and position descriptions are used for non-action grounding and the function description is used for action-based grounding. To simulate all possible positions of elements and accommodate a wider range of usage scenarios, we perform data augmentation. This includes simulating higher resolutions by stitching two images together, as well as cropping elements and pasting them onto solid-color backgrounds or real-world backgrounds from images captured by the author's own device.

For each unique interface screenshot, GPT-4o is also used to generate an overall caption. If the image was not the final step of a trajectory, we additionally provided GPT-4o with the subsequent screenshot along the same exploration path to summarize the UI changes and infer the intention. These are used for Screen Transition Captioning and User Intention Prediction tasks.

For all trajectories, we provide Claude-3.7-Sonnet with the current and next screenshots, as well as a cropped image of the interacted element, to infer both the step-level instruction and the reasoning process. For weakly semantic trajectories that primarily involve navigation across pages, we generate high-level task objectives. To do this, we provide Claude-3.7-Sonnet with the first and last screenshots of the trajectory to synthesize a navigation-related task goal. Considering that different annotators have varying styles of writing instructions and different operational habits, we implement two types of augmentations for trajectories to improve model generalization. The first is instruction augmentation, where we prompt the model to generate task instructions in diverse styles, aiming to cover all possible user scenarios. The second is trajectory augmentation, for which we prompt the model to generate several step-level instructions and the reasoning process based on the trajectory. This can help mitigate the noise introduced by model labeling. All prompts used for annotation are provided in the Appendix A.10.2.

### A.6.8 MORE DETAILS OF DATA DISTRIBUTION

In Fig. 11, we visualize the data distribution for each task domain. Fig. 11b provides a hierarchical view of the trajectory composition across platforms and types. By integrating agent-generated and expert-curated signals, we ensure both data diversity and quality. Our ScaleCUA-Data delivers the largest GUI grounding dataset to date, coupled with substantial understanding and planning examples. Its platform coverage and hierarchical task composition form a comprehensive foundation for training robust, cross-platform GUI agents. The performance of ScaleCUA validates the quality of ScaleCUA-Data, and highlights future directions in data-centric agent training.

### A.7 DATA VISUALIZATION

### A.7.1 GUI UNDERSTANDING

To qualitatively demonstrate our data in GUI understanding tasks, we provide examples that cover both element-level and screenshot-level understanding. At the element level, we have designed five distinct tasks regarding individual GUI elements. Table 17 showcases specific examples of these tasks. At the screenshot level, we focus on the ability to comprehend the entire GUI interface globally and its dynamic changes. Table 18 provides examples for these two tasks.

Table 16: Distribution of examples in our training corpus.

| Task Domain | Tasks | #Images | #Examples |
|---|---|---|---|
| Understanding | Element Appearance Captioning, Referring OCR, Element Layout Understand, Element Functionality Captioning, User Intention Prediction, Interface Captioning, Screen Transition Captioning | 355.5K | 471.4K |
| Grounding | Bounding Box Point, Action | 1.6M | 17.1M |
| Task Planning | Weak Semantics Trajectories Human-Curated Trajectories Enhanced Trajectories | 5.5K 29.3K 29.3K | 15.0K 4.0K 48.2K |

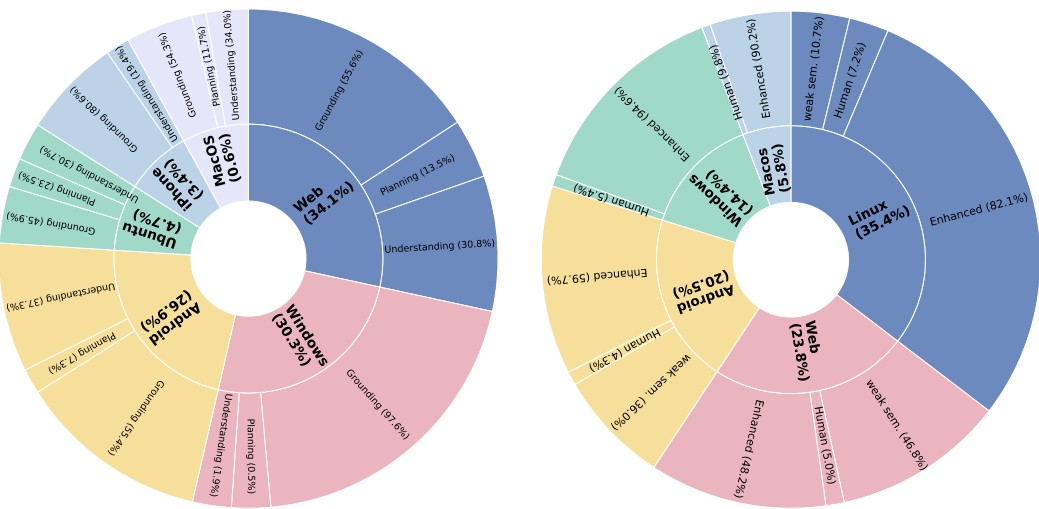

(a) Task domain distribution of our dataset.    (b) The composition of Trajectories.

Figure 11: Data distribution of our dataset.

### A.7.2 GUI GROUNDING

To enable precise interaction, a model must be able to map natural language instructions to specific elements on an interface. As mentioned in Section 4.2, we support three distinct GUI grounding modes to accommodate a variety of scenarios and Table 19 provides a visual illustration of these three modes.

### A.7.3 WEAK-SEMANTIC TRAJECTORY

As mentioned in Sec. A.6.5, the weak-semantic trajectory processing pipeline begins with rule-driven agents performing depth-first exploration across GUI environments. These agents generate extensive interaction sequences by randomly selecting actions from available options at each step, while heuristic pruning eliminates redundant branches to improve efficiency. Although these trajectories lack explicit high-level goals, they provide broad coverage of interface states. To enhance the utility of these unsupervised trajectories, we segment them into coherent sub-sequences based on visual similarity between screenshots.

Fig. 12 demonstrates the exploration pattern in desktop environments, while Fig. 13 and 14 show the equivalent processes for mobile and web platforms, respectively. In each case, the random-walk agent generates comprehensive navigation patterns through systematic exploration of the interface.

Table 17: Task visualization for element-level understanding.

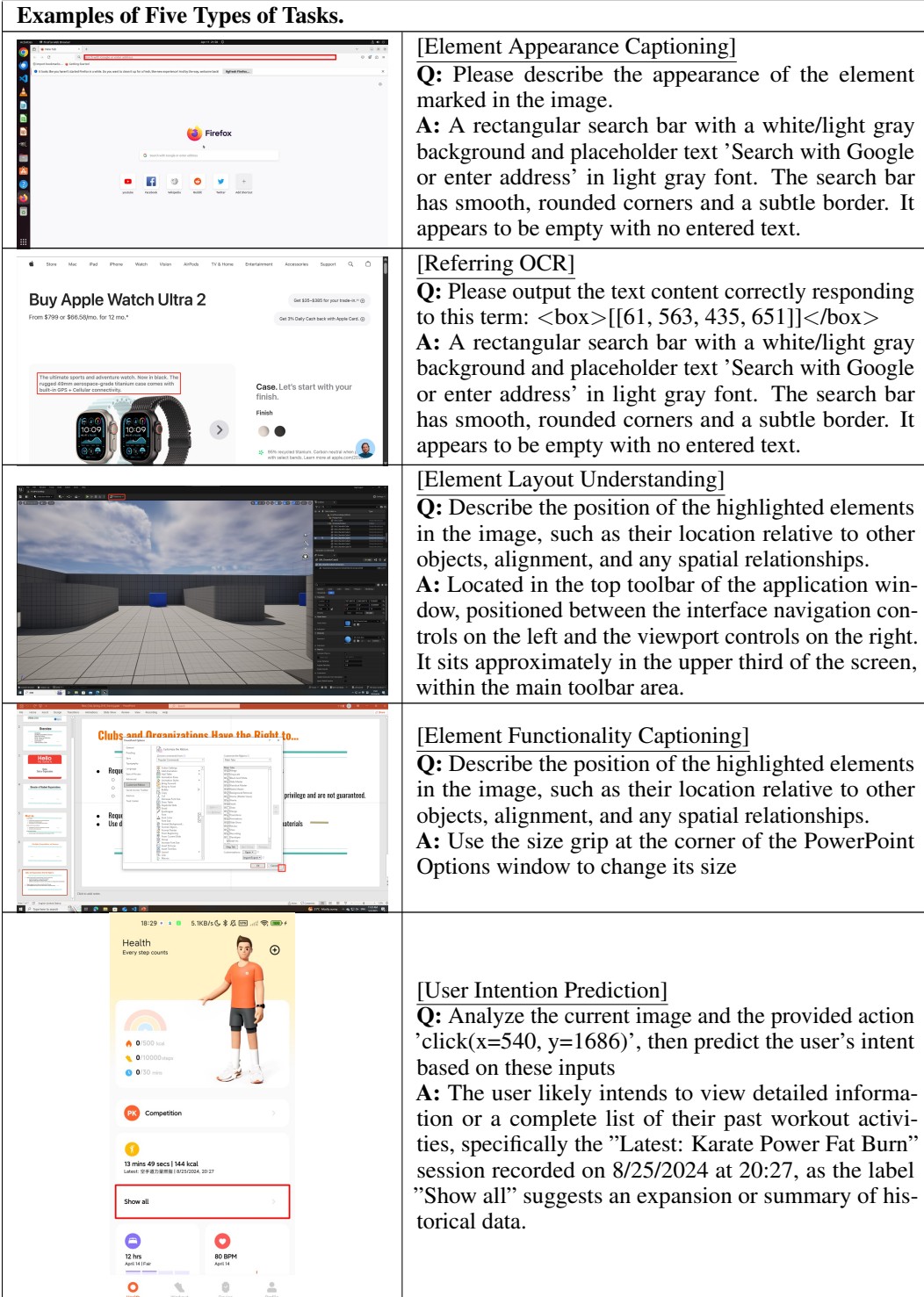

These weak-semantic trajectories serve as valuable training signals despite their lack of strong task alignment. They expose models to common GUI interaction patterns and state transitions, complementing the more structured but limited human-curated data.

Table 18: Task visualization for screenshot-level understanding.

| Examples of Two Types of Tasks. | |
|---|---|
| 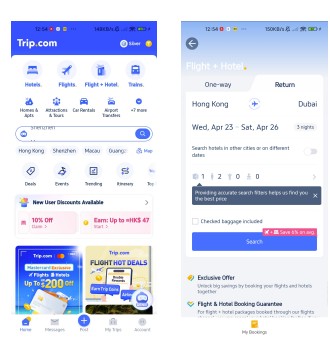 | [Interface Captioning]
**Q:** Provide a detailed description in the current image.
**A:**The image shows a weather application interface displaying the forecast for Shanghai. The current temperature is 66°F (about 19°C) with sunny conditions. The interface shows a detailed forecast for Friday, April 4, 2025, with a high of 69°F and a low of 48°F. The hourly forecast indicates temperatures will range from 66°F to 50°F throughout the day, with sunset occurring at 6:14 PM. Wind conditions show a speed of 3 MPH with gusts up to 11 MPH. The interface also displays additional weather information including visibility (14 mi), UV index (7 - High), and a five-day forecast showing temperatures for upcoming days. The air quality map shows green areas indicating good air quality around the user's location. |
| | [Screen Transition Captioning]
**Q:** Analyze the differences between two consecutive GUI screenshots. Describe the initial state, highlight the changes in the second screenshot.
**A:**In the first screenshot, we see the Trip.com mobile app homepage with various booking options including Hotels, Flights, Flight + Hotel, Trains, and other services. After tapping on Flight + Hotel option, the second screenshot shows the Flight + Hotel booking page where users can input their travel details. The page displays a one-way/return toggle (with 'Return' selected), origin and destination fields set to Hong Kong and Dubai respectively, travel dates (Wed, Apr 23 - Sat, Apr 26 for 3 nights), traveler count settings, and a search button at the bottom. |

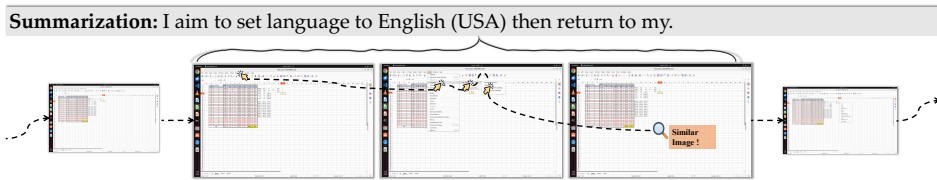

Figure 12: An example of a weak semantic trajectory on the Ubuntu platform.

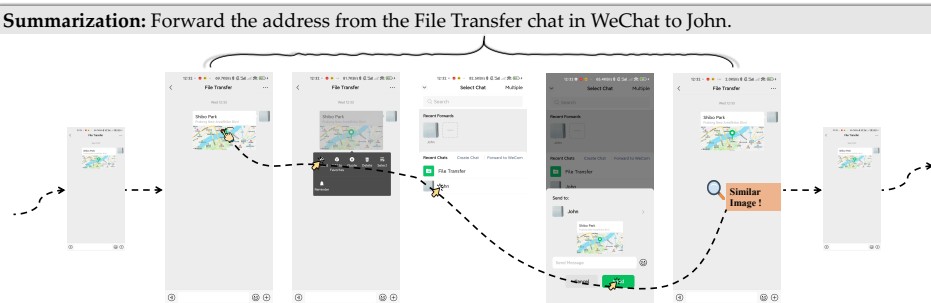

Figure 13: An example of a weak semantic trajectory on the Android platform.

Table 19: Task visualization for GUI grounding

| Examples of Three Modes. | |
|---|---|
| | [Point Grounding]
**Q:** Return the point within this UI element: <ref>Preformatted Text button in the EDITOR tab's formatting toolbar that allows users to insert pre-formatted text tags in MATLAB's editor.</ref>
**A:** <ref>Preformatted Text button in the EDITOR tab's formatting toolbar that allows users to insert pre-formatted text tags in MATLAB's editor.</ref><point>[[223, 45]]</point> |
| | [Bbox Grounding]
**Q:** Indicate the location with a bounding box to this UI element: <ref>A white-faced analog clock with black numerals (1-12) and three hands, placed in the upper left corner.<ref>
**A:** <ref>A white-faced analog clock with black numerals (1-12) and three hands, placed in the upper left corner.<ref>>[[97, 69, 218, 227]]<bbox> |
| | [Action Grounding]
**Q:** Click the 'Open' option to open the selected file
**A:** <action>click(x=0.7983, y=0.4967) </action> |

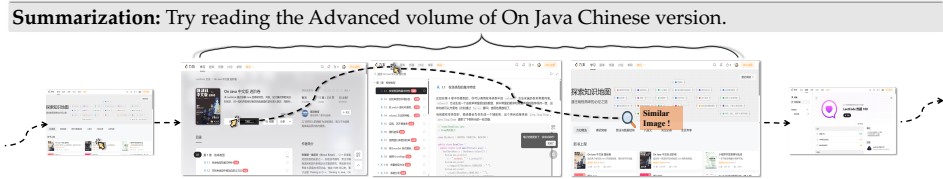

Figure 14: An example of a weak semantic trajectory on the web platform.

### A.7.4 HUMAN-CURATED TRAJECTORY

Fig. 15-19 illustrate human-curated trajectories across five platforms: Windows, Ubuntu, macOS, Android and Web. Each trajectory demonstrates precise human-annotated interactions, rendered as `mouse/gesture` traces over consecutive screenshots, forming high-quality demonstrations for data collection. These trajectories span diverse applications such as Excel, SolidWorks, Gmail, Numbers, Amap, Twitter/X, and GitHub, showcasing real-world complexity in cross-platform environments. The visualizations highlight platform-specific GUI logic (e.g., desktop file operations vs. mobile touch navigation), as well as long-horizon reasoning steps (e.g., multi-page exploration, search-before-edit workflows).

### A.7.5 TRAJECTORY ANNOTATION

Building upon the annotation schemes detailed in Sec. A.6.7, we systematically process trajectory data to generate high-quality training corpora. Our trajectory annotation focuses on two key aspects: (1) low-level operational instructions generated for each interaction step, (2) chain-of-thought rationales explaining the decision process. As demonstrated in Ta-

**Task:** Visualize the 2024 business performance metrics using a clustered column chart in Excel.

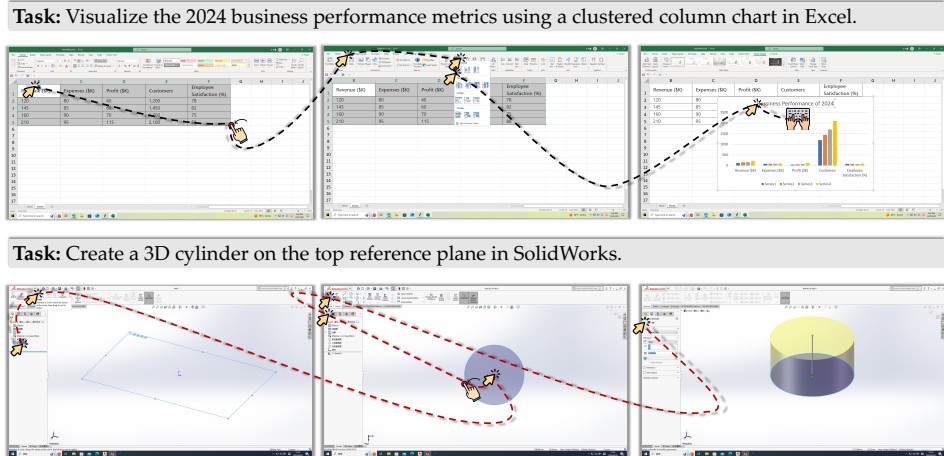

**Task:** Create a 3D cylinder on the top reference plane in SolidWorks.

Figure 15: Examples of human-curated trajectories on the Windows platform.

**Task:** Make a copy of sheet1 and name it as "sheet1(copy)", positioning it after all existing sheets.

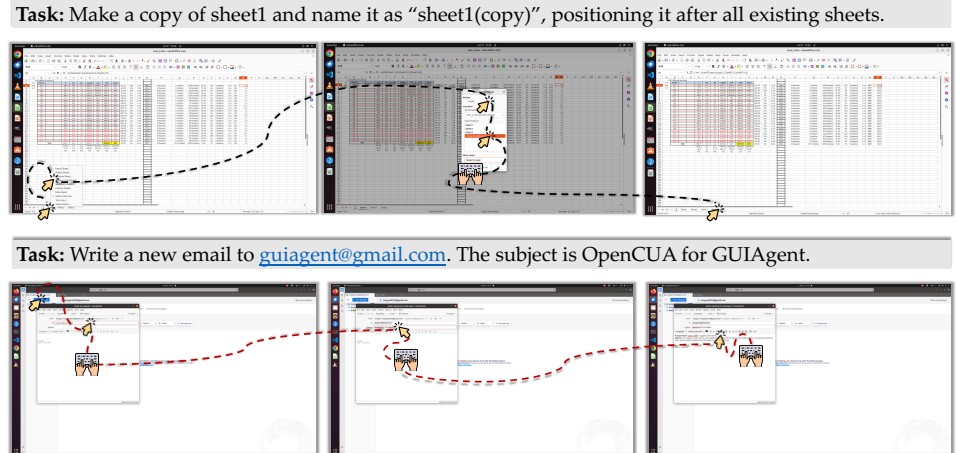

**Task:** Write a new email to guiagent@gmail.com. The subject is OpenCUA for GUIAgent.

Figure 16: Examples of human-curated trajectories on the Ubuntu platform.

**Task:** Use conditional formatting to highlight values greater than 20 in Numbers.

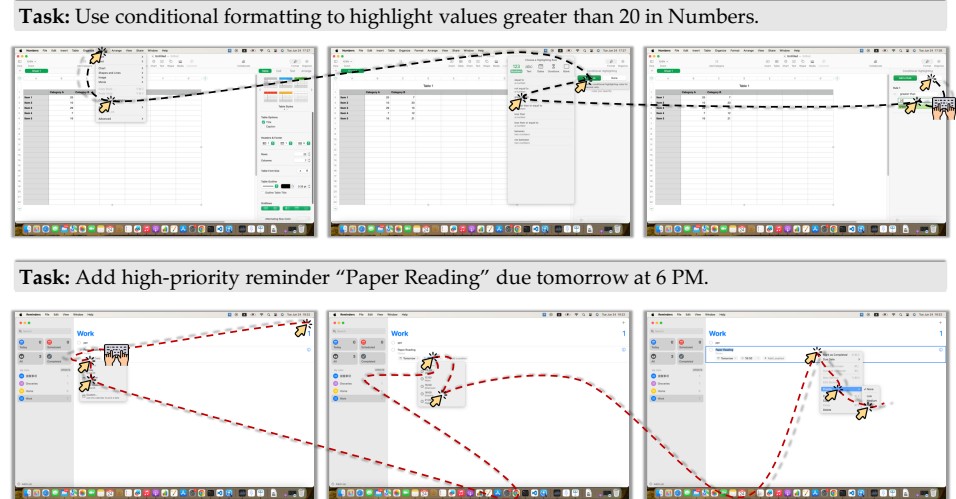

**Task:** Add high-priority reminder "Paper Reading" due tomorrow at 6 PM.

Figure 17: Examples of human-curated trajectories on the macOS platform.

ble 20, these annotations are formally represented using XML tags to distinguish between operational instructions (`<operation>...</operation>`) and their cognitive justification

**Task:** Search "ChatGPT" in X and like the first tweet in the results.

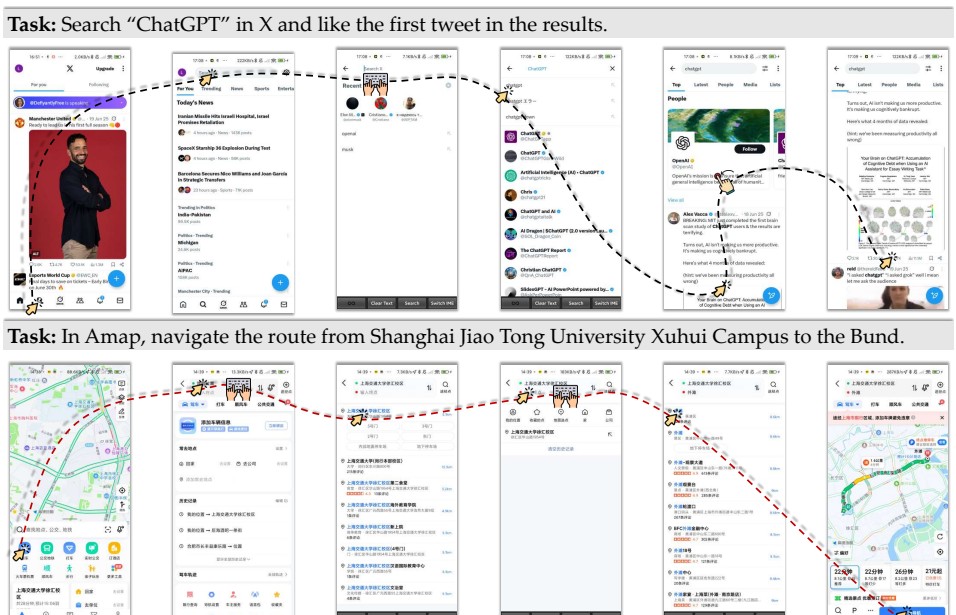

Figure 18: Examples of trajectory data collection on the Android platform.

**Task:** Check the public list of members of the "google" organization.

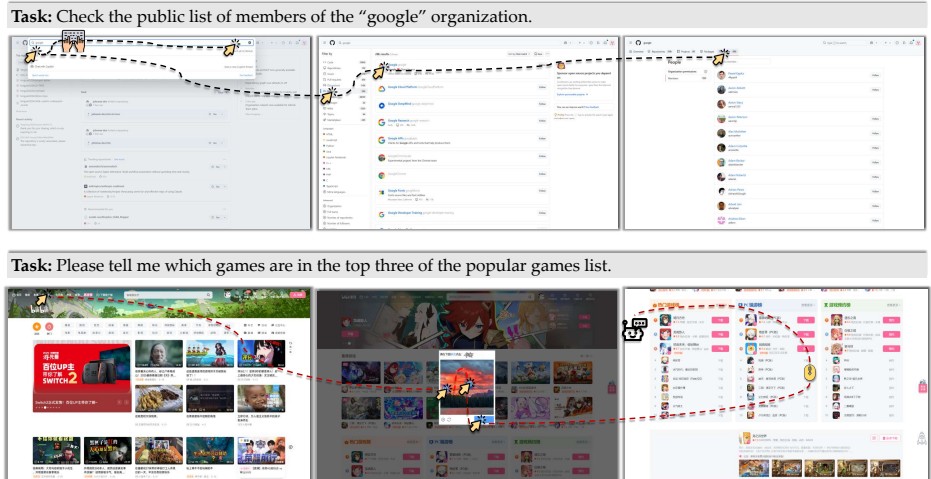

Figure 19: Examples of trajectory data collection on the web platform.

(`<think>...</think>`), which support both direct-action execution and reasoned-action paradigms.

## A.8 LESSONS FROM DATA ACQUISITION

### A.8.1 COMMON PROBLEMS

**Adaptive Bounding Box Tightening(ABT).** The heterogeneity across platforms results in significant discrepancies in UI interaction logic and responsive regions. Therefore, we aim to ensure that the bounding boxes of UI elements are as close as possible to their central regions, in order to minimize inconsistencies in actionable areas across different platforms. As illustrated at the top of Fig. 20, these parsed bounding boxes are aligned with the boundaries of the UI elements; however, their centers, which serve as the ground truth for click prediction, often reside in empty regions rather than in areas with meaningful content. This misalignment introduces a detrimental click-whitespace bias during model training. To mitigate this issue, we propose the Adaptive Bounding Box Tightening (ABT)

Table 20: Visualization of trajectory annotation

| **Examples of Two Modes.** | |
|---|---|
|  | [Reasoned-Action Mode]
**Q:** Please generate the next move according to the UI screenshot, task and previous operations.
Task:
Search openai and like the first video, sorted by view count and video type.
Previous operations:
Step 1: Click on the search box at the top of the YouTube page and type "openai"
Step 2: Search for OpenAI on YouTube and press Enter to see search results and the OpenAI channel page
**A:** \<think\>
The YouTube interface shows a search for "openai" with a filters button visible in the top navigation area. Clicking on the filters option would allow sorting videos by criteria such as view count, which is needed to complete the task of finding the most viewed OpenAI videos. This filtering functionality is essential to organize search results in a way that aligns with the requirement to sort by view count and video type before liking the first video.
\</think\>
\<operation\>
Click on the "Filters" button at the top right of the YouTube search results to access advanced filtering options for your search.
\</operation\>
\<action\>
click(x=0.9043, y=0.0788)
\</action\> |
|  | [Direct-Action Mode]
**Q:** Please generate the next move according to the UI screenshot, task and previous operations.
Task:
In the Calendar app, show only events in the 'School' calendar.
Previous operations:
None
**A:** \<operation\>
Click on the blue checkbox next to "Family" in the calendar sidebar under "On My Mac" section.
\</operation\>
\<action\>
click(x=0.0187, y=0.1128)
\</action\> |

algorithm. ABT dynamically refines bounding boxes through iterative, proportional contraction of their lateral boundaries. This contraction is guided by the detection of contiguous uniform-color regions, representing target whitespace, within defined tolerance thresholds. The process terminates when no significant contractible uniform regions remain, yielding substantially tighter bounding boxes where centers align with actual content, as shown in the bottom of Fig. 20. While ABT's effectiveness is inherently limited by complex backgrounds and gradients, empirical validation confirms its significant value in improving ground truth alignment for interfaces featuring simple solid-color backgrounds. This paradigm remains dominant in modern systems and web design.

**Deep Exploration.** Modeling GUI platform state transition graphs presents inherent complexity. Random walks, a common approach, suffer from limitations: unpredictable transitions induce pervasive back edges, causing frequent state revisitation or trapping in local loops due to insufficient backtracking mechanisms. To address these issues and enable automated deep exploration for acquiring meaningful weakly semantic trajectories, we propose a single-history-frame element filtering algorithm. Specifically, we use a queue to maintain all interactive elements appearing in the last screenshot. At each exploration step during random walk, some of elements are filtered out when their Intersection over Union (IoU) exceeds a predefined threshold and their textual content exactly matches any element in the queue. This guarantees exclusive interaction with elements absent in the preceding state, thereby actively steering exploration toward novel pages. This mechanism proves particularly effective for interfaces with persistent components (e.g., navigation bars, sidebars) or dense icon arrays, as evidenced in Fig. 21 where it achieves significantly broader page coverage and yields non-redundant, semantically valuable trajectories compared to conventional random walks.

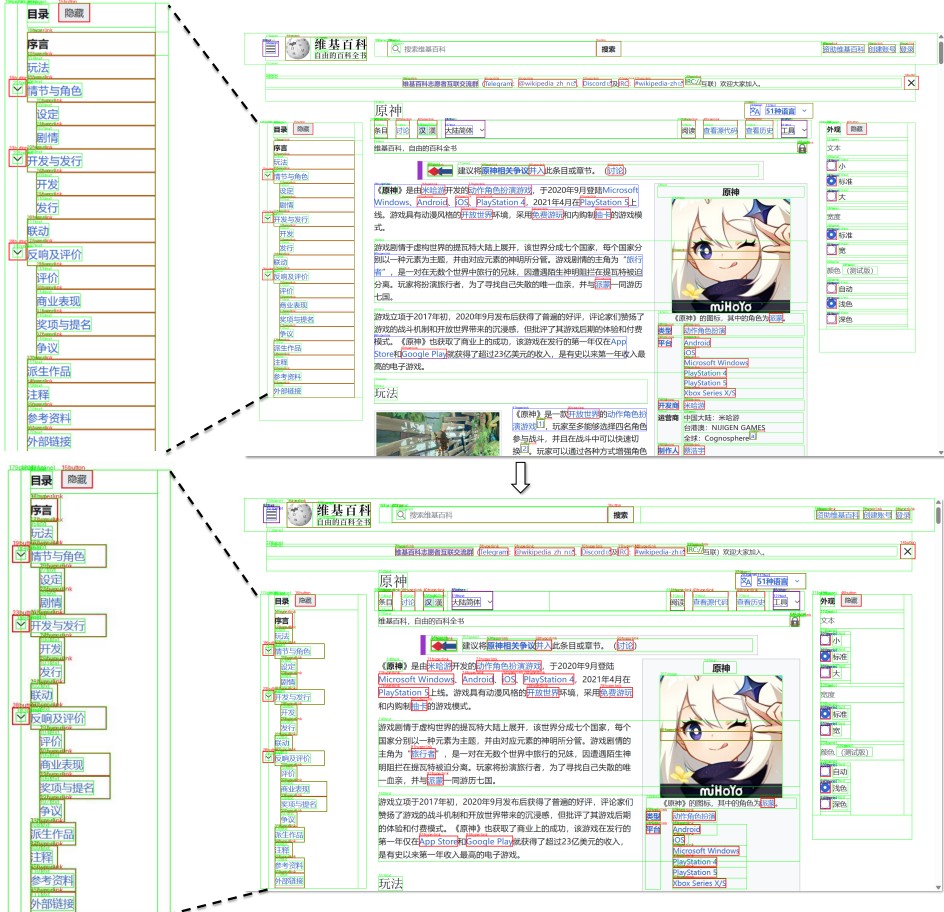

Figure 20: Examples of adaptive bounding box tightening(ABT) algorithm.

### A.8.2 WINDOWS

**Cross-Framework UI Parsing Challenges and Denoising Strategies.** When processing Java-based software like PyCharm and Android Studio, the standard `Win32` API exposes significant limitations. As illustrated in Fig. 22, the `Win32` API fails to effectively parse their UI structure, resulting in an incomplete A11y Tree. Consequently, we must switch to using the specialized Java Access Bridge (`JAB`[8]) API. The JAB successfully retrieves the complete A11y Tree (as shown in Fig. 23, thus resolving the issue. This requirement to adapt different APIs for various application frameworks significantly increases the complexity of our data collection efforts. Moreover, the raw A11y Trees

---

[8]https://docs.oracle.com/javase/8/docs/technotes/guides/access/jab/index.html

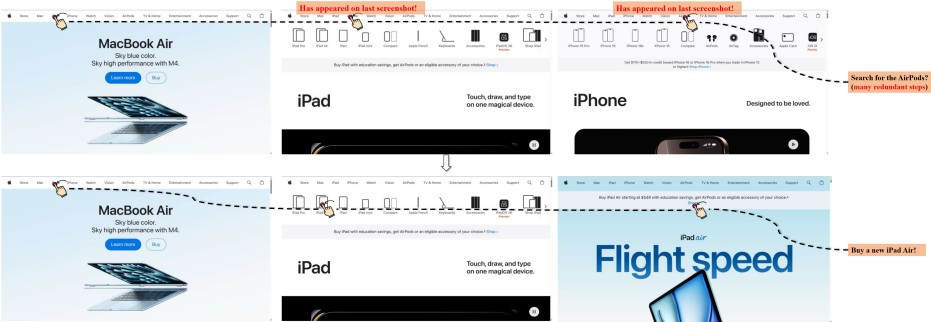

Figure 21: Examples of deep exploration algorithm.

present challenges: they are typically deeply nested, noisy, and the density of functionally relevant UI elements is low. To address these issues and improve data quality, we apply a set of heuristic filters to prune and refine the tree. 24 visualizes this transformation, showing a comparison of the A11y Tree before and after processing. Our filtering strategies exclude elements from background applications and select elements by their screen-to-area ratio, roles (e.g., `button`, `text`, `hyperlink`).

**Data Deduplication and Geometric Refinement.** Data acquisition in Windows faces several significant data quality challenges. First, minimal UI changes following user interactions lead to high redundancy of UI elements due to nearly identical screen captures. Second, lack of layer information in the A11y Trees results in erroneous inclusion of occluded elements (e.g., `dropdown`).To overcome these challenges, we implemented a multi-stage refinement pipeline. We first mitigate redundancy with a similarity algorithm that filters images based on the Euclidean distance of their feature vectors. A post-processing filter then identifies occluded elements by detecting solid-color regions within their bounding boxes.

**Prioritized Random Walk for Automated UI Exploration.** The random walk algorithm is central to our automated data acquisition on the Windows platform. To minimize redundant interactions and enhance element diversity, we have augmented the standard Random Walk with principles from Depth-First Search (DFS). As mentioned in the above common problems, our modified algorithm prioritizes interaction with newly appeared UI elements while concurrently reducing the selection priority of elements that have already been interacted with. If no new elements are detected, or if their count falls below a predefined threshold, the algorithm defaults to interacting with any remaining, previously unvisited elements within the current view's A11y Trees. Furthermore, we account for scenarios where interactions navigate away from the primary application, such as launching a web browser to view a user manual. In such cases, our algorithm allows for limited interaction within the external application (e.g., the browser) before automatically shifting focus back to continue navigating the initial application.

### A.8.3 UBUNTU

This section details the challenges encountered and solutions developed for autonomous agent interaction with the Ubuntu environment. The primary challenges originate from the inherent structure of the accessibility tree (A11y tree), which serves as the main interface for observing and interacting with the application. Our solutions focus on refining the accessibility tree data and optimizing the agent's interaction strategy to ensure reliable and efficient operation. The successful resolution of these issues is paramount, as a clean, accurate, and efficiently navigable UI representation is the foundation for any effective automated UI-based task.

**Denoising in the Accessibility Tree.** The raw data provided by the accessibility tree on Ubuntu is often noisy, containing redundant information and occasional inaccuracies that can mislead an autonomous agent. We identified and implemented solutions for three primary issues. First, the A11tree's hierarchical structure often includes redundant parent elements that do not correspond to distinct interactive components. This is particularly prevalent in applications built with Web, such as Chrome. To address this, we apply a two-stage filtering process. We begin by pruning elements whose roles are typically non-interactive or structural based on `type`, such as 'heading',

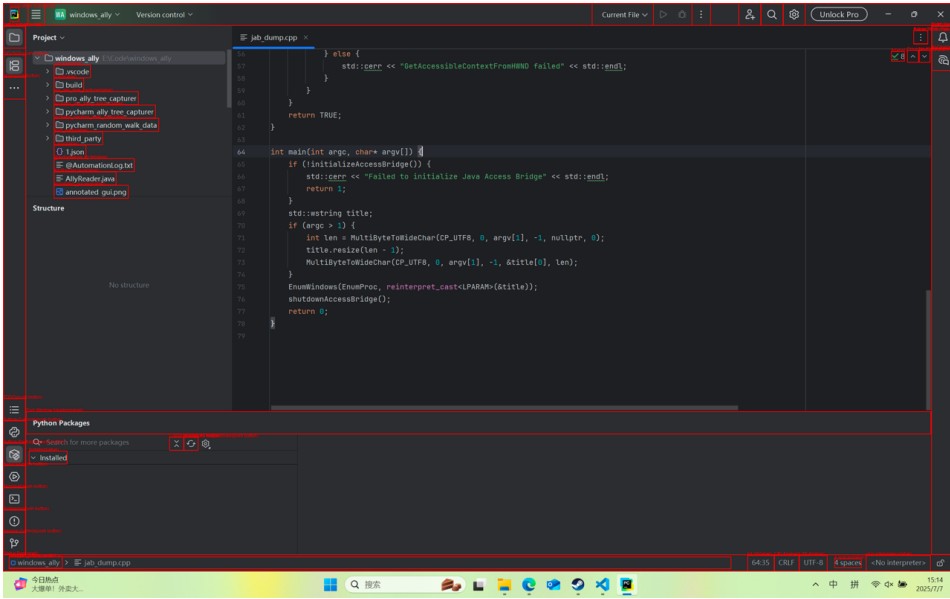

Figure 22: An example of `Win32` API failing to parse A11y Trees in PyCharm.

Figure 23: An example of `JAB` API successfully parsing A11y Trees in PyCharm.

'paragraph', and 'section'. This denoising is critical because it exposes the true, underlying interactive elements, preventing the agent from attempting to interact with large, non-interactive container widgets. Subsequently, we analyze the geometric relationships of the remaining elements' bounding boxes. If an element's bounding box is significantly occluded by a smaller one (i.e., the smaller box's area occupies a large percentage of the larger box's area), we infer a container-child relationship and discard the larger, containing element. Second, the standard accessibility tree does not inherently account for the visual occlusion and invalidity of elements. An element may be present in the tree but be completely obscured by another element on the screen as shown in Fig. 25. We tackle this with LLMs. Third, we observed that for certain applications, the accessibility tree reports incorrect coordinates for all UI elements immediately after the application is launched, as shown in Fig. 26. The entire tree appears to have a coordinate offset. Through empirical testing, we discovered a practical solution: initiating a short sequence of random interactions within the application window causes the accessibility tree's coordinate system to recalibrate, restoring correct positional data. Ensuring coordinate accuracy is fundamental; without it, any attempt by the agent to click or type at a specific location would fail, rendering automation impossible.

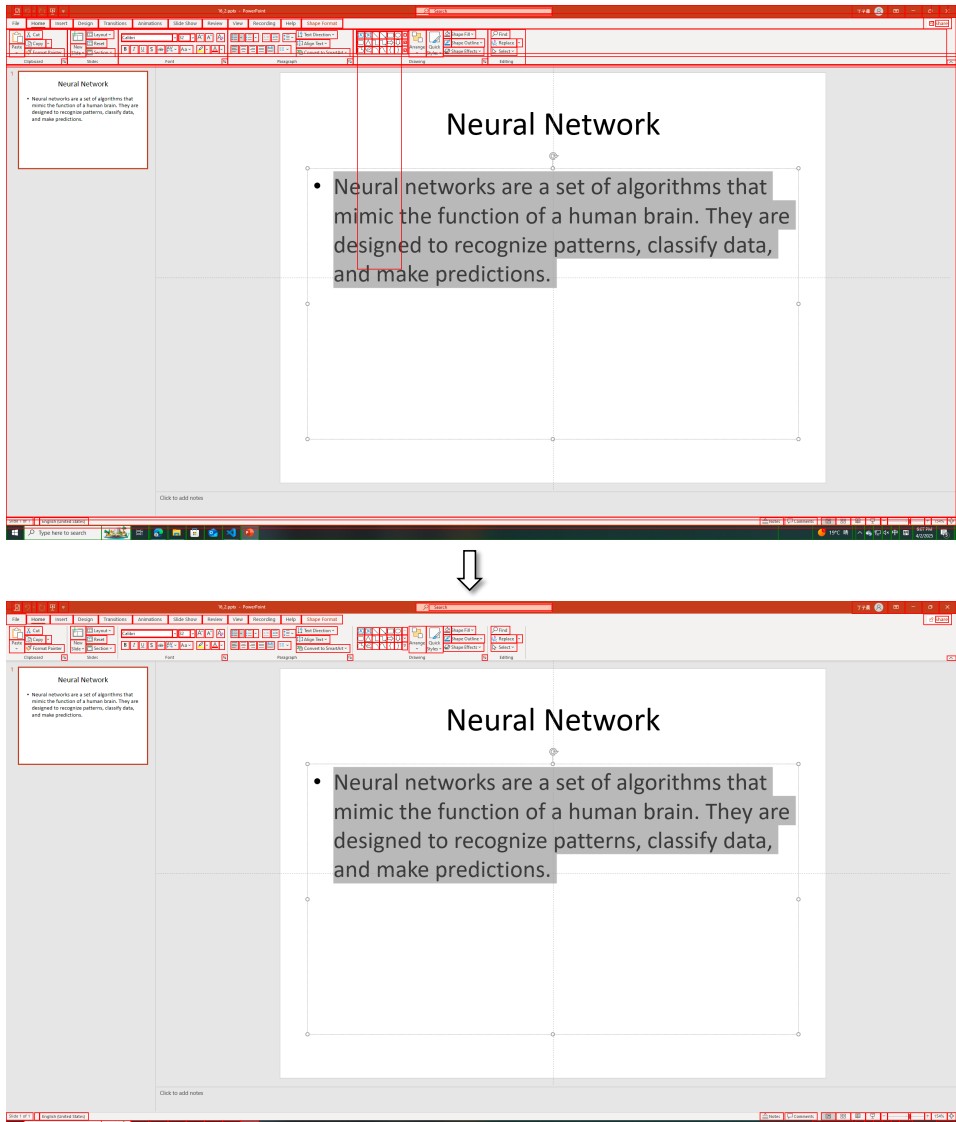

Figure 24: An example of denoising on Windows, transforming a raw A11y Tree (top) into a clear structure (down).

**GUI Exploration Optimization via an Improved Random Walk.** A pure random walk over all available UI elements is highly inefficient. To improve the agent's ability to explore an application's state space, we developed a more intelligent interaction strategy. This strategy is based on filtering the action space and prioritizing the exploration of novel UI states. To reduce the number of futile actions, the agent's action space is constrained to only include elements that are designated as interactive. We maintain a whitelist of interactive `type`, including 'button', 'box', 'menu', 'entry', 'link', 'bar', and 'item'. Conversely, elements with non-interactive roles are excluded from the potential action set. These non-interactive roles include 'heading', 'static', 'document', 'label', 'cell', 'text', 'icon', 'paragraph', and 'section'. To prevent the agent from becoming trapped in interaction loops within a static UI state, we implemented a state-aware exploration logic. After the agent acts, we only visit newly appeared UI elements. These novel elements are given interaction priority, as they are most likely to lead to a new application state. If the action does not yield any new elements, the agent then selects an action randomly from the set of previously known elements that it has not yet interacted with in the current state. This process continues until all interactive elements have been exhausted. This exploration strategy is vital for efficiency, as it directs the agent towards discovering

new functionalities and application states, thereby maximizing the coverage of the application's features in a limited time and avoiding redundant, non-productive interactions.

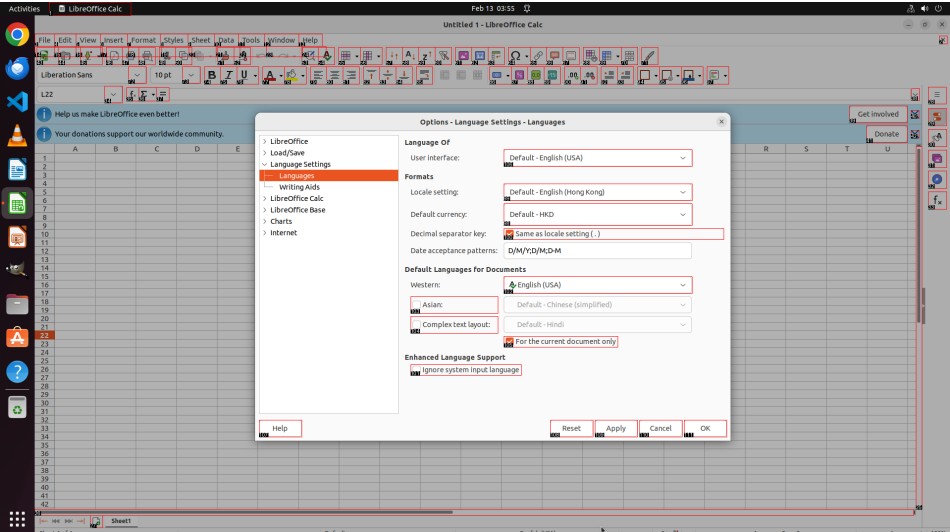

Figure 25: Examples of visual occlusion and invalidity of elements.

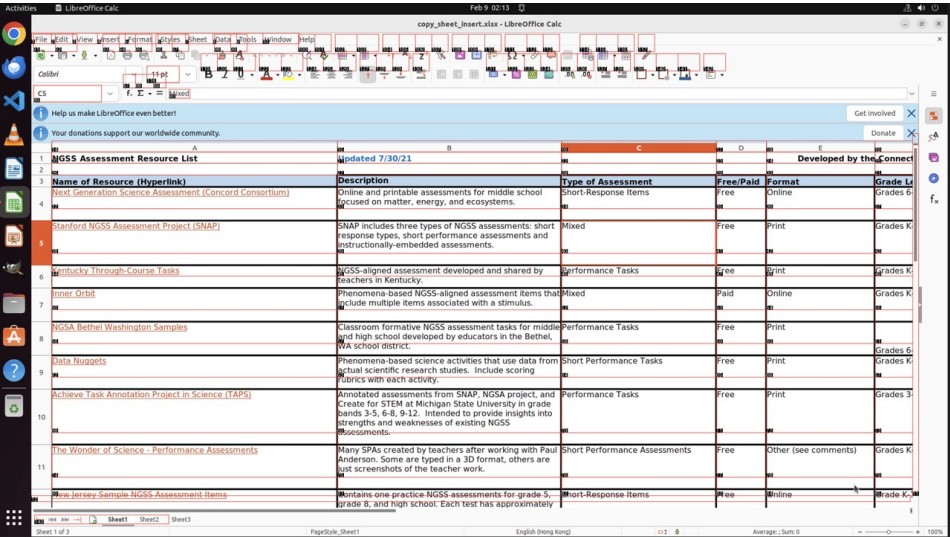

Figure 26: Examples of coordinate offset.

### A.8.4 MACOS

**Robust A11y Tree Extraction and Denoising.** The macOS pipeline first locates the active top-level window, then exhaustively traverses its accessibility hierarchy. Every bounding box is mapped from logical coordinates to device pixels by multiplying by the screen-scale factor. After flattening the tree, only nodes whose roles are interactive (e.g. `AXButton`, `AXPopUpButton`, `AXTextField`) are retained. Moreover, we would discard boxes with a width or height of 2px or less and remove nodes whose `text`, `description`, and `value` are all empty or punctuation. A role-aware merging process replaces overlapping AXStaticText siblings and their interactive parent with a minimal bounding box. The resulting set contains clean, tightly localised interactive elements. (see Fig. 27).

**Hybrid A11y Tree & Omniparser combination for System Panels.** Several built-in utilities, most notably *System Settings*, draw controls in private layers that have no corresponding accessibility tree Yu et al. (2025), as shown in Fig. 28. To recover these missing widgets, each screenshot is processed by Omniparser, yielding a set of vision-detected bounding boxes. An element would be

retained when its IoU with any Omniparser box exceeds $0.15$ or when it is selected during exploration. This combination renders previously invisible elements in the A11y tree, thereby yielding a more comprehensive understanding of macOS applications.

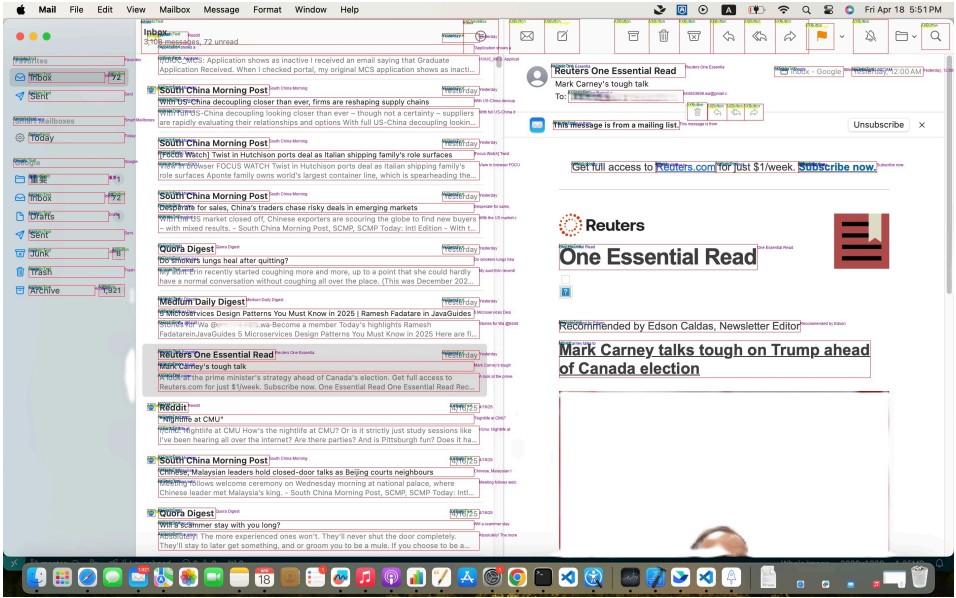

Figure 27: Refined `AXTree` overlay on the Mail application: all interactive elements are tightly bound after heuristic pruning.

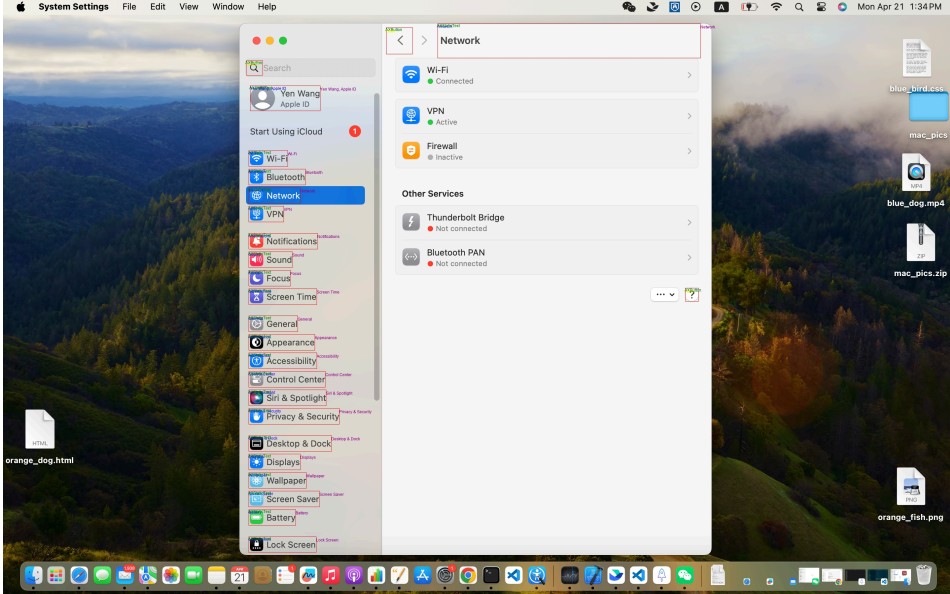

Figure 28: The failure case in System Settings: the `AXTree` omits right-pane controls, illustrating the necessity of Omniparser fusion.

### A.8.5 MOBILE

**Occlusion and Invisibility Correction.** Mobile interfaces frequently employ high-level components such as dialogs, side drawers, and floating menus. These are rendered at the topmost Z-order, so underlying nodes remain in the XML yet can no longer be clicked, producing "ghost" targets(see Fig.29a). To improve visibility and hierarchy accuracy at the source, we replace the traditional `adb shell uiautomator dump` with `uiautomator2.dump_hierarchy()`. The latter

prunes recognisably occluded nodes while generating the XML and, for pages that `adb` fails to parse, still returns a complete hierarchy—significantly increasing data coverage. Coupled with the random-walk heuristic that "prioritises newly appeared elements," this greatly reduces mis-clicks caused by occlusion. In addition, UIAutomator2 markedly lowers the probability of XML retrieval failures, accelerating exploration efficiency.

**Attribute Completion and Correction.** Many commercial apps do not fully propagate accessibility traits in their custom views; a typical pattern is a parent node with `clickable=true` while all its children are `clickable=false`, leading to the issue shown in Fig.29b. Genuine clickable regions are thus ignored. We employ an "inherit-then-suppress" strategy: when a parent is clickable and every descendant is marked non-clickable, the clickable flag is inherited downward; if any descendant is already declared clickable, inheritance stops to avoid creating false hotspots. Experiments show that this method restores the vast majority of missing attributes while maintaining a low false-positive rate.

**Semantic and Functional Ambiguities.** Semantic ambiguity arises when an XML bounding box is too large and covers multiple sub-controls (for example, the playback button, author area and more-options button), making a single node unable to convey precise meaning. In Fig.29c, the green box shows one clickable bounding-box region in the XML, but taps in different parts of that region may produce different results, creating semantic ambiguity. To address this, we prioritise leaf nodes and tighten their bounding boxes; we only retain a parent node when its centre lies outside every child's bounds, thus preserving the overall intent of the composite control. Functional ambiguity occurs when the same layout triggers different actions in different software or operating systems. In Fig.29d and Fig.29e, for example, both the text and the icon of a switch are tappable in stock Android settings, whereas in iOS only the icon responds to taps and the rest of the region is inert. To reduce such mispredictions, whenever we detect an "icon + text" sibling pattern we give the icon a higher click priority. This approach produces more consistent cross-device behaviour during training and testing. By systematically handling overlay occlusion, attribute omissions and both semantic and functional ambiguities, we significantly improve the reliability of mobile-side data collection and increase the success rate of downstream automation tasks.

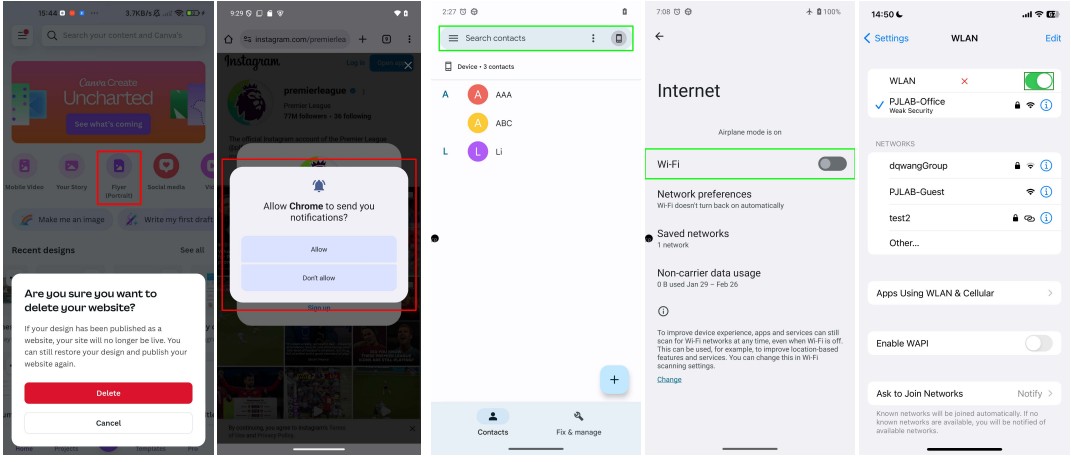

| (a) Mobile element occlusion | (b) Mobile element attribute loss | (c) Semantic ambiguity | (d) Effective bounding box for setting WiFi in Android | (e) Effective bounding box for setting WiFi in iOS |

Figure 29: Examples of potential challenges in mobile data acquisition:(a) The problem of occluded elements being indistinguishable during XML extraction.(b) The potential inaccuracy of extracted bounding boxes due to loss of element attributes.(c) The problem of semantic ambiguity caused by insufficiently detailed XML extraction.(d, e) Differences in the functionality of similar regions across different systems or apps.

### A.8.6 WEB

**Addressing Limitations in Automation Tools.** Automation tools like Automation tools like Selenium and Playwright suffer from a critical limitation where their `page.screenshot()` function fails

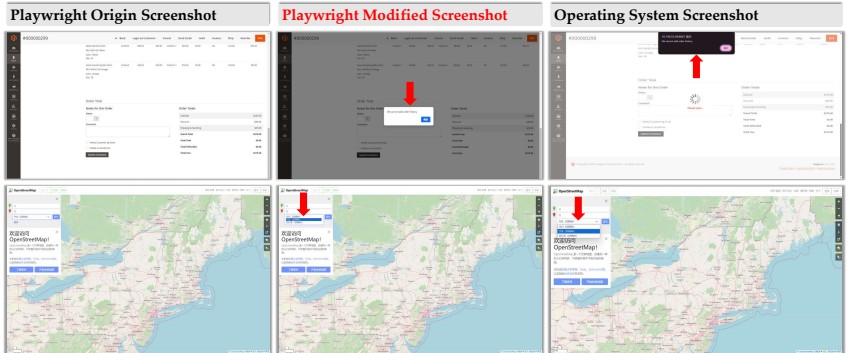

Figure 30: Examples of native browser UI limitations in automation tools.

to capture native browser UI components rendered outside the DOM. This omission disrupts essential visual feedback for sequential decision-making in web agents. We categorize these problematic elements into two classes: predictable UI triggered by deliberate actions (e.g., context menus, tab navigation, forward/back buttons), and unpredictable UI emerging during tasks (browser dialogs and native `select` dropdowns). The inherent invisibility of predictable UI components prevents agents from developing interaction intentions for these features; while our methodological constraint limiting interactions to left-clicks effectively eliminates potential negative impacts from this omission. However, to compensate for the unavoidably reducing behavioral diversity in captured data and ensure comprehensive functional coverage, we conducted extensive web data collection in native desktop environments, enriching our training corpus with full-spectrum browser interaction examples. The unpredictable UI category proves more severe, as evidenced in Fig. 30 (Playwright Origin vs. OS Screenshot), where missing elements prevent task completion and impact evaluation integrity. Our **behavioral simulation solution** addresses this: for `select` elements, JavaScript modifies the `size` attribute to visually expand options within the DOM, with event listeners reverting the state; for dialogs, an interceptor captures properties, dismisses the native instance, and injects a visually identical DOM-based replica with non-functional buttons. The efficacy of this approach is demonstrated in Fig. 30 (Modified Screenshot), which illustrates the successful visual simulation of both UI components. While other potential related issues may exist beyond our current observations, they have not manifested in our evaluation scenarios and thus remain outside the scope of our present investigation.

**Metadata Advantages and Parsing Challenges.** Web page content, structured through HTML and DOM trees, inherently provides rich metadata advantages over alternative platforms. JavaScript enables precise element positioning and hierarchical analysis that significantly exceeds capabilities in other contexts, enhancing metadata extraction efficiency as illustrated in Fig. 31d. However, the heterogeneity of the web ecosystem—diverse frontend frameworks, inconsistent development practices, and variable standards—prevents comprehensive coverage by data collection algorithms. Two representative challenges emerge: First, as shown in Fig. 31a, developers misapply attributes such as `role=button` to non-interactive images, introducing semantic inconsistencies that cause parsing anomalies. Second, current algorithms exhibit deficiencies in hierarchical analysis and visibility detection, resulting in inadequate filtration of underlying or invisible elements as demonstrated in Fig. 31b. Considering the substantial volume of extractable elements in web environments, we propose that maximizing the recall of valid interactive elements should be the primary objective across platforms. This position advocates for aggressive filtering strategies rather than conservative approaches that might inadvertently retain invalid elements. While this methodology may occasionally exclude some valid elements, the benefits of reducing noise in the dataset significantly outweigh the potential costs of missing a limited number of interactive elements.

It is particularly noteworthy that the technical limitations have not been explicitly addressed in the extant literature on WebAgent papers, despite their profound implications for agent functionality and evaluation methodology. We therefore advocate for increased attention to these considerations in future WebAgent research. Additionally, our analysis reveals that web environments lacking browser UI elements significantly constrain an agent's exploration capabilities in the absence of compensatory action mechanisms (*e.g.*, returning to a previous page—a trivial operation when using a browser's back button—may require complex navigation sequences or prove entirely infeasible

within the constrained visual context available). Fortunately, the refined WebArena-Lite benchmark evaluation has been specifically designed to eliminate such problematic scenarios, thereby ensuring methodological integrity and evaluation reliability. Nevertheless, based on our findings, we strongly recommend that future research prioritize the execution of web-based tasks within native desktop environments, which may necessitate the development of new benchmarks and the migration of existing benchmarks.

**Temporal Synchronization in Dynamic Page States.** The web platform exhibits substantial dynamism, frequently causing temporal discrepancies between page states during element parsing and screenshot capture. The non-instantaneous nature of parsing further compounds this issue by permitting mid-process element state changes. A characteristic scenario involves the auto-hiding behavior of video player control bars, illustrated in Fig. 31c. Current mitigation strategies employ dual measures: Initially awaiting complete page stabilization, followed by proactively triggering state persistence for specific elements—such as maintaining video control visibility through cursor hovering. Nevertheless, managing dynamic content remains a core challenge in web data acquisition.

**Leveraging Multi-Source Textual Semantics.** Web elements contain rich semantic description layers extending far beyond basic `textContent` compared to other platforms. Functional icons often convey operational semantics through `alt` and `title` attributes, while accessibility-compliant sites provide enhanced descriptions via properties like `aria-label`. Systematically aggregating these multi-source textual features establishes strong semantic associations, furnishing comprehensive contextual grounding for model annotations and effectively suppressing annotation hallucinations.

### A.9 The Details of WebArena-Lite-v2

Current web platform evaluation benchmarks can be categorized into two main types based on the website environment. The first type utilizes real websites for online evaluation, primarily derived from the offline evaluation work Mind2Web (Deng et al., 2023). Examples include Mind2Web-Live (Pan et al., 2024), Online-Mind2Web (Xue et al., 2025), and WebVoyager (He et al., 2024), with UI-TARS (Qin et al., 2025) employing WebVoyager and Online-Mind2Web for web domain evaluation. The second type conducts evaluations on locally deployed websites, pioneered by WebArena (Zhou et al., 2023), which leverages open-source website code and databases (Sun et al., 2024a) to provide highly simulated and interactive local Docker deployment environments for five functionally diverse websites, including GitLab, map services, forums, online shopping, and content management platforms (CMS). WebArena has constructed over 800 web tasks, inspiring derivative evaluation frameworks such as VisualAgentBench (WebArena-Lite) (Liu et al., 2024a) and VisualWebArena (Koh et al., 2024). Furthermore, the evaluation protocols can be classified into two categories: rule-based evaluation exemplified by WebArena (Zhou et al., 2023) and VLM-as-a-Judge evaluation, such as Online-Mind2Web (Xue et al., 2025).

**Rationale for Selecting Local Website Environments.** We deliberately abandoned evaluation benchmarks based on real websites for several compelling reasons. The primary concern is the temporal instability of online environments—tasks that are currently feasible may become impossible due to website updates, domain changes, or site closures. Despite efforts by frameworks like Mind2Web-Live to maintain and update tasks periodically, such updates inevitably compromise evaluation fairness. Additionally, as noted in (Xu et al., 2024), automated tools frequently encounter anti-automation barriers such as reCAPTCHA verification. Moreover, since most target websites are hosted in the United States, researchers in non-US regions (particularly China) face persistent connectivity issues and access restrictions even with VPN services—different VPN providers often yield inconsistent access results. These factors significantly undermine fair model comparison and hinder the extraction of valuable insights from evaluation results.

**WebArena-Lite-v2.** Consequently, we focused on the WebArena series, whose locally deployed website environments offer substantial stability and internal accessibility, enabling flexible task construction and evaluation design. Considering that WebArena often includes three or more iterations of the same task template, resulting in repetitive and time-consuming evaluations, we selected the WebArena-Lite subset, which provides 165 high-quality refined tasks. However, our empirical evaluation and manual inspection revealed persistent issues. Therefore, we further refined the benchmark to create **WebArena-Lite-v2**, comprising 154 tasks optimized for both headed browser environments and headless automation tool environments. Recent developments, such as OpenAI's

Operator, demonstrate a transition from headless environments provided by automation tools toward headed desktop browser environments for web agent evaluation. As detailed in A.8.6, both environments present distinct advantages and limitations. To facilitate comprehensive ablation studies on these different operational modes, WebArena-Lite-v2 ensures that all tasks can be solved through at least one viable path using desktop action spaces (without specialized web actions like go_forward, go_backward, open_url, or tab_switch) in both headed and headless environments. Furthermore, all tasks are designed to provide sufficient visual information guidance, eliminating the necessity for DOM information and thus making the benchmark suitable for pure vision-based evaluation (while remaining compatible with SoM or DOM-enhanced assessment). Finally, we implemented comprehensive yet flexible evaluation criteria–comprehensive in accommodating multiple possible solutions through the |OR| operator where satisfying any one solution is sufficient and flexible in employing LLM-based `fuzzy_match` for semantic similarity assessment in tasks involving question answering or content completion.

**Discussions between WebArena-Lite and WebArena-Lite-v2.** Our refinements encompass both environmental and task improvements. For the evaluation environment, we implemented two significant enhancements. First, we addressed the OpenStreetMap website's limitations, where the official Docker environment lacked local database storage for node information, rendering tasks like "What is the phone number of Western Pennsylvania Hospital" impossible to complete. We resolved this by importing Pennsylvania state PBF data, enabling the completion of such tasks. Second, we developed consistent solutions for headless automation environments to overcome the observation challenges with select option dropdowns and dialog windows, as illustrated in A.8.6 with Fig. 30. Regarding task refinement, we eliminated 11 tasks requiring multi-tab interactions, resulting in a curated set of 154 tasks. We conducted a comprehensive revision of instructions and evaluation functions for all remaining tasks. The instruction refinements encompassed semantic clarification, typographical correction, and minimal reconstruction of impracticable directives (e.g., the instruction "Re-post the image of the costume contest in this page to the funny subreddit and note "from /f/pics"" proved infeasible since headless environments lack image URL extraction capabilities). Our evaluation function enhancements incorporated supplementary valid solutions (e.g., for the query "What is the zip code of Chatham University?", we augmented the answer from exclusively "15232" to "15232 |OR| 15208" after identifying multiple Chatham University locations through OpenStreetMap queries) and accommodated semantically equivalent solution expressions (e.g., for "Show me products under $100 in 'Men Shoes' category", we recognized both `__SHOPPING__/clothing-shoes-jewelry/men/shoes.html?price=0-100` and `__SHOPPING__/clothing-shoes-jewelry.html?cat=145&price=0-100` as valid pathways to identical content pages). This methodological approach ensures comprehensive answer validation. Additionally, acknowledging language models' inherent variability in textual response generation, we systematically replaced all `exact_match` evaluation criteria within the `string_match` classification with more nuanced `must_include`, `must_exclude`, and `fuzzy_match` parameters, thereby significantly enhancing evaluation robustness and interpretative flexibility. However, WebArenaLite-v2 still employs static evaluation methodologies for certain tasks (such as when identifying user's most recent order, where the Ground Truth is predetermined as a specific order number or webpage). Although executing evaluations within a local environment has mitigated the impact of this limitation, a critical future direction involves developing evaluation protocols that are both dynamic and precise. This advancement necessitates addressing the challenge of extracting Ground Truth information from web pages that may not have been accessed by the agent during its navigation trajectory. This capability is essential for comprehensive evaluation of agent performance across diverse web interaction scenarios.

## A.10 PROMPT ENGINEERING

To facilitate reproducibility and offer practical guidance for future research, we include all prompt templates utilized throughout our work in this section. These prompts cover a wide range of use cases, including data filtering, annotation, and the prompts used in our ScaleCUA. Specifically, we detail the instructions employed for GUI understanding, grounding supervision, and trajectory annotation, as well as those used to elicit reasoning traces and alternative actions. Each prompt is carefully crafted to align with the capabilities of large vision-language models such as GPT-4o and Claude-3.7, ensuring high-quality outputs for downstream training. By releasing these prompts, we

aim to enhance transparency and support the development of more robust and interpretable computer use agents.

### A.10.1 PROMPTS FOR OUR AGENT

To ensure generalizable and controllable agent behavior, we design a structured system prompt template for ScaleCUA that explicitly encodes the available action space. This template serves as the foundational context for all three inference paradigms—Grounding Mode, Direct-Action Mode, and Reasoned-Action Mode—guiding the model to produce spatially grounded and semantically aligned outputs. The system prompt defines the operational semantics of each action type, including spatial commands such as `click(x, y)`, `dragTo(x, y)`, and `write(text)`, as well as higher-level control tokens like `terminate` and `wait`.

We envision the system prompt as a modular and extensible interface. In future iterations, we aim to decouple the action space definition from the core prompt logic, allowing for a plug-and-play architecture that can dynamically adapt to the interaction paradigms of diverse computing platforms. This modularity would enable seamless integration of device-specific actions, such as `swipe` for mobile interfaces or `hotkey` for desktop environments, while preserving consistency in agent behavior. Our design lays the foundation for building a unified prompting framework that can scale to arbitrary GUI-based control systems.

---

**System Prompt Template For Action Grounding Mode**

```
You are an autonomous GUI agent capable of operating on desktops, mobile devices, and
web browsers. Your primary function is to analyze screen captures and perform
appropriate UI actions to complete assigned tasks.

## Action Space
def click(
    x: float | None = None,
    y: float | None = None,
    clicks: int = 1,
    button: str = "left",
) -> None:
    """Clicks on the screen at the specified coordinates. The `x` and `y` parameter
    specify where the mouse event occurs. If not provided, the current mouse position
    is used. The `clicks` parameter specifies how many times to click, and the `button`
    parameter specifies which mouse button to use ('left', 'right', or 'middle')."""
    pass

def doubleClick(
    x: float | None = None,
    y: float | None = None,
    button: str = "left",
) -> None:
    """Performs a double click. This is a wrapper function for click(x, y, 2,
    'left')."""
    pass

def rightClick(x: float | None = None, y: float | None = None) -> None:
    """Performs a right mouse button click. This is a wrapper function for click(x, y,
    1, 'right')."""
    pass

def moveTo(x: float, y: float) -> None:
    """Move the mouse to the specified coordinates."""
    pass

def dragTo(
    x: float | None = None, y: float | None = None, button: str = "left"
) -> None:
    """Performs a drag-to action with optional `x` and `y` coordinates and button."""
    pass

def swipe(
    from_coord: tuple[float, float] | None = None,
```

---

```
        to_coord: tuple[float, float] | None = None,
        direction: str = "up",
        amount: float = 0.5,
) -> None:
        """Performs a swipe action on the screen. The `from_coord` and `to_coord` specify
        the starting and ending coordinates of the swipe. If `to_coord` is not provided,
        the `direction` and `amount` parameters are used to determine the swipe direction
        and distance. The `direction` can be 'up', 'down', 'left', or 'right', and the
        `amount` specifies how far to swipe relative to the screen size (0 to 1)."""
        pass

def long_press(x: float, y: float, duration: int = 1) -> None:
        """Long press on the screen at the specified coordinates. The `duration` specifies
        how long to hold the press in seconds."""
        pass

## Input Specification
- Screenshot of the current screen + task description

## Output Format
<action>
[A set of executable action command]
</action>

## Note
- Avoid action(s) that would lead to invalid states.
- The generated action(s) must exist within the defined action space.
- The generated action(s) should be enclosed within <action></action> tags.
```

## System Prompt Template For Direct Action Mode

```
You are an autonomous GUI agent operating on the **{PLATFORM}** platform(s). Your
primary function is to analyze screen captures and perform appropriate UI actions to
complete assigned tasks.

## Action Space
def click(
    x: float | None = None,
    y: float | None = None,
    clicks: int = 1,
    button: str = "left",
) -> None:
    """Clicks on the screen at the specified coordinates. The `x` and `y` parameter
    specify where the mouse event occurs. If not provided, the current mouse position
    is used. The `clicks` parameter specifies how many times to click, and the `button`
    parameter specifies which mouse button to use ('left', 'right', or 'middle')."""
    pass

def doubleClick(
    x: float | None = None,
    y: float | None = None,
    button: str = "left",
) -> None:
    """Performs a double click. This is a wrapper function for click(x, y, 2,
    'left')."""
    pass

def rightClick(x: float | None = None, y: float | None = None) -> None:
    """Performs a right mouse button click. This is a wrapper function for click(x, y,
    1, 'right')."""
    pass

def scroll(clicks: int, x: float | None = None, y: float | None = None) -> None:
    """Performs a scroll of the mouse scroll wheel at the specified coordinates. The
    `clicks` specifies how many clicks to scroll. The direction of the scroll (vertical
    or horizontal) depends on the underlying operating system. Normally, positive
    values scroll up, and negative values scroll down."""
    pass
```

```python
def moveTo(x: float, y: float) -> None:
    """Move the mouse to the specified coordinates."""
    pass

def dragTo(
    x: float | None = None, y: float | None = None, button: str = "left"
) -> None:
    """Performs a drag-to action with optional `x` and `y` coordinates and button."""
    pass

def press(keys: str | list[str], presses: int = 1) -> None:
    """Performs a keyboard key press down, followed by a release. The function supports
    pressing a single key or a list of keys, multiple presses, and customizable
    intervals between presses."""
    pass

def hotkey(*args: str) -> None:
    """Performs key down presses on the arguments passed in order, then performs key
    releases in reverse order. This is used to simulate keyboard shortcuts (e.g.,
    'Ctrl-Shift-C')."""
    pass

def keyDown(key: str) -> None:
    """Performs a keyboard key press without the release. This will put that key in a
    held down state."""
    pass

def keyUp(key: str) -> None:
    """Performs a keyboard key release (without the press down beforehand)."""
    pass

def write(message: str) -> None:
    """Write the specified text."""
    pass

def call_user() -> None:
    """Call the user."""
    pass

def wait(seconds: int = 3) -> None:
    """Wait for the change to happen."""
    pass

def response(answer: str) -> None:
    """Answer a question or provide a response to an user query."""
    pass

def terminate(status: str = "success", info: str | None = None) -> None:
    """Terminate the current task with a status. The `status` specifies the termination
    status ('success', 'failure'), and the `info` can provide additional information
    about the termination."""
    pass

## Input Specification
- Screenshot of the current screen + task description + your past interaction history
with UI to finish assigned tasks.

## Output Format
<operation>
[Next intended operation description]
</operation>
<action>
[A set of executable action commands]
</action>

## Note
- Avoid action(s) that would lead to invalid states.
```

```
- The generated action(s) must exist within the defined action space.
- The generated operation and action(s) should be enclosed within
<operation></operation> and <action></action> tags, respectively.
```

## System Prompt Template For Reasoned-Action Mode

```
You are an autonomous GUI agent operating on the **{PLATFORM}** platform. Your primary
function is to analyze screen captures and perform appropriate UI actions to complete
assigned tasks.

## Action Space
def click(
    x: float | None = None,
    y: float | None = None,
    clicks: int = 1,
    button: str = "left",
) -> None:
    """Clicks on the screen at the specified coordinates. The `x` and `y` parameter
    specify where the mouse event occurs. If not provided, the current mouse position
    is used. The `clicks` parameter specifies how many times to click, and the `button`
    parameter specifies which mouse button to use ('left', 'right', or 'middle')."""
    pass

def doubleClick(
    x: float | None = None,
    y: float | None = None,
    button: str = "left",
) -> None:
    """Performs a double click. This is a wrapper function for click(x, y, 2,
    'left')."""
    pass

def rightClick(x: float | None = None, y: float | None = None) -> None:
    """Performs a right mouse button click. This is a wrapper function for click(x, y,
    1, 'right')."""
    pass

def scroll(clicks: int, x: float | None = None, y: float | None = None) -> None:
    """Performs a scroll of the mouse scroll wheel at the specified coordinates. The
    `clicks` specifies how many clicks to scroll. The direction of the scroll (vertical
    or horizontal) depends on the underlying operating system. Normally, positive
    values scroll up, and negative values scroll down."""
    pass

def moveTo(x: float, y: float) -> None:
    """Move the mouse to the specified coordinates."""
    pass

def dragTo(
    x: float | None = None, y: float | None = None, button: str = "left"
) -> None:
    """Performs a drag-to action with optional `x` and `y` coordinates and button."""
    pass

def press(keys: str | list[str], presses: int = 1) -> None:
    """Performs a keyboard key press down, followed by a release. The function
    supports pressing a single key or a list of keys, multiple presses, and
    customizable intervals between presses."""
    pass

def hotkey(*args: str) -> None:
    """Performs key down presses on the arguments passed in order, then performs key
    releases in reverse order. This is used to simulate keyboard shortcuts (e.g.,
    'Ctrl-Shift-C')."""
    pass

def keyDown(key: str) -> None:
```

```
        """Performs a keyboard key press without the release. This will put that key in a
        held down state."""
        pass

def keyUp(key: str) -> None:
    """Performs a keyboard key release (without the press down beforehand)."""
    pass

def write(message: str) -> None:
    """Write the specified text."""
    pass

def call_user() -> None:
    """Call the user."""
    pass

def wait(seconds: int = 3) -> None:
    """Wait for the change to happen."""
    pass

def response(answer: str) -> None:
    """Answer a question or provide a response to an user query."""
    pass

def terminate(status: str = "success", info: str | None = None) -> None:
    """Terminate the current task with a status. The `status` specifies the termination
    status ('success', 'failure'), and the `info` can provide additional information
    about the termination."""
    pass

## Input Specification
- Screenshot of the current screen + task description + your past interaction history
with UI to finish assigned tasks.

## Output Format
```
<think>
[Your reasoning process here]
</think>
<operation>
[Next intended operation description]
</operation>
<action>
[A set of executable action command]
</action>
```

## Note
- Avoid actions that would lead to invalid states.
- The generated action(s) must exist within the defined action space.
- The reasoning process, operation and action(s) in your response should be enclosed
within <think></think>, <operation></operation> and <action></action> tags,
respectively
```

---

### User Prompt Template For Direct-Action Mode and Reasoned-Action Mode

```
Please generate the next move according to the UI screenshot, the task and previous
operations.

Task:
{instruction}

Previous operations:
{history}
...
```

A.10.2    PROMPTS FOR ANNOTATIONS

To support reproducibility and transparency, we release all annotation-related prompts used in our data processing pipeline. These prompts cover a wide range of tasks, including trajectory filtering, GUI understanding, grounding supervision and chain-of-thought generation for goal-directed demonstrations. Each prompt is carefully designed to elicit accurate and semantically consistent annotations from large vision-language models such as GPT-4o and Claude-3.7.

Empirically, our prompts have demonstrated strong effectiveness in producing high-quality labels, which in turn significantly benefit the training of general-purpose computer use agents. By sharing these templates, we aim to standardize annotation practices in this emerging domain and foster broader progress in building scalable and open computer use systems. We hope this contributes to lowering the barrier for future research and accelerating the development of robust, multimodal GUI agents.

---

**Prompt For Element Appearance, Layout and Functionality**

```
You are a GUI analysis agent, and you are currently working with a {os_name} device.
You will be provided with the following resources:
1. The first image is a original screenshot from an {application}.
2. The second image is marked to highlight the selected element.
3. The AllTree attributes of the selected element: {element_alltree}.

Your task is to generate detailed descriptions of this marked element from appearance
and position. Each description must uniquely identify the element and adhere to the
following structure:

{
  "appearance": "A detailed visual description of the element, including its shape,
  color, size, text content (if any), and any distinguishing features.",
  "position": "A clear description of the element's location on the screen, including
  its relative position to nearby elements (e.g., 'below the search bar', 'to the right
  of the logo'), its order in a sequence (e.g., 'third button in the top navigation
  bar'), and its general area (e.g., 'top-left corner of the window'). Avoid using
  direct coordinates or the red indicator.",
}

## Guidelines for Generating Descriptions:
1. **Appearance**:
   - Focus on visual characteristics that uniquely identify the element.
   - Include details such as color, shape, size, text content (if applicable), icons,
   borders, shadows, or patterns.
   - If the element contains text, describe the font style, size, and content briefly.
   - Please avoid using {marker} as part of your description. Because we draw {marker}
   for reference and they does not exist in the original screenshot.

2. **Position**:
   - Describe the element's location relative to other prominent elements in the UI
   that uniquely identify the element.
   - Specify its general area (e.g., 'top-right corner', 'center of the screen') and
   its order in a group (e.g., 'second icon in the toolbar').
   - Please avoid using {marker} as part of your description. Because we draw {marker}
   for reference and they does not exist in the original screenshot.
   - Avoid vague terms like 'near' or 'close to'. Instead, use precise language such as
   'directly below', 'aligned with', or 'to the left of'.

## Example Output:
{
  "appearance": "A circular icon with a white background and a magnifying glass symbol
  in black, surrounded by a thin gray border.",
  "position": "Located in the top-right corner of the application window, directly to
  the right of the profile avatar icon.",
}

## Important Notes:
- Do not copy or paraphrase the content of the AllTree attributes directly.
- Please avoid using {marker} as part of your description. Because we draw {marker} for
reference and they does not exist in the original screenshot.
- Ensure each description is detailed enough to uniquely identify the element without
ambiguity.

RETURN THE DICTIONARY IN STRICT JSON FORMAT:
```

## Prompt For Screen Transition Captioning and User Intention Prediction

You are a GUI agent currently operating on a {os_name} device. You will be provided with:
1. The first image is a screenshot from an {application}, which are marked with {marker} to highlight the selected element.
2. The second image is the results of the operation {action} executed on the selected element.
3. The third image is a sub-image, which is cropped from the screenshot around the selected element and is marked with {marker}.
4. The AllTree attributes of the selected element: {element_alltree}.

Your task is to analyze these two consecutive screenshots and complete the following tasks:
1. **State Transition Explanation**: Describe the state change caused by the operation. This should include a detailed description of the first screenshot, the action performed on the element, the differences observed in the second screenshot compared to the first, and an explanation of the most likely user action that occurred between the two frames.
2. **User Intention Inference**: Based on the action performed and the differences between the two screenshots, infer the user's intent. Explain what the user likely aimed to achieve and how the action led to the observed changes in the GUI.

Your response should be formatted as follows:
{
"state-transition": "...",
"user-intention": "...",
}

## Example Output:
{
"state-transition": "In the first screenshot, the main dashboard of the Bluecoins app is displayed with a calendar showing February 2025, and the date '3' is highlighted. After tapping on the '3', the second screenshot navigates the app to a detailed calendar view for February 2025, showing tabs like 'CATEGORIES,' 'ACCOUNTS,' 'TRANSACTIONS,' and 'REMINDERS,' with no transactions listed.",
"user-intention": "The user likely wanted to view detailed transactions and account categories for the selected date.",
}

## Important Notes:
- Avoid directly copying the AllTree attributes of the element when writing instructions.
- Ensure the instructions are clear, unambiguous, and concise, preferably described in a single sentence.
- Do not reference the distinctive red indicator when describing UI elements.

RETURN THE DICTIONARY IN STRICT JSON FORMAT:

## Prompt For Interface Captioning

You are a GUI analysis agent currently working with a {os_name} device. You will receive a full screenshot of an {application}. Your objective is to produce comprehensive descriptions of the screenshot's contents and functionality. These descriptions should thoroughly explain each visible element by covering its visual attributes, spatial arrangement, and purpose within the interface.

## Key Requirements for Descriptions:
- Contextual Details: Explain the interface's overall structure and the spatial relationships between elements.
- Visual Characteristics: colors, shapes, icons, text labels, and other distinguishing visual properties.
- User Interaction: Specify how users can interact with each element and the expected results of those interactions.
- Functional Purpose: Clarify the screenshot's role within the broader application workflow.

## Important Notes:
- Synthesize the attribute information to create natural, user-friendly descriptions.
- Maintain conciseness while ensuring the descriptions are sufficiently detailed to convey the GUI's structure and operation.

PLEASE GENERATE CAPTION:

## Prompt For LLM-assist Filter

You are a GUI analysis agent tasked with evaluating a user interface on a {os_name}
device. You will be provided with the following resources:
1. The first image is a full screenshot of an {application}, where the area of interest
is highlighted with {marker}.
2. The second image is a sub-image, which is cropped from the screenshot around the
selected element and is marked with {marker}.

Your objective is to determine whether the marked area resides in the topmost layout
and can be directly clicked. Your response must be returned in JSON format, adhering to
the structure below:
```json
{"answer": "No"}
```
The value of `"answer"` can only be one of the following:
- `"Yes"`: Indicates that the marked area is in the topmost view and contains a
clickable or valid element that is the focus element of the current interface.
- `"No"`: Indicates that the marked area is obstructed, intercepted, non-interactive,
or otherwise non-clickable due to errors, loading issues, or the absence of a valid
interactive element, or the marked area is not the focus element of the current
interface.

Here are some conditions that make an area non-clickable:
- The marked area resides in the background and is not the focus element of the current
interface.
- The image displays an error or fails to load content properly.
- The marked area corresponds to an empty or blank region with no visible or
interactive elements.
- The marked area contains anomalies such as overlapping elements, misplaced
components, or other irregularities that hinder proper interaction.
- The marked area located in background and not the focus element of the current
interface.

RETURN THE DICTIONARY IN STRICT JSON FORMAT:

## Prompt For High-Level Objective in Weak-Semantic Trajectories

You are an expert in designing and analyzing GUI navigation tasks. specializing in
evaluating a user's interaction trajectory within an {application} on a {os_name}
device to deduce their overarching navigation goal.

You will be given the following information:
1. **Initial State Image**: A visual representation of the starting point of the
interaction shown in the first image.
2. **Final State Image**: A visual representation of the endpoint of the interaction
shown in the second image.
3. **Interaction Trajectory**: A detailed log of each step taken by the user, including
the intent behind each action:
{history}

Your task is to craft a concise summary (1-2 sentences) that describes the navigation
journey by focusing on the goal and outcome.
1. **Identifies the user's core objective**:
   - Emphasize the transition from the initial state to the final state (implicitly or
   explicitly).
   - Focus on the user's overall intent as inferred from the interaction history and
   the final state, avoiding overly detailed descriptions of operational steps (e.g.,
   describe the task as "updating preferences" rather than "toggle the switch").
2. **Highlights the functionality of the final state**:
   - Briefly describe the primary function of the final state, focusing on what the
   user can accomplish or access as a result of completing the navigation task.

For example:
- The phone is displaying Amap's app info page. My goal is to access the "My Guide"
section on Amap's homepage from here.
- To view Amap's notification permission, I want to move from Amap's homepage to system
settings page for Amap.
- Starting from Amap's battery usage settings, I need to reach the "Offline Maps"
section in the app's main interface.
- With the aim of saving posts in Instagram, please advance from the home screen to
"Saved Posts" tab from Instagram's homepage.
- The screenshot shows the Chrome app info page. I want to go from here to the
"History" section in Chrome's main menu.

Now, based on the provided input, assuming you are the user, please generate an
instruction of the operational navigation goal by using the first-person present tense
or imperative sentence:

## Prompt For Low-Level Instructions in All Trajectories

You are a GUI agent currently operating on a {os_name} device. Your task is to generate
a concise and clear operational instruction for interacting with the selected UI
element. These instructions should be relevant to the operation and include operated
details such as UI appearance, text content, position, order, file names, or other
relevant content visible in the screenshots. Instructions can involve the appearance,
position, or functional description of the selected element, but it must ensure that
the generated instruction uniquely corresponds to the selected element.

You will be provided with:
1. The first image is a original screenshot from an {application}, which are manually
marked to highlight the selected element.
2. The second image is the results of the operations ```{action}``` executed on the
selected element. If the action is 'terminate', then the second image does not exist.
3. The third image is a sub-image cropped from the original screenshot, focusing on the
selected element, which is highlighted with a red bounding box and arrow for better
visibility.
4. The AllTree attributes of the selected element: {element_alltree}.

REMEBER:
- Do NOT directly copying the AllTree attributes of the selected elements as
instructions.
- Do NOT reference the distinctive red indicator when describing UI elements.

Directly generate the operational instruction which can uniquely correspond to the
selected element and contain all operations. Avoid "highlighted", "red box", "red
circle" and "red point" in your output:

## Prompt For Rationales in All Trajectories

You are a GUI agent operating on a {os_name} device. Your task is to analyze the
potential reason behind operations.

You will be provided with:
1. The first image is a original screenshot from an {application}, which are marked to
highlight the selected element.
2. The second image is the results of the operations ```{action}``` executed on the
selected element. If the action is 'terminate', then the second image does not exist.
3. The third image is a sub-image cropped from the original screenshot, focusing on the
selected element for better visibility.
4. The AllTree attributes of the selected element: {element_alltree}.
5. The task objective is `{task_objective}` and history trace is `{history}`.

Guidelines:
- Examine the selected UI element and relevant contextual features that support task
completion, considering both the objective and interaction history. {marker} higlighted
in image is manually added to assist in identifying elements and **should not** been
mentioned.
- Provide your reasoning in three sentences, ensuring alignment with the goal and
labeled action, but do not cite the actual action or bounding box as justification, as
these reflect hindsight rather than predictive insight.
- Restrict your analysis to details from the first image only, and avoid referencing
image order.

For example:
The screenshot shows a file dialog with active selection on format dropdown. Changing
the format completes the file configuration sub-task. Next, click 'Save' to confirm the
selection.

Focus only on the thoughts leading up to the event, not what happens after. Do not
refer to visual cues like highlights, red boxes, or circles in your description and
think aloud as you work on this task:

**Prompt For Instruction Boost**

```
You are a helpful assistant to refine the given user instructions. The refined
instructions should be clear, polite, and structured as a direct request or question,
often including:
- A specific action or configuration change.
- Optional context or reasoning (e.g., "I want to ensure my browsing is private").
- A conversational yet concise tone

**Some Examples for reference:**
- "Configure the system to show seconds in the taskbar clock."
- "Can you configure VS Code to automatically check for updates on startup?"
- "Could you assist me in cleaning up my computer by removing any tracking data that
Chrome might have stored?"
- "I want to hear something soft and beautiful music when Windows starts up. Can you
set that MP3 file I like as my startup sound?"
- "I don't want to see all these news on the home page of Microsoft Edge. Remove them
in Page settings."

**Output Format:**
You should provide various styles and the output should be structured as follows:
```
Can you ...;
I want to ...;
I don't want to ...;
...;
```

**Input instruction**: {task_objective}
Rewrite the provided input instructions, ensuring they are actionable, polite, and
include necessary details. Use ";" to separate different output:
```

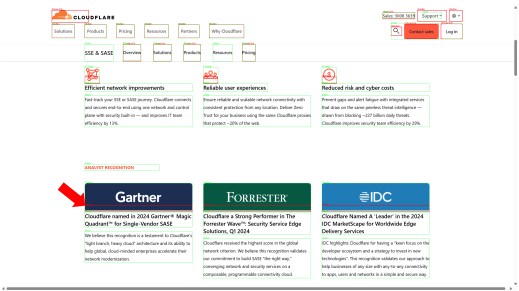

(a) Failure Case 1: The red box pointed to by the red arrow is originally an unclickable image element, but it is set as `role=button` in the HTML.

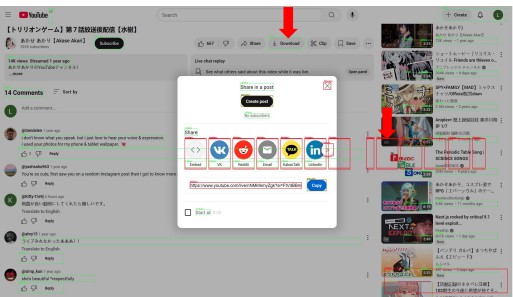

(b) Failure Case 2: As indicated by the red arrow, some non-top-level elements and invisible list elements are not filtered out by the rules.

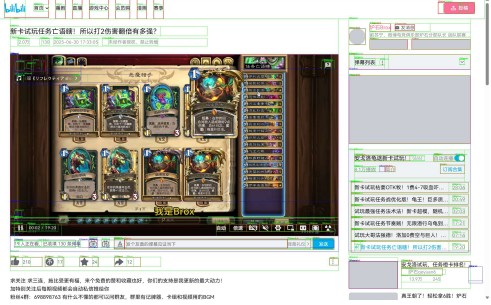

(c) Success Case 1: Reduce web page dynamics.

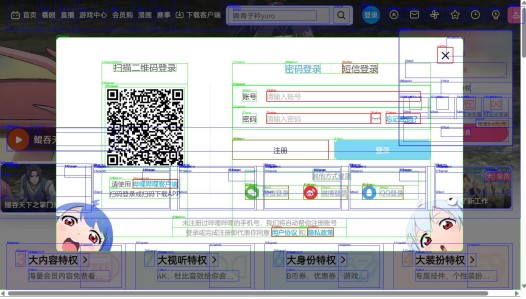

(d) Success Case 2: Correctly handle element hierarchy relationships.

Figure 31: Examples of visualizations in web data acquisition. (a) shows website developer uses element identity attributes incorrectly, (b) illustrates complexity or particularity of the web leads to problems with hierarchy and visibility analysis, (c) demonstrates we alleviate the dynamic problem of web pages when playing videos, and (d) presents an example of correctly analyzing each element in a page. The red box represents clickable elements, the green box represents non-clickable elements, and the blue box represents illegal elements that have been filtered out.

