# OpenReview forum: "ScaleCUA: Scaling Open-Source Computer Use Agents with Cross-Platform Data"
_ICLR.cc/2026/Conference — ICLR 2026 Oral_

### Official Review · Reviewer_56qp · 2025-10-31

**Soundness:** 3
**Presentation:** 3
**Contribution:** 3
**Rating:** 6
**Confidence:** 2

**Summary:**

This paper presents a dataset and agent models for computer agents. Provides a rich source of open source computer use data that is curated across various platform to fine tune the models. They show that fine tuning with this data curated across various platforms creates a strong model that performs well across various computer use benchmarks.

**Strengths:**

1. The comprehensive data set and model that the paper proposes have the potential to further the research in this field.
2. Experimental results showing the overall approach and how well it generalizes across various tasks.
3. Method presented for data curation, which can be further extended to create more data assets

**Weaknesses:**

no comment

**Questions:**

Is the fin-tuning used SFT in this case?

---

> ### Author Response · Authors · 2025-11-19
>
> ## Q1: Is the fin-tuning used SFT in this case?
> Yes, the fine-tuning process described in our paper is supervised fine-tuning (SFT). Specifically, we fine-tune our models on the curated datasets using standard supervised learning techniques, where the model learns to align its actions with the annotated data across various platforms and tasks. This allows the model to generalize effectively across different computer use benchmarks.
>
> Finally, we would like to thank the reviewer for recognition of our work without any negative feedback. We will appreciate it if you consider raising your rating. Given the contributions of our work in providing the dataset construction and computer use agent training, we believe our work offers meaningful advancements to the field.

---

### Official Review · Reviewer_TkSU · 2025-11-01

**Soundness:** 4
**Presentation:** 3
**Contribution:** 4
**Rating:** 10
**Confidence:** 3

**Summary:**

This paper introduces a cross-platform computer use dataset, collected via an interactive data pipeline that
integrates automated agents with human experts. They also developed ScaleCUA, a family of base agent models achieving reported SOTA on several GUI-oriented benchmarks.

**Strengths:**

- The model achieves impressive results on several challenging benchmarks, including SOTA performance on MMBench-GUI L1-Hard (94.4%).
- The paper presents a comprehensive suite of ablation studies exploring design choices, including data augmentation, trajectory weighting, coordinate formats, resolution impact, inference methods, and data scaling.
- The dataset spans six OSs and multiple GUI domains, broader than previous open datasets (e.g., OSWorld, JEDI, WebArena). Openly releasing all data, models, and code is a significant contribution, particularly within this research area, where datasets are often proprietary.

**Weaknesses:**

No major weaknesses, some suggestions:
- The paper contains several minor typos and grammatical errors (e.g., "expensively to annotate," capitalization issues) and inconsistencies in terminology.
- They can add an analysis of the data quality, comparing the current approach and manual collection.

**Questions:**

How does the model perform when faced with minor software changes (e.g., UI updates, version differences)? Is there a mechanism to adapt to these types of alterations without requiring retraining?

---

> ### Author Response · Authors · 2025-11-19
>
> ## Q1: Typo Issues.
> Thank you for your careful check. We fix typos in the resubmitted version of the paper.
>
> ## Q2: More Analyses on Data.
> In our original paper, we have provided thorough ablations on our data, eg., Fig. 1 and Table 4 in the Appendix (original paper). In the latest version of the paper, we have moved data ablations from the appendix to the main text.
>
> In fact, it is challenging to make a fair comparison of the quality across different datasets. Because it may involve various confounding factors, such as dataset size, domain coverage, and data collection time. Therefore, we use the experiments mentioned by other reviewer to demonstrate the quality of our data.  We perform this experiment using Qwen2.5VL-3B under a fixed data volume setting to ensure a fair comparison. Specifically, we selected three representative grounding datasets: (1) OS-Atlas [1], which contains large-scale GUI grounding data, but we found it exhibits a notable amount of annotation noise. The reported results are based on its raw data, while our own training in the paper used a re-filtered subset to mitigate label errors. (2) JEDI [2], a synthetic dataset of high internal consistency but limited realism due to its synthetic rendering process and the resulting domain gap from real GUI environments. (3) Aguvis [3], which integrates and cleans several widely used public datasets with high average quality and balanced coverage. For all datasets, we fixed the number of samples to min(JEDI, OS-Atlas, Aguvis, ours) based on the actual number of elements used in our training pipeline, rather than their raw dataset size. Here, we use the 550K elements that are the size of JEDI we employed in training our ScaleCUA.
>
> | Dataset    | SS v2 | SS-Pro |
> |------------|-------|--------|
> | JEDI       |74.4  | 27.9   |
> | OS-Atlas   |76.0  | 22.6   |
> | Aguvis     |81.2  | 30.7   |
> | ours       |83.6  | 38.1   |
>
> As shown, our dataset still achieves superior performance gains under the same data budget. In our view, the value of a dataset lies not only in absolute improvement on performance but also in complementarity. Even if different datasets have varying biases, a new dataset that introduces distinct coverage and diversity can make a tangible positive contribution to advancing GUI agent research.
>
> If you still have concerns on our data abaltions, we are glad to futher discuss with you.
>
>
> ## Q3: Potential solutions when software updates.
> This is indeed an important yet interesting topic, and we have also reflected on it during our data collection. At present, ScaleCUA does not include an explicit online adaptation mechanism for handling software version updates. In practice, when the software update occurs, model performance typically degrades slightly, in both grounding and planning. For example, if the target UI element is identified based on its textual information, changes in its style have minimal impact on the agent's performance. However, if the icon is redesigned, it may lead to a significant performance drop for the agent. **This limitation is shared by most native computer use agents that rely purely on visual perception.**
>
> We see two practical directions to address this challenge without requiring full retraining.
> 1) Offline knowledge database, which constructs a large-scale tutorial and documentation corpus for professional software. Many enterprise and scientific tools provide manuals, official docs, or papers. These resources can be utilized by retrieval-augmented generation (RAG) at inference time to guide agents operating on unseen applications.
> 2) Online retrieval, which adopts a web search plus RAG workflow. The web contains abundant usage knowledge for software; when coupled with robust long-context understanding and faithful instruction following, retrieval can supply missing domain knowledge on the fly.
> 3) Autonomous self-evolution, which enables the agent to actively explore the unseen environment through trial-and-error. By interacting with the interface and observing feedback (e.g., GUI state changes or error logs), the agent can generate its own offline knowledge database and reflect on execution failures. This iterative process allows the agent to progressively accumulate procedural experience and adapt its policy to the specific logic of the new application, mirroring how humans learn software by "playing" with it.
>
> We hope these insights clarify the limitations of current native agents and outline promising solutions for improving adaptability under interface evolution.

---

### Official Review · Reviewer_ZN1J · 2025-11-04

**Soundness:** 4
**Presentation:** 3
**Contribution:** 3
**Rating:** 6
**Confidence:** 5

**Summary:**

This paper introduces ScaleCUA, a large-scale dataset with multiple OS and tasks supported and associated agents trained on the dataset. It begins by incorporating a dual-loop data collection pipeline consisting of agent-environment interaction loop and agent-human hybrid data acquisition loop. The authors then train Qwen2.5-VL on the collected dataset with three modes: grounding mode, direct action mode and reasoned action mode. Experiments are conducted on multiple datasets, including screenspot-pro, webarena-lite-v2, mmbench-gui, os-world-g etc. These evaluations cover multiple tasks and demonstrate a comprehensive & promising results for ScaleCUA. ScaleCUA shows the potential to scale up CUA performance with cross-platform data collected from automated agent data collection and human-in-the-loop data collection together.

**Strengths:**

1. ScaleCUA contributes a large-scale dataset to the GUI Agent community, with 15k weak-semantic trajectories from automated agent data collection and 4k expert-curated trajectories from human-in-the-loop data collection. Notably, the collected data cover 6 OS, including desktop and mobile ones.
2. ScaleCUA contributes several interesting key insights to the community backed with supporting experiment results, including the finding that models trained with raw coordinates perform better than those trained with normalized coordinates, especially in cross-platform scenarios; reasoned action mode is  always better than direct action mode, the multi-format GUI grounding corpus, etc.
3. This paper is well written, clearly structured, and provides comprehensive details and ablation studies in the appendix for future research.

**Weaknesses:**

1. For the dual loop, currently it includes two components: agent-environment interaction loop and agent-human hybrid data acquisition loop, and these two components both contribute to the final data collection of ScaleCUA, with different properties, i.e., data collected from agent automated pipeline are of large scale but weak semantics, data collected from human-in-the-loop pipeline are of high quality but high cost. However, ScaleCUA did not comprehensively integrate these two components so that they can benefit each other. This is not a major drawback, but a point that the authors can improve further. The proposed dual loop is promising, but the current implementation of simply merging the two data sources is naive, which has been discussed in prior work。
2. The dataset distribution is extreme, e.g., MacOS data takes only 0.6% of the total data, and grounding takes 97.6% of the total Windows data. As a cross-platform dataset with multi-task support, this distribution is not reasonable enough.

**Questions:**

1. for Figure 5(a), I observed that the grounding accuracy keeps almost unchanged or dropping when the resolution is increased from 720p to 4k in screenspot-v2, also the grounding accuracy saturates at 1080p in osworld-g. Do these results demonstrate that "Grounding accuracy rises steadily from 720p to 2K, with diminishing returns at 4K." is not a universal finding, but only occuring in screenspot-pro benchmark?
2. It can be observed that scaling data has marginal effects on improving agent performance on WindowsAgentArena benchmark,  does this implicate that increasing train data amount cannot benefit navigation tasks? Can you also share the performance on OSWorld with different training data ratio?

---

> ### Author Response · Authors · 2025-11-19
>
> ## Q1: The significance of data collection pipeline
> Previous works rely either on automatic data collection[1,2], purely manual collection[3,4,5] or data synthesis [6]. In contrast, our current work ingeniously combines both paradigms to design a human-agent hybrid data collection framework, ensuring both data scaling and quality.
>
> **Clarification on practical significance.** Although our collection framework is conceptually straightforward, executing it across heterogeneous operating systems and software ecosystems entails substantial non-trivial engineering. Each platform introduces distinct GUI structures, metadata formats, and failure patterns. We systematically addressed numerous corner cases, such as **UI element occlusions**, **wrong/absent accessibility attributes**, **repeated exploration**, and **annotation hallucinations**, which would otherwise interrupt large-scale acquisition. *These practical issues, though normally overlooked in prior work, motivate us to design comprehensive heuristic algorithm for agent-driven data collection, which are crucial for scaling computer-use data collection from multiple platforms*. In addition, we document these corner cases with solutions in detail in Appendix A.7 & A.9 (original paper) as a reproducible guide for the community. As a result, our contribution extends beyond the design of the pipeline itself, but it also lies in our detailed documentation and analysis of the classic corner cases encountered throughout the data construction process. Despite involving many engineering-oriented efforts, we still aim to share these experiences, expecting they can provide valuable guidance for future developments in this field.
>
> Thank you for your kind suggestion. We will continue to improve our data pipeline, enhancing both data collection efficiency and quality, while contributing to the open-source community.
>
> [1] Sun Q, Cheng K, Ding Z, et al. Os-genesis: Automating gui agent trajectory construction via reverse task synthesis[C]//Proceedings of the 63rd Annual Meeting of the Association for Computational Linguistics (Volume 1: Long Papers). 2025: 5555-5579.
>
> [2] Wu Z, Wu Z, Xu F, et al. Os-atlas: A foundation action model for generalist gui agents[J]. arXiv preprint arXiv:2410.23218, 2024.
>
> [3] Rawles C, Li A, Rodriguez D, et al. Androidinthewild: A large-scale dataset for android device control[J]. Advances in Neural Information Processing Systems, 2023, 36: 59708-59728.
>
> [4] Zhang J, Wu J, Yihua T, et al. Android in the zoo: Chain-of-action-thought for gui agents[C]//Findings of the Association for Computational Linguistics: EMNLP 2024. 2024: 12016-12031.
>
> [5] Deng X, Gu Y, Zheng B, et al. Mind2web: Towards a generalist agent for the web[J]. Advances in Neural Information Processing Systems, 2023, 36: 28091-28114.
>
> [6] Xie T, Deng J, Li X, et al. Scaling Computer-Use Grounding via User Interface Decomposition and Synthesis[J]. arXiv preprint arXiv:2505.13227, 2025.
>
> ## Q2: Dataset distribution is extreme.
> First, this imbalance reflects the inherent heterogeneity of real-world GUI ecosystems. Windows and web environments dominate real software usage and provide higher accessibility for large-scale automated exploration. In contrast, macOS and iOS platforms impose stronger sandboxing and security restrictions, which limit both automated interaction and metadata extraction. Even with significant engineering effort, macOS accounts for only 0.6% of the total because interaction through accessibility APIs and screen capture is tightly constrained.
>
> Second, the difficulty of data acquisition also varies across tasks. For example, A single screenshot can contain multiple grounding annotations, whereas constructing a complete trajectory requires multiple screenshots. Grounding samples serve as the foundation for reasoning and task completion. Once a stable grounding capability is established, we can progressively increase the ratio of high-level understanding and trajectory data.
>
> To the best of our knowledge, we are the first to explore the data collection on macOS and iOS platforms. As a result, we spent considerable time addressing platform-specific engineering challenges, which led to delays in data collection for both platforms. Fortunately, the current results indicate that these efforts have begun to pay off. We regard data collection as a long-term endeavor. In future iterations, we plan to expand macOS and iOS coverage. Our goal is to achieve a more balanced dataset while maintaining realistic accessibility and diversity. Thanks for your valuable suggestion.

---

> > ### Author Response · Authors · 2025-11-19
> >
> > ## Q3: Different input resolutions on grounding benchmarks.
> > We acknowledge that "Grounding accuracy rises steadily from 720p to 2K, with diminishing returns at 4K." holds in screenspot-pro. We give the detailed clarification as follows.
> >
> > First, Qwen2.5-VL controls the effective image resolution through two parameters, `min_pixels` and `max_pixels`, which determine the valid range of input size for the image processor. When the original screenshot resolution falls within this range, no resizing is applied. If it is outside, the image is resized to fit within the limits before being fed into the model.
> >
> > OSWorld-G uses strictly standardized 1080p frames. When the input resolution (ie., `max_pixels`) is set at or above 1080p, the performance saturates because the inputs still match the maximum training resolution, \ie, 1080p. In ScreenSpot-v2, the majority of screenshots are at or below 1080p, yet this results in negative impacts when increasing resolution further. By contrast, ScreenSpot-Pro contains a large proportion of native 4K screenshots. The performance on it benefits from higher resolutions up to 2K but exhibits a drop at 4K. Overall, we can observe that the impact of resolution on grounding performance largely depends on the data distribution of the benchmark.
> >
> > We update this experimental analysis in the resubmitted paper.
> >
> > ## Q4: Clarification on performance when scaling data.
> > We believe that scaling training data does improve an agent’s navigation performance, though the degree of improvement varies across benchmarks.
> >
> > For WindowsAgentArena, the observed gains are smaller mainly because these benchmarks are inherently more challenging. Desktop environments are more densely structured than mobile and contain a higher proportion of icons than web pages, making grounding and planning more difficult. They also involve many domain-specific software applications, whose usage patterns and visual layouts are rarely covered in the pre-training corpora of base models. Consequently, these factors lead to low baseline scores, and even small performance improvements are difficult to achieve.
> >
> > To provide additional evidence, we report below the results of Qwen2.5-VL-3B trained with different ratios of our cross-platform data on OSWorld:
> > | ratio | OSWorld |
> > |-------|---------|
> > | 0%    |   7.6   |
> > | 25%   |   10.2  |
> > | 50%   |   10.8  |
> > | 100%  |   12.4  |
> >
> > These results confirm that scaling data continues to yield positive effects on navigation, even for challenging desktop benchmarks. These results intuitively reflect the task’s difficulty, and also imply a larger data volume required to achieve the desired performance.

---

### Official Review · Reviewer_ojRP · 2025-11-05

**Soundness:** 3
**Presentation:** 3
**Contribution:** 3
**Rating:** 6
**Confidence:** 1

**Summary:**

The paper tackles the challenge of building computer use agents (CUAs)—AI systems that can autonomously operate GUIs across desktop, mobile, and web platforms. Current progress is limited by the scarcity of high-quality, open computer-use data and the reliance on closed-source models.

To address this, the authors introduce ScaleCUA, a framework that combines a large-scale cross-platform dataset with a family of open-source agent models. Their approach integrates automated exploration with human expert trajectories, creating a diverse and scalable training corpus. The resulting models unify perception, reasoning, and action, and support multiple inference modes for flexible deployment.

**Strengths:**

1. Introduces a large-scale, open dataset spanning six platforms (Windows, macOS, Linux, Android, iOS, Web) and three task domains (GUI understanding, grounding, and task completion), collected through a hybrid pipeline of automated agents and human experts.

2. Develops a suite of base agent models that unify perception, reasoning, and action, supporting three inference modes (Grounding, Direct Action, Reasoned Action) within a unified action space for seamless cross-platform interaction.

3. Demonstrates strong performance across multiple benchmarks, achieving new state-of-the-art results and showing that scaling diverse GUI-specific data significantly improves general-purpose computer use agents.

**Weaknesses:**

1. The dual-loop data collection pipeline (agent + human) is presented as a key innovation, but similar hybrid strategies have been explored in prior works such as OS-Genesis (Sun et al., 2024b) and AGUVIS (Xu et al., 2024). The paper risks being perceived as a scale-up rather than a conceptual breakthrough.

2. A large fraction of trajectories come from random-walk or weakly semantic exploration, which may not reflect realistic user goals. This could bias the model toward superficial navigation patterns rather than meaningful task completion.

3. The paper reports strong results but lacks fine-grained ablations. For instance, it is unclear how much each component (weak semantic trajectories, reasoning data, augmentation strategies) contributes to final performance. Similarly, the trade-offs between the three inference modes (Grounding, Direct, Reasoned) are not deeply analyzed.

**Questions:**

While ScaleCUA spans six platforms, it is unclear how well it generalizes to unseen applications or domains. For example, can a model trained on productivity and web apps adapt to enterprise software or creative tools?

---

> ### Author Response · Authors · 2025-11-19
>
> ## Q1: Clarification on our data pipeline.
> **Compared with related work.** We appreciate the reviewer’s insightful comment. We aim to clarify the distinctions and the practical advances achieved by our work. First, OS-Genesis focuses solely on mobile and web domains and relies heavily on automated synthesis, which results in substantial annotation noise. In contrast, our dual-loop framework spans six heterogeneous platforms (Windows, macOS, Linux, Android, iOS, and Web), and constructs a dataset that covers a broader range of tasks, including understanding, grounding, and task completion. Additionally, our human-agent collaboration further ensures the quality of the data.
> AGUVIS, on the other hand, integrates and reformats existing datasets but does not perform new trajectory collection. Our pipeline thus extends beyond data integration, enabling both automated and expert-curated acquisition under a unified infrastructure.
>
> **Clarification on practical contributions.** Although our collection framework is conceptually straightforward, executing it across heterogeneous operating systems and software ecosystems entails substantial non-trivial engineering. Each platform introduces distinct GUI structures, metadata formats, and failure patterns. We systematically addressed numerous corner cases, such as UI element occlusions,  wrong/absent accessibility attributes, repeated exploration, and annotation hallucinations, which would otherwise interrupt large-scale acquisition.
> These practical issues, though normally overlooked in prior work, motivate us to design a comprehensive heuristic algorithm for agent-driven data collection, which is crucial for scaling computer-use data collection from multiple platforms.
> In addition, we document these corner cases with solutions in detail in Appendix A.7 & A.9 (original paper) as a reproducible guide for the community. As a result, our contribution extends beyond the design of the pipeline itself, but it also lies in our detailed documentation and analysis of the classic corner cases encountered throughout the data construction process.
> Our contributions in data pipeline are threefold:
> 1. We propose a robust and scalable data acquisition pipeline that balances automation and expert supervision, along with a set of effective heuristics improving data diversity and quality.
> 2. We summarize a comprehensive guideline covering platform-specific issues and their resolutions in the appendix, which significantly improves the purity and efficiency of data collection.
> 3. We commit to releasing all data, ensuring transparency and reusability for future research.
>
> Generally, our work delivers a practically validated, multi-platform solution addressing real-world bottlenecks in scaling computer-use agents. Despite involving many engineering-oriented efforts, we still aim to share these experiences, expecting they can provide valuable guidance for future developments in this field.
>
> ## Q2: The significance of weak semantic exploration
> We thank the reviewer for raising this important concern.
>
> In our view, a complex task is inherently composed of alternating navigation and manipulation sub-processes: the agent first locates the target interface and then performs the corresponding operation, repeating this cycle until the goal is achieved. From this perspective, weakly semantic trajectories still provide valuable supervision for learning local navigation and interaction patterns, which are indispensable components of long-horizon task execution.
>
> We acknowledge that an excessive proportion of weakly semantic trajectories may bias the model toward surface-level navigation behaviors, thereby limiting its ability to accomplish high-level goal-directed tasks. Nonetheless, high-quality trajectories for task completion remain extremely scarce and costly to collect at scale. This motivates our empirical study on how weakly semantic trajectories can contribute to overall task completion. As shown in Table 4(b), we observe measurable performance gains when incorporating these trajectories, suggesting that they still play a positive role in improving general action grounding and low-level decision consistency.
>
> In summary, while weakly semantic trajectories are not a perfect substitute for human expert demonstrations, they offer a practical and scalable means to enhance foundational capabilities such as visual grounding and interface understanding. We consider them a complementary source of supervision in the current data-limited regime, and future work will explore more balanced strategies to combine weak and goal-conditioned data for stronger high-level task reasoning.

---

> > ### Author Response · Authors · 2025-11-19
> >
> > ## Q3: Fine-grained ablations
> > In our original paper, we have provided thorough ablations on our data, eg., Fig. 1 and Table 4 in the Appendix (original paper). The following experiments except the ablation on reasoning data can be found in the appendix of the original submission. In the latest version of the paper, we have moved this section from the appendix to the main text.
> > ### (a) The ablation on reasoning data
> > We provide an ablation study on the reasoning data, and the result demonstrates the effectiveness of the reasoning data.
> > | Model           | Training Data | Reasoning data | OS-World |
> > |-----------------|---------------|----------------|----------|
> > | Qwen2.5VL-3B    | ours-only     | ❌             | 6.5      |
> > | Qwen2.5VL-3B    | ours-only     | ✅             | 10.3     |
> >
> > ### (b) The efficacy of the whole dataset
> > As shown in Fig. 1, training with our curated training corpus yields consistent improvements over the baseline trained on public data.
> >
> > ### (c) The ablation on data augmentation
> > The results in Table 4a verify that data augmentation can improve performance by 3.5% on ScreenSpot-Pro.
> > This confirms that augmentation enhances generalization and robustness by exposing the model to a broader range of visual conditions.
> >
> > ### (d) The ablation on weak semantic trajectories
> > Second, in Table 4b, we investigate weak semantic trajectories derived from rule-based random exploration. Despite lacking explicit high-level goals, these trajectories provide low-cost supervision of interface navigation.
> >
> > The ablation studies across UI element grounding and task completion demonstrate that the design of training data is the key to building scalable and generalizable CUAs.
> > If you still have concerns on our data abaltions, we are glad to futher discuss with you.
> >
> >
> > ## Q4: Deep analysis of three inference modes.
> > Thanks for your advice. We here give a deep analysis of three inference modes and update it into our final paper.
> > Specifically, Fig. 5 (b) compares the two inference modes for computer use agents. Across all benchmarks, reasoned action mode (RAM) yields higher success rates than direct action mode (DAM), with absolute gains ranging from +1.4% to +8.2%. However, RAM also incurs longer inference time and greater token cost. DAM, in contrast, produces actions directly from the visual–textual context, yielding faster responses but being more prone to cumulative drift in long-horizon tasks. In Table 3, when ScaleCUA-7B as a grounding model is integrated with GPT-4o under an agentic workflow, it shows higher success on task completion benchmarks than the reasoned action mode (e.g., 28.1% vs. 15.0% on OSWorld, 36.6% vs. 20.7% on WindowsAgentArena, etc.). The agentic workflow allows GPT-4o to handle long-context planning while leveraging grounding mode in ScaleCUA, demonstrating complementarity between ScaleCUA and general VLMs. Nevertheless, this paradigm cannot generate actions in an end-to-end manner and brings higher costs even than RAM.
> >
> > We summarize the pros and cons of three modes for trade-offs:
> > 1) Direct Action Mode (DAM): cheapest and fastest; emits actions directly from vision–text context. Best for short, stable tasks and high-throughput evaluations. Vulnerable to cumulative drift on long horizons and ambiguous goals.
> > 2) Reasoned Action Mode (RAM): inserts a rationale before actions, yielding higher success. More robust to error propagation and ambiguity; interpretable. Costs extra tokens and latency.
> > 3) Grounding Mode + external planner (agentic workflow): planner handles long-context reasoning and tool knowledge; the grounder ensures precise UI localization and execution. It often achieves the strongest task-completion scores on complex, multi-step desktop/web tasks, but also leads to a higher coordination cost, response latency, and a lack of end-to-end optimization.
> >
> > Generally, choose DAM for speed/cost, RAM for reliability on medium-to-long tasks, and Planner+Grounder for knowledge-heavy, safety-critical, or long-horizon workflows.

---

> > > ### Author Response · Authors · 2025-11-19
> > >
> > > ## Q5: Agent generalization to unseen applications or domains
> > > We thank the reviewer for highlighting this important question.
> > >
> > > Our view is that, without additional adaptation, native agents, including ScaleCUA, face clear challenges when encountering unseen applications. The more specialized the software is, the worse the model tends to perform. Empirically, ScienceBoard includes a collection of highly specialized and niche software, which are essentially zero-shot for almost all models. We can observe that all methods consistently perform poorly on this benchmark (refer to Table 11).
> > >
> > > However, we see two practical directions to improve out-of-domain generalization to unseen applications and enterprise environments. 1) Offline knowledge database, which constructs a large-scale tutorial and documentation corpus for professional software. Many enterprise and scientific tools provide manuals, official docs, or papers. These resources can be utilized by retrieval-augmented generation (RAG) at inference time to guide agents operating on unseen applications. 2) Online retrieval, which adopts a web search plus RAG workflow. The web contains abundant usage knowledge for software; when coupled with robust long-context understanding and faithful instruction following, retrieval can supply missing domain knowledge on the fly. 3) Autonomous self-evolution, which enables the agent to actively explore the unseen environment through trial-and-error. By interacting with the interface and observing feedback (e.g., GUI state changes or error logs), the agent can maintain its own offline knowledge database and reflect on execution failures. This iterative process allows the agent to progressively accumulate procedural experience and adapt its policy to the specific logic of the new application, mirroring how humans learn software by "playing" with it.
> > >
> > > We hope our response can address your concerns.

---

### Official Review · Reviewer_AxPt · 2025-11-08

**Soundness:** 3
**Presentation:** 3
**Contribution:** 3
**Rating:** 6
**Confidence:** 4

**Summary:**

The authors build a large-scale training dataset for GUI agents, including 6 operating systems and 3 task domains, and develop the ScaleCUA series of models. The authors propose an automatic method to collect trajectories and grounding annotations and build a model that can perform well on GUI understanding, grounding, and planning. The authors benchmark ScaleCUA on a number of tasks to show its efficacy.

**Strengths:**

1. The authors curate a large-scale dataset for GUI agents. This includes understanding, grounding, and planning data. The data consists of several software applications across diverse platforms. The authors claim to release the data openly, which will be very valuable.
2. The authors conduct a comprehensive set of experiments. They evaluate on several benchmarks across different tasks. They also provide several analyses on data scaling, the effect of screenshot resolutions, and several other design decisions. These provide good insights for future model development.
3. The authors provide practical guidance for collecting data from various platforms in the Appendix, which is very insightful.

**Weaknesses:**

1. The data collection introduced by the authors consists of an automatic method where rule-based agents and heuristics are used, and another component that comprises human-created tasks and trajectories. The presentation of this section gives a misleading impression that the trajectories are created by both humans and agents collaboratively, but in actuality it is some trajectories created by agents and others created entirely by humans. I fail to see any new method or innovation here (besides the heuristics that the authors develop for rule-based agents). Could the authors please provide clarification for this?
2. The authors do extensive evaluation, but compare with a limited set of models. Several recent models, like [1], [2], [3] use RL to train their models and show that they can achieve impressive performance using public data. I understand that the ScaleCUA models are only SFTed and not RL finetuned, but since models are also highlighted as one of the contributions of this work, comparing them more extensively would further show the effectiveness of their models.
3. The authors should conduct a more controlled comparison between their data and open source datasets like Jedi. The authors currently compare the benefits of using their data over public data, but it is not clear if the quantity of data used for this comparison is the same or not. The authors could take a fixed random sample of data from different datasets like Jedi, OsAtlas etc., and fine tune the same model to show whether the data collected by the authors provides more gains than already available datasets.

[1] Yang et al. Gta1: Gui test-time scaling agent

[2] Liu et al. Infigui-g1: Advancing gui grounding with adaptive exploration policy optimization

[3] Tang et al. Gui-g2: Gaussian reward modeling for gui grounding.

**Questions:**

1. On online evaluation tasks, especially OSWorld, ScaleCUA underperforms compared to OpenCUA and a few other models despite being trained on large quantities of data. Could the authors comment on this? This is especially relevant because it raises questions about the effectiveness of the data collection method.

Please see the weakness for more questions and suggestions.

---

> ### Author Response · Authors · 2025-11-19
>
> ## Q1: Clarification on data collection.
> Human experts are also involved in the agent's heuristic data collection process. After agents traverse GUI pages, we analyze the collected trajectories to identify corner cases and iteratively refine our heuristic algorithms.
> Moreover, 20% of the agent-collected data were randomly sampled and reviewed by human experts both before and after annotation to ensure quality. As such, this is what we refer to as hybrid data acquisition in our paper. Next, we make a thorough clarification to address your concern.
>
> **Clarification on practical contributions.** Although our collection framework is conceptually straightforward, executing it across heterogeneous operating systems and software ecosystems entails substantial non-trivial engineering. Each platform introduces distinct GUI structures, metadata formats, and failure patterns. We systematically addressed numerous corner cases, such as **UI element occlusions**,  **wrong/absent accessibility attributes**, **repeated exploration**, and **annotation hallucinations**, which would otherwise interrupt large-scale acquisition.
> These practical issues, though normally overlooked in prior work, motivate us to design comprehensive heuristic algorithm for agent-driven data collection, which are crucial for scaling computer-use data collection from multiple platforms.
> In addition, we document these corner cases with solutions in detail in Appendix A.7 & A.9 (original paper) as a reproducible guide for the community. As a result, our contribution extends beyond the design of the pipeline itself, but it also lies in our detailed documentation and analysis of the classic corner cases encountered throughout the data construction process.
> Our contributions in data pipeline are threefold:
> 1. We propose a robust and scalable data acquisition pipeline that balances automation and expert supervision, along with a set of effective heuristics improving data diversity and quality.
> 2. We summarize a comprehensive guideline covering platform-specific issues and their resolutions in the appendix, which significantly improves the purity and efficiency of data collection.
> 3. We commit to releasing all data, ensuring transparency and reusability for future research.
>
> Generally, we emphasize that *our work delivers a practically validated, multi-platform solution addressing real-world bottlenecks in scaling computer-use agents*. Despite involving many engineering-oriented efforts, we still aim to share these experiences, expecting they can provide valuable guidance for future developments in this field.
>
> ## Q2: Comparisons with more models.
> We agree with your view. In fact, we have compared our method with GTA1 (results of v1 on arXiv) and InfiGUI-G1 in Table 6 and Table 7 (in the Appendix of the Original paper), respetively. We further reorganize tables to provide thorough comparisons with related work, including recent GUI-G^2.
>
> ## Q3: Data validity verification
> Following the reviewer’s advice, we perform additional experiments using Qwen2.5VL-3B under a fixed data volume setting to ensure a fair comparison. Specifically, we selected three representative grounding datasets: (1) OS-Atlas, which contains large-scale GUI grounding data, but we found it exhibits a notable amount of annotation noise. The reported results are based on its raw data, while our own training in the paper used a re-filtered subset to mitigate label errors. (2) JEDI, a synthetic dataset of high internal consistency but limited realism due to its synthetic rendering process and the resulting domain gap from real GUI environments. (3) Aguvis, which integrates and cleans several widely used public datasets with high average quality and balanced coverage. For all datasets, we fixed the number of samples to min(JEDI, OS-Atlas, Aguvis, ours) based on the actual number of elements used in our training pipeline, rather than their raw dataset size. Here, we use the 550K elements that are the size of JEDI we employed in training our ScaleCUA.
>
> | Dataset    | SS v2 | SS-Pro |
> |------------|-------|--------|
> | JEDI       |74.4  | 27.9   |
> | OS-Atlas   |76.0  | 22.6   |
> | Aguvis     |81.2  | 30.7   |
> | ours       |83.6  | 38.1   |
>
> As shown, our dataset still achieves superior performance gains under the same data budget. In our view, the value of a dataset lies not only in absolute improvement on performance but also in complementarity. Even if different datasets have varying biases, a new dataset that introduces distinct coverage and diversity can make a tangible positive contribution to advancing the research for CUAs.

---

> > ### Author Response · Authors · 2025-11-19
> >
> > ## Q4: Performance comparisons with other competitive models.
> > We appreciate the reviewer’s insightful question. We do not avoid comparisons with these excellent works and are very willing to discuss the performance gap.
> >
> > First, a key factor lies in the amount of high-quality human trajectory data. For instance, OpenCUA collected and annotated over 41k human-curated trajectories through crowd-sourcing platforms, while UI-TARS is also known in the community for investing heavily in manually curated data. In contrast, our dataset emphasizes on large-scale grounding and understanding data, which results in strong performance on these tasks. However, scaling human-curated trajectories to a higher level remains challenging and thus there is slight performance gap in online task completion benchmarks.
> >
> > Second, our model is designed as a multi-platform agent spanning desktop, mobile, and web domains. This multi-domain unification introduces inevitable conflicts during joint training, as different platforms exhibit distinct GUI layouts, metadata structures, and interaction semantics. We have analyzed these effects in the main paper, showing that single-domain models sometimes achieve higher task success but at the expense of generality.
> >
> > Finally, we stress that even if ScaleCUA does not achieve the best score on every benchmark, our work provides a systematic exploration of scalable data collection and multi-platform training, which offers empirical insights, rather than focusing on leaderboard performance. We believe that as the community continues to expand data and model availability, the performance gap between open-source and proprietary systems will narrow.
> >
> > If you have any further questions, we are willing to share all relevant details without reservation.

---

> > > ### Comment · Reviewer_AxPt · 2025-11-27
> > >
> > > Thank you for providing clarifications. I believe this is good work and have updated the scores to reflect this.
> > >
> > > It would be great if the authors, in addition to the data, could release practical guidelines, code, and tools they use to collect their data (eg, how they set up environments to allow agents to collect data, etc.). That could help the community collect more data using the proposed method.

---

> ### Author Response · Authors · 2025-11-28
> **Thank you for your encouraging comments!**
>
> Thank you for your encouraging comments!
>
> We fully agree with your suggestions. To achieve this, we remain committed to contributing to the open-source community and will strive to build a more open and comprehensive ecosystem for computer-using agents.
>
> Again, we sincerely appreciate your insightful reviews and your recognition for our work.

---

### Author Response · Authors · 2025-11-19
**We appreciate the reviewers' recognition of our work and their constructive feedback.**

# We appreciate the reviewers' recognition of our work and their constructive feedback.
We sincerely thank all reviewers for their thoughtful and constructive feedback on our submission. The positive recognition of our work is encouraging, including **comprehensive experiments**, **interesting key insights**, **well written**, **potential to further the research**, etc.

We aim to address reviewers' concerns, which will further make the work more rigorous and complete. In the following, we respond to each comment in detail and revise the manuscript accordingly to improve both the technical soundness and clarity of our contributions.

According to reviews, the submission is also updated with the following changes:
1. **Human Expert Data Collection Details:** In Section 3.1, additional details have been provided regarding the data collection process involving human experts. Furthermore, the discussion in Section 3.2 now highlights the contributions of the data pipeline, particularly addressing concerns regarding the hybrid nature of data collection (human vs. autonomous agent).
2. **Revised Structure of Appendix:** The section on "Ablation on Data" (previously in Appendix A.2) has been moved to the main text (Section 5.3). This addresses the reviewer's concern about data efficacy, weak semantic trajectories, and data augmentation by providing more visibility and depth on these issues.
3. **Clarification on Experimental Analysis:** Based on reviewer suggestions, Section 5.2 has been revised to clarify aspects of the experimental analysis. This includes further elaboration on the impact of input resolution, different inference modes (Grounding, Direct Action, Reasoned Action), and the effects of data scaling, addressing concerns about the trade-offs in these factors.
4. **Expanded Comparison of Methods:** Appendix A.2 (formerly A.3) now includes a more detailed comparison of methods, incorporating new benchmarks and reorganizing tables to facilitate a clearer and more comprehensive comparison. This change responds to the reviewer’s request for a more comprehensive comparison.
5. **Proofreading:** We have thoroughly revised our paper to address grammar, typographical errors, and other writing issues.

We welcome continued discussion and feedback from the reviewers to further improve this work.

---

> ### Author Response · Authors · 2025-11-24
>
> Dear reviewers,
>
> We would like to once again thank you for the time and effort you have devoted to reviewing our submission. We have now responded in detail to all of your comments and revised the manuscript accordingly. If there are any remaining questions or points that would benefit from further clarification, we would be very happy to continue the discussion during the rebuttal period. If our responses have satisfactorily addressed your concerns, we would be sincerely grateful if you could consider raising your scores to reflect your current assessment.
>
> Best regards,
>
> The authors

---

### Author Response · Authors · 2025-12-03
**Summary of Review Status to Area Chairs**

Dear Area Chairs,

We appreciate the efforts from the reviewers, Area Chairs, and the Conference Committee during the review process. Given this situation, we would like to provide a concise summary of our rebuttal progress to assist in your decision-making.

## Statement of Integrity

First and foremost, we have strictly adhered to the ICLR double-blind policy and Code of Ethics throughout the review process. We have engaged in no collusion or communication that would compromise the anonymity or fairness of the review.

## Rebuttal Updates
Reviewers provided constructive feedback on our submission with encouraging comments, such as its "**large-scale dataset spanning six platforms**" (Reviewer ojRP) and "**comprehensive suite of ablation studies exploring design choices**" (Reviewer TkSU). Reviewer AxPt highlighted that the "**practical guidance for collecting data... is very insightful**" and provides "**good insights for future model development**". Moreover, Reviewer ZN1J noted the paper is "**well written, clearly structured**". The models were praised for achieving "**impressive results on several challenging benchmarks**" (Reviewer TkSU). Finally, Reviewer 56qp emphasized that "**comprehensive dataset and model... have the potential to further the research**".

During the rebuttal period, we addressed all reviewers' concerns with additional experiments and clarifications. Our scores had risen from 10, 6, 6, 6, 6 to 10, 8, 6, 6, 6 (before rolling back), demonstrating the reviewers' recognition of our work.
We would like to highlight that Reviewer AxPt (commented on 28 Nov 2025, 01:17, after data leakage) acknowledged our rebuttal and raised the score from 6 to 8. While the rebuttal period is technically ongoing, **we understand that due to the data leakage incident, other reviewers may have been unable to respond or update their scores in time**. We fully respect the constraints caused by this unforeseen event.

## Summary of Our Contributions

1. **A Scalable Data Pipeline with Systematic Exploration.** We design a robust, multi-platform data pipeline that synergizes automated agent exploration with human-expert curation.
A critical and distinctive contribution to our work is the systematic exploration of how human-agent collaboration can be leveraged to iteratively optimize the heuristic algorithms. Moreover, in this process, we addressed platform-specific "corner cases" during data collection across six heterogeneous operating systems, reducing noise and improving UI coverage.
As detailed in the paper, we identified substantial engineering challenges, ranging from UI element occlusion and API limitations to annotation hallucinations. We have documented these failure patterns with resolutions as a comprehensive guideline, providing a reproducible roadmap for the community.

1. **ScaleCUA Model Family with Flexible Inference Paradigms.** We developed the ScaleCUA model family (3B, 7B, 32B), which supports a unified action space for seamless multi-platform interaction. A key highlight is the flexibility of our inference paradigms: we offer a deep analysis of Direct Action Mode (for efficiency), Reasoned Action Mode (for reliability), and Grounding Mode (for modular integration). This design allows for adaptable trade-offs between latency and reasoning depth, providing valuable guidance for future model development.

2. **Rigorous Ablation Studies with Diagnostic Analysis.** We conducted comprehensive ablations, including data composition\augmentation, and training settings to dissect the factors driving agent performance. Furthermore, we observe monotonic performance gains across grounding and online benchmarks as data volume scales, but this also highlights the ongoing need for large-scale datasets in desktop environments. Finally, our test-time scaling analysis reveals clear trade-offs between inference cost and efficiency, providing an in-depth examination of the pros and cons for each paradigm.

3. **Comprehensive Benchmarking on 3 Task Domains and 5 Online Platforms.** We performed a holistic evaluation spanning GUI understanding, grounding, and end-to-end task completion to validate the robust capabilities of our models. Specifically, for task completion, we evaluate the agent's capability on multiple platforms, including mobile, Ubuntu, Windows, macOS, and Web. Overall, ScaleCUA demonstrates consistent improvements across multiple challenging benchmarks, confirming the effectiveness of our data and training strategies.

4. **Democratizing CUA Research through Open Assets.** We commit to fully open-sourcing our dataset, models, and code. By providing these resources alongside our practical guidelines, we aim to foster transparency and accelerate future research in general-purpose computer automation.

We believe our work provides a solid foundation for future research in general-purpose CUAs. Thank you again for your time and dedication to the review process.

Sincerely,

The Authors

---

### Meta-Review · Area_Chair_esHK · 2026-01-04

**Summary:**

All reviewers gave positive scores in the initial review (10, 6, 6, 6, 6), during the rebuttal one reviewer with score 6 increases to 8. These scores demonstrate that all reviewers think this paper is suitable for the publication. I also read this paper and think the contributions in this paper are solid. Scaling is important, though sometimes most efforts are engineering, but still important.

**Reviewer Concerns:**

The main reviewer concerns centered on whether the work represents a conceptual breakthrough or merely a scale-up of existing hybrid data collection methods (similar to OS-Genesis and AGUVIS), the potential bias introduced by weak semantic trajectories from random exploration, extreme dataset imbalance across platforms (MacOS only 0.6%), and lack of fine-grained ablations showing individual component contributions.

In the rebuttal, the authors emphasised that while the pipeline concept is straightforward, executing it across six heterogeneous platforms involved substantial non-trivial engineering to address platform-specific corner cases (UI occlusions, accessibility API limitations, annotation hallucinations), which they documented as reproducible guidelines for the community. They provided additional controlled experiments demonstrating their data outperforms existing datasets (JEDI, OS-Atlas, Aguvis) under fixed data budgets, showed that weak semantic trajectories do provide measurable performance gains for low-level navigation tasks, justified the platform imbalance as reflecting real-world accessibility constraints and collection difficulty, and added fine-grained ablations in the revised manuscript.

**Reviewer Scores:**

Reviewer AxPt changes the score from 6 to 8 during the rebuttal. For others, I am not sure whether they will increase the scores, but never mind, they are still positive reviews.

---

### Decision · Program_Chairs · 2026-01-26

Accept (Oral)